# Algorithm Configuration for Structured Pfaffian Settings

**Maria Florina Balcan**  *ninamf@cs.cmu.edu*
*School of Computer Science*
*Carnegie Mellon University*

**Anh Tuan Nguyen**  *atnguyen@cs.cmu.edu*
*Machine Learning Department*
*Carnegie Mellon University*

**Dravyansh Sharma***  *dravy@ttic.edu*
*Toyota Technological Institute at Chicago*

Reviewed on OpenReview: *https://openreview.net/forum?id=Xmk1or5eH8*

## Abstract

Data-driven algorithm design uses historical problem instances to automatically adjust and optimize algorithms to their application domain, typically by selecting algorithms from parameterized families. While the approach has been highly successful in practice, providing theoretical guarantees for several algorithmic families remains challenging. This is due to the intricate dependence of the algorithmic performance on the parameters, often exhibiting a piecewise discontinuous structure. In this work, we present new frameworks for providing learning guarantees for parameterized data-driven algorithm design problems in both statistical and online learning settings.

For the statistical learning setting, we introduce the *Pfaffian GJ framework*, an extension of the classical *Goldberg-Jerrum (GJ) framework* (Bartlett et al., 2022; Goldberg & Jerrum, 1993), that is capable of providing learning guarantees for function classes for which the computation involves Pfaffian functions. Unlike the GJ framework, which is limited to function classes with computation characterized by rational functions (quotients of two polynomials), our proposed framework can deal with function classes involving Pfaffian functions, which are much more general and widely applicable. We then show that for many parameterized algorithms of interest, their utility function possesses a *refined piecewise structure*, which automatically translates to learning guarantees using our proposed framework.

For the online learning setting, we provide a new tool for verifying the *dispersion* property of a sequence of loss functions, a sufficient condition that allows no-regret learning for sequences of piecewise structured loss functions where the piecewise structure involves Pfaffian transition boundaries. We use our framework to provide novel learning guarantees for many challenging data-driven design problems of interest, including data-driven linkage-based clustering, graph-based semi-supervised learning, and regularized logistic regression.

## 1 Introduction

*Data-driven algorithm design* (Ailon et al., 2011; Gupta & Roughgarden, 2016; Balcan, 2020) is a modern approach that develops and analyzes algorithms based on the assumption that problem instances come from an underlying application domain. For example, a news website might have its own patterns in how bots and humans interact with their content, and this distinctive information can be leveraged to tune clustering algorithms that distinguish automated crawls from human access patterns. Unlike traditional worst-case or average-case analyses, this approach leverages observed problem instances to design algorithms that achieve

---

*Most of the work was done when DS was at CMU.

high performance for specific problem domains. For many applications, say combinatorial partitioning problems (Balcan et al., 2017) or mixed integer linear programming (Balcan et al., 2018a), algorithms are often parameterized, meaning they are equipped with tunable hyperparameters which significantly influence their performance. We develop general techniques for establishing learning guarantees for data-driven algorithm selection by learning the parameters from historical problem instances. Prior approaches for obtaining concrete learning guarantees based on the classical Goldberg-Jerrum (GJ) (Goldberg & Jerrum, 1993; Bartlett et al., 2022) framework are limited to families where, roughly speaking, the algorithmic performance as a function of the hyperparameters has a piecewise polynomial structure. Our techniques apply to a larger variety of parameterized algorithm families characterized by piecewise Pfaffian functions, which is a broader class of functions that also includes exponential and logarithmic functions.

Applications of data-driven algorithm design span various fundamental areas. These include low-rank approximation (Indyk et al., 2019; 2021; Li et al., 2023), sparse linear systems solvers (Luz et al., 2020), dimensionality reduction (Ailon et al., 2021), and many more fundamental problems. This empirical success underscores the necessity for a theoretical understanding of this approach. Intensive efforts have been made towards theoretical understanding for data-driven algorithm design, including learning guarantees for numerical linear algebra methods (Bartlett et al., 2022), tuning regularization hyperparameters for regression problems (Balcan et al., 2022a; 2023b), unsupervised and semi-supervised learning (Balcan et al., 2018c; 2020a; Balcan & Sharma, 2021), applications to integer and mixed-integer programming (Balcan et al., 2018a; 2021c).

Prior theoretical work on data-driven algorithm design focuses on two main settings: statistical learning (Balcan et al., 2021a; Bartlett et al., 2022) and online learning (Balcan et al., 2018b). In the statistical learning setting, there is a learner trying to optimize hyperparameters for the algorithm within a specific domain. The learner has access to problem instances from that domain, which are assumed to be drawn from a fixed but unknown problem distribution. In this case, a key question is understanding the sample complexity, i.e. answering the following question: how many problem instances are required to learn hyperparameters that yield good expected performance on future unseen problem instances? In the online learning setting, there is a sequence of problem instances chosen by an adversary arriving over time. The goal now is to design a no-regret learning algorithm: adjusting the algorithm's hyperparameters on the fly so that the difference between the average utility of the learner and the utility corresponding to the best fixed hyperparameters in hindsight, that is, the learner's regret, diminishes over time. We develop tools to answer both these questions, applicable to a broad class of problems.

**Technical formulation.** Typically, the performance of an algorithm is evaluated by a specific utility function, measuring its run-time, memory, or solution quality, for example. More formally, for an algorithm with input instance space $\mathcal{X}$ and parameterized by a parameter class $\mathcal{A}$, consider the utility function class $\mathcal{U} = \{u_{\boldsymbol{a}} : \mathcal{X} \to [0, H] \mid \boldsymbol{a} \in \mathcal{A}\}$, where $u_{\boldsymbol{a}}(\boldsymbol{x})$ gauges the performance of the algorithm with hyperparameters $\boldsymbol{a}$ when run on a problem instance $\boldsymbol{x} \in \mathcal{X}$, and $H$ is an upper-bound for the utility value. In the statistical learning data-driven algorithm design setting, we assume an unknown underlying distribution $\mathcal{D}$ over $\mathcal{X}$, representing the application domain on which the algorithm operates. In this setting, designing parameterized algorithms tailored to a specific domain corresponds to the optimal selection of hyperparameters $\boldsymbol{a}$ for the given application-specific distribution $\mathcal{D}$.

The main challenge in establishing learning guarantees for the utility function classes lies in the complex structure of the utility function. In other words, even a minimal perturbation in $\boldsymbol{a}$ can lead to a drastic change in the performance of the algorithm, making the analysis of such classes of utility functions particularly challenging. In response to this challenge, prior work takes an alternative approach by analyzing the dual utility function class $\mathcal{U}^* = \{u_{\boldsymbol{x}}^* : \mathcal{A} \to [0, H] \mid \boldsymbol{x} \in \mathcal{X}\}$. Each function in the dual function class often admits piecewise structured behavior (Balcan et al., 2021a; 2017; Bartlett et al., 2022).

Building upon this observation, in the statistical learning setting, Balcan et al. (2021a) propose a general approach that analyzes the learnability of the utility function class via the learning-theoretic complexity of the piece and boundary function classes induced by the piecewise structure of the dual. Approaching this from an alternative perspective, Bartlett et al. (2022) introduced a new version of the *GJ framework*, where the piecewise structure of the dual function is expressed as a tree of computations involving fundamental

arithmetic operations at the nodes and branches according to simple conditional statements. The complexity of the operations and the expressions that appear across the tree are shown to be related to the sample complexity of parameter tuning. Despite their broad applicability, these general frameworks have inherent limitations. In the statistical learning setting, the framework introduced by Balcan et al. (2021a) reduces the problem of computing the learning-theoretic complexity of a piecewise structured utility function to bounding the complexity of the corresponding piece and boundary functions, but this might be challenging to compute for certain function classes (see e.g., Bartlett et al. 2022). On the other hand, the GJ framework instantiated by Bartlett et al. (2022) is limited to the cases where the computation of utility functions only involves rational functions (i.e., ratios of polynomials) of the hyperparameters.

In the online learning setting, prior work has similar limitations. Balcan et al. (2018b) introduce *dispersion*, which is a sufficient condition for no-regret learning of piecewise Lipschitz functions. Essentially, the dispersion property implies that if the discontinuities of utility function sequences do not densely concentrate in any small region of the hyperparameter space, then no-regret learning is possible. However, the dispersion property is generally challenging to verify (Balcan et al., 2018b; 2020b; Balcan & Sharma, 2021), and requires further assumptions on the discontinuities of the utility function sequence. Moreover, when the form of the discontinuities goes beyond affine and rational functions, prior techniques for verifying dispersion no longer apply.

Motivated by the limitations of prior research, part of this work aims to present theoretical frameworks for data-driven algorithm design when the utility function admits a specific structure. In the statistical learning setting, we introduce a powerful *Pfaffian GJ framework* that can be used to establish learning guarantees for function classes whose discontinuity involves *Pfaffian functions* (Definition 4). Roughly speaking, the Pfaffian function class is a very general class of functions that captures a wide range of functions of interest, including rational, exponential, logarithmic, and combinations of these functions. Furthermore, we demonstrate that many data-driven algorithm design problems exhibit a specific *Pfaffian piecewise structure*, which, when combined with the Pfaffian GJ framework, can establish learning guarantees for these problems. In the online learning setting, we introduce a novel tool to verify the dispersion property, where the discontinuities of the utility function sequence involve Pfaffian functions, going beyond affine and rational functions.

Another aim of this work is to provide learning guarantees for several under-investigated data-driven algorithm design problems, where the piecewise structure of the utility functions involves Pfaffian functions. The problems we consider have been investigated in simpler settings, including data-driven agglomerative hierarchical clustering (Balcan et al., 2017; 2020a), data-driven semi-supervised learning (Balcan & Sharma, 2021), and data-driven regularized logistic regression (Balcan et al., 2023b). However, previous investigations have limitations: they either have missing results for natural extensions of the settings under study (Balcan et al., 2017; Balcan & Sharma, 2021), require strong assumptions (Balcan et al., 2020a), or solely consider statistical learning settings (Balcan et al., 2023b). Moreover, we emphasize that *the techniques used in prior work are insufficient and cannot be applied in our settings*, which involve Pfaffian function analysis.

By carefully analyzing the utility functions associated with these problems, we uncover their underlying Pfaffian structure and carefully control the corresponding Pfaffian complexity, which allows us to leverage our proposed framework to establish learning guarantees. It is important to note that analyzing the Pfaffian structure for specific problems poses a significant challenge. A loose estimation of the Pfaffian function complexity when combined with our proposed framework would still lead to weak learning guarantees.

**Overview of the Pfaffian GJ framework.** A *Pfaffian GJ algorithm* takes as input real-valued parameters $\boldsymbol{a} \in \mathcal{A} \subseteq \mathbb{R}^d$. Intuitively, the computation of the dual function $u_{\boldsymbol{x}}^*$ can be expressed via a tree using only certain kinds of operations (including Pfaffian functions). More precisely, for each input $\boldsymbol{x} \in \mathcal{X}$ and each real threshold $r$, one constructs a fixed binary tree $T_{\boldsymbol{x},r}$ that can compute $\text{sign}(u_{\boldsymbol{x}}^*(\boldsymbol{a}) - r)$ for each $\boldsymbol{a} \in \mathcal{A}$. Each node $\nu$ of the tree corresponds to a fixed function evaluation $f_\nu$, involving binary arithmetic operations (i.e. one of $\{+, -, \times, \div\}$) or a Pfaffian function, and the arguments include the parameters $\boldsymbol{a} = (a_1, \ldots, a_d)$ and previously computed values on the path from the root to the node. The left and right children of $\nu$ correspond to $f_\nu \geq 0$ and $f_\nu < 0$ respectively. The tree can be used to compute $\text{sign}(u_{\boldsymbol{x}}^*(\boldsymbol{a}) - r)$ for any $\boldsymbol{a} \in \mathcal{A}$ by evaluating a series of expressions along some root-to-leaf path (see Figure 1 for a simple illustration). The Pfaffian GJ framework then corresponds to the following recipe for bounding the learning-theoretic com-

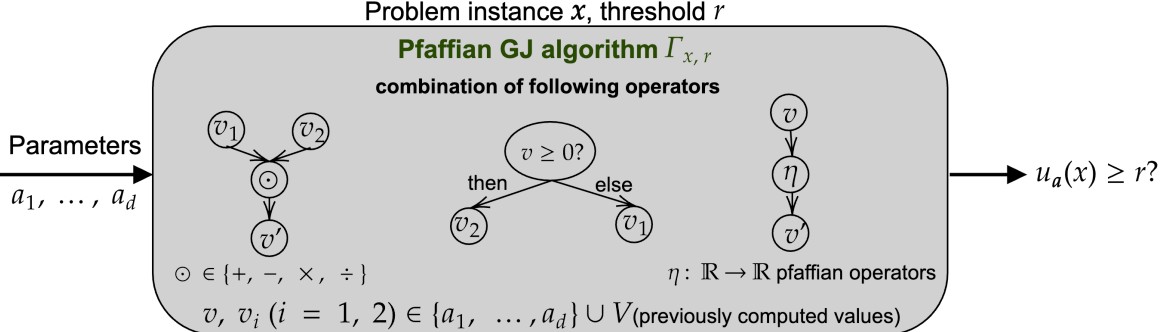

Figure 1: A simple illustration of the general idea of the Pfaffian GJ framework. Given a problem instance $\boldsymbol{x}$ and a real-valued threshold $r$, a Pfaffian GJ algorithm $\Gamma_{\boldsymbol{x},r}$ takes as the inputs any possible parameters $\boldsymbol{a} = (a_1, \ldots, a_d)$ and outputs if $u_{\boldsymbol{a}}(\boldsymbol{x}) \geq r$, by combining basic arithmetic operators, Pfaffian functions, and conditional statements. If we can bound the complexity of $\Gamma_{\boldsymbol{x},r}$, our results imply that we can bound the pseudo-dimension of the utility function class $\mathcal{U}$.

plexity of $\mathcal{U}$: Express the computation of the dual function $u_{\boldsymbol{x}}^*$ as a tree $T_{\boldsymbol{x},r}$ for any $\boldsymbol{x}, r$ and give bounds on the complexity of the tree (in terms of the number of distinct functional expressions across all nodes as well as the worst-case complexity of any functional expression). This automatically yields a bound on the sample complexity of tuning the parameters $\boldsymbol{a}$ using our results.

**Contributions.** In this work, we provide a new framework for theoretical analysis of data-driven algorithm design problems. We then investigate many under-investigated data-driven algorithm design problems, analyzing their underlying problem structure, and then leveraging our newly proposed framework to provide learning guarantees for those problems. Concretely,

- We present the Pfaffian GJ framework (Definition 5, Theorem 4.2), a general approach for analyzing the pseudo-dimension of various function classes of interest in the data-driven setting. This framework draws inspiration from the refined version of the GJ framework introduced by Bartlett et al. (2022). However, in contrast to the conventional GJ framework which is only capable of handling computation related to rational functions, the Pfaffian GJ framework can handle computations that involve Pfaffian functions (Definition 4)—a much broader function class—significantly increasing its applicability (see Figure 1 for an illustration). We note that our proposed Pfaffian GJ framework is of independent interest and can be applied to other research areas beyond data-driven algorithm design.

- For the statistical learning setting, we introduce a refined notion of piecewise structure (Definition 8, see Figure 2 for an example) for the dual utility function class, which applies whenever the piece and boundary functions are Pfaffian (see Figure 4). In contrast to the prior piecewise structure proposed by Balcan et al. (2021a); Bartlett et al. (2022), our framework can be used to obtain concrete learning guarantees when the piece and boundary functions belong to the class of Pfaffian functions, which includes widely used utility functions, including the exponential and logarithmic functions. We then show how the refined piecewise structure can be combined with the newly proposed Pfaffian GJ framework to provide learning guarantees (Theorem 5.2) for problems that satisfy this property.

- For online data-driven algorithm design, we introduce a general approach (Theorem 7.2, Theorem 7.3) for verifying the *dispersion* property (Balcan et al., 2018b)—a sufficient condition for online learning in data-driven algorithm design. Prior work (Balcan et al., 2020b; Balcan & Sharma, 2021) provides techniques for verifying dispersion only when the piece boundaries are algebraic functions. We significantly expand the class of functions for which online learning guarantees may be obtained by establishing a novel tool which applies for Pfaffian boundary functions.

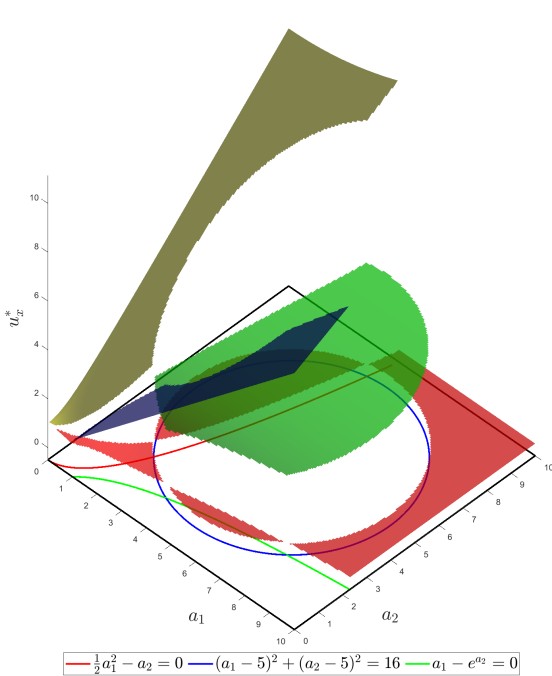

Figure 2: A simple illustration of Pfaffian piecewise structure. For each problem instance $\boldsymbol{x}$, there are Pfaffian boundaries that partition the space of parameters $\boldsymbol{a}$ into regions. In each region, the value of $u_{\boldsymbol{x}}^*(\boldsymbol{a})$ is a Pfaffian function. See Figure 3 for more details.

- We derive novel learning guarantees for a variety of understudied data-driven algorithm design problems, including data-driven agglomerative clustering (Theorem 6.1), data-driven graph-based semi-supervised learning (Theorem 6.4). We also recover the guarantee for data-driven regularized logistic regression in previous work by Balcan et al. (2023b). By carefully analyzing the underlying structure of the utility functions for these problems, we convert them into the form required in our proposed framework, automatically yielding learning guarantees.

## 2 Related work

**Data-driven algorithm design.** Data-driven algorithm design (Ailon et al., 2011; Gupta & Roughgarden, 2016; Balcan, 2020) is a modern approach for automatically configuring algorithms and making them adaptive to specific application domains. In contrast to conventional algorithm design and analysis, which predominantly focuses on worst-case or average-case scenarios, data-driven algorithm design assumes the existence of an (unknown) underlying problem distribution that determines the problem instances that the algorithm encounters. The main objective is to identify the optimal configuration for the algorithm, leveraging available problem instances at hand drawn from the same application domain. Empirical work has consistently validated the effectiveness of data-driven algorithm approaches in various domains, including matrix low-rank approximation (Indyk et al., 2019; 2021), matrix sketching (Li et al., 2023), mixed-integer linear programming (Cheng et al., 2024), among others. These findings underscore the significance of the data-driven algorithm design approach in real-world applications.

**Statistical learning guarantees for data-driven algorithm design.** A growing body of work focuses on theoretically analyzing data-driven algorithm configuration by providing statistical generalization guarantees. This includes learning guarantees for data-driven algorithm design of low-rank approximation and sketching algorithms (Bartlett et al., 2022; Sakaue & Oki, 2023), learning metrics for clustering (Balcan et al., 2020a; Balcan & Sharma, 2021; Balcan et al., 2018c), integer and mixed-integer linear programming

(Balcan et al., 2022b; 2018a; Cheng & Basu, 2024), simulated annealing (Blum et al., 2021), hyperparameter tuning for regularized regression (Balcan et al., 2022a; 2023b), decision tree learning (Balcan & Sharma, 2024), robust nearest-neighbors (Balcan et al., 2023a), deep neural networks (Balcan et al., 2025), pricing problems (Balcan & Beyhaghi, 2024; Xie et al., 2024) among others.

Balcan et al. (2021a) introduced a general framework for establishing learning guarantees for problems that admit a specific piecewise structure. Despite its broad applicability, the framework has certain limitations: (1) it requires an intermediate task of analyzing the dual piece and boundary function classes, which can be non-trivial, and (2) naively applying the framework can lead to sub-optimal bounds (see e.g. Balcan et al., 2020a, Lemma 7, and Bartlett et al., 2022, Appendix E.3). Recently, Bartlett et al. (2022) developed a new approach for establishing generalization guarantees when the computation of the data-driven algorithm's dual utility function can be described by basic arithmetic operators $(+, -, \times, \div)$ and conditional statements, by extending the classical GJ framework (Goldberg & Jerrum, 1993).

**Online learning guarantees for data-driven algorithm design.**   Another line of work focuses on providing no-regret learning guarantees for data-driven algorithm design problems in online learning settings. This includes online learning guarantees for greedy knapsack, SDP-rounding for integer quadratic programming (Balcan et al., 2018b), pricing problems (Balcan & Beyhaghi, 2024), and data-driven linkage-based clustering (Balcan et al., 2020b).

Online learning for data-driven algorithm design is generally a challenging task due to the discontinuity and piecewise structure of the utility functions. Most prior work provides learning guarantees in this setting by verifying the *dispersion* of the utility function sequence, a sufficient condition proposed by Balcan et al. (2018b), roughly stating that the discontinuity of the sequence is not highly concentrated in any small region of the hyperparameter domain. However, verifying the dispersion property is generally challenging. Balcan et al. (2020b) provided a tool for verifying the dispersion property, targeting cases where the discontinuity is described by the roots of random polynomials for one-dimensional hyperparameters or algebraic curves in the two-dimensional case. Balcan & Sharma (2021) generalized this by providing a tool for discontinuities described by algebraic varieties in higher-dimensions. Balcan et al. (2020d) show no-regret online learning under dispersion with respect to more challenging dynamic baselines, and Sharma (2024) establishes how to obtain low internal regret in the data-driven setting. Our dispersion based results in this work imply online learning in these settings as well, using their results. Sharma & Suggala (2025) show how to tune hyperparameters in online bandit learning algorithms using data-driven algorithm design. Balcan et al. (2021b) establish connections between this line of work and online meta-learning.

**Algorithms with predictions.**   Another modern approach for designing learning-based algorithms is *algorithms with predictions* (Mitzenmacher & Vassilvitskii, 2022), in which predictions about certain aspects of the problem (provided either by machine learning models or human experts) are integrated at certain stages of algorithms to enhance their performance. The final performance of these algorithms is closely tied to the quality of predictions. A higher quality of predictions generally correlates with improved algorithmic performance. Hence, the algorithm, now integrated with predictive models, is analyzed based on its inherent algorithmic performance and the quality of predictions. Algorithms with predictions have proved their efficacy in various classical problems, including support estimation (Eden et al., 2021), page migration (Indyk et al., 2022), online matching, flows, and load balancing (Lavastida et al., 2020), among others (Khodak et al., 2022; Lykouris & Vassilvitskii, 2021; Wei & Zhang, 2020).

While having many similarities and overlapping traits, there is a fundamental distinction between the two approaches. Data-driven algorithm design primarily seeks to optimize algorithmic hyperparameters directly for specific application domains. On the other hand, algorithms with predictions aim to incorporate prediction in some stages, with the hope that it will improve the performance of the algorithms if the quality of the prediction is good. However, one still needs to decide for what aspect of the problem should a prediction be used and what the prediction should be on an input problem instance, and these choices heavily affect the performance and properties of the algorithms. It is worth noting that these two directions can complement each other and be integrated into the same system, as explored in recent work (Khodak et al., 2022). Most of the work in the literature has focused on how to incorporate good predictions, but there are no general

results for end-to-end guarantees where the predictions are also being learned in addition to being used by the algorithm. Though there are some positive results for some very specific algorithmic problems e.g. online bipartite matching and the ski rental problem (Khodak et al., 2022).

## 3   Preliminaries

**Parameterized algorithms, utility function class, and dual utility function class.**   In this work, our main focus is on parameterized algorithms, in which each algorithm has a set of hyperparameters $\boldsymbol{a} \in \mathcal{A} \subseteq \mathbb{R}^d$ that have great influence on the performance of the algorithm. Let $\mathcal{X}$ represent the set of input problem instances on which the algorithm operates. For example, in the problem of agglomerative hierarchical clustering (see Section 6.1 for details), a problem instance $\boldsymbol{x} \in \mathcal{X}$ is given by $(S, \boldsymbol{\delta})$, and consists of a set of points $S$ that need to be clustered, and a set of candidate distance functions $\boldsymbol{\delta} = (\delta_1, \ldots, \delta_L)$ used for measuring the distances between pairs of points (the set remains the same for every problem instance)[1]. For example, $\boldsymbol{\delta}$ might contain $\ell_p$ distances for difference values $p \in [0, \infty)$, or some other distance function that is suitable for the geometric structure of the points in $S$ (cf. Balcan et al. 2020a).

The performance of the algorithm for any hyperparameter is measured by the *utility function* $u : \mathcal{X} \times \mathcal{A} \to [0, H]$, where $u(\boldsymbol{x}, \boldsymbol{a})$ represents the performance of the algorithm when operating on input problem instance $\boldsymbol{x}$ and be parameterized by $\boldsymbol{a}$. In the example above, $\boldsymbol{a}$ could be in the $L$-probability simplex, served as the weights to combine distance functions in $\boldsymbol{\delta}$, i.e. $\delta_{\boldsymbol{a}} = \sum_{i=1}^{L} a_i \delta_i$, used in the clustering algorithm. The *utility function class* $\mathcal{U}$ of the algorithm is then defined as $\mathcal{U} = \{u_{\boldsymbol{a}} : \mathcal{X} \to [0, H] \mid \boldsymbol{a} \in \mathcal{A}\}$. The utility function class $\mathcal{U}$ plays a central role in our analysis since the problem of tuning hyperparameter $\boldsymbol{a}$ for the algorithm can be formulated as the problem of analyzing the learnability of $\mathcal{U}$ in both statistical and online learning described below.

However, in data-driven algorithm design, the structure of the utility function class $\mathcal{U}$ can be very intricate in the sense that: (1) a very small variation of the hyperparameter $\boldsymbol{a}$ can lead to sharp, unpredictable changes in the utility function $u_{\boldsymbol{a}}$, and (2) the utility function $u_{\boldsymbol{a}}$ corresponding to a fixed $\boldsymbol{a}$ admits a very complicated structure as a function of $\boldsymbol{x}$. Hence, analyzing $\mathcal{U}$ is often conducted via analyzing the dual utility function class $\mathcal{U}^*$, which often admits a certain degree of structure. The *dual utility function class* $\mathcal{U}^*$ of $\mathcal{U}$ can be defined as $\mathcal{U}^* = \{u_{\boldsymbol{x}}^* : \mathcal{A} \to [0, H] \mid \boldsymbol{x} \in \mathcal{X}\}$, of which each dual utility function $u_{\boldsymbol{x}}^*$ corresponding to a fixed problem instance $\boldsymbol{x}$ is defined as $u_{\boldsymbol{x}}^*(\boldsymbol{a}) := u_{\boldsymbol{a}}(\boldsymbol{x})$, which consists of utility functions obtained by varying the hyperparameter $\boldsymbol{a}$ for fixed problem instances from $\mathcal{X}$. In this work, we will show that if $\mathcal{U}^*$ admits Pfaffian piecewise structure, we can recover the statistical and online learning guarantees for the utility function class $\mathcal{U}$ in several cases of interest.

**Statistical learning.**   In contrast to traditional worst-case or average-case algorithm analysis, we assume the existence of an unknown problem distribution $\mathcal{D}$ over $\mathcal{X}$, which encapsulates information about the relevant application domain. Under such an assumption, our goal is to answer the sample complexity question, i.e. how many problem instances are sufficient to learn near-optimal hyperparameters of the algorithm for any application-specific problem distribution. To this end, it suffices to bound the *pseudo-dimension* (Pollard, 1984) of the corresponding utility function class $\mathcal{U}$.

**Definition 1** (Pseudo-dimension, Pollard, 1984)**.** Consider a real-valued function class $\mathcal{U}$, of which each function takes input in $\mathcal{X}$. Given a set of inputs $S = (\boldsymbol{x}_1, \ldots, \boldsymbol{x}_N)$, we say that $S$ is shattered by $\mathcal{U}$ if there exists a set of real-valued threshold $r_1, \ldots, r_N \in \mathbb{R}$ such that $|\{(\operatorname{sign}(u(\boldsymbol{x}_1) - r_1), \ldots, \operatorname{sign}(u(\boldsymbol{x}_N) - r_N)) \mid u \in \mathcal{U}\}| = 2^N$. The pseudo-dimension of $\mathcal{U}$, denoted as $\operatorname{Pdim}(\mathcal{U})$, is the maximum size $N$ of a input set that $\mathcal{U}$ can shatter.

If the function class $\mathcal{U}$ is binary-valued, this corresponds to the well-known VC-dimension (Vapnik & Chervonenkis, 1974). Standard results in learning theory show that a bound on the pseudo-dimension implies a bound on the sample complexity (see Appendix A for further background), which is formalized as follows.

---

[1]Even though $\boldsymbol{\delta}$ is the same across instances, we follow the convention from prior work (Balcan et al., 2021a) and include it as a part of the problem instance to define the dual utility functions more clearly.

**Theorem 3.1** (Pollard, 1984)**.** *Consider a real-valued function class $\mathcal{U}$, of which each function takes value in $\mathcal{X}$. Assume that $\mathrm{Pdim}(\mathcal{U})$ is finite and $\mathcal{U}$ is bounded by $H$. Then given $\epsilon > 0$ and $\delta \in (0, 1)$, for any $m \geq m(\delta, \epsilon)$, where $m(\delta, \epsilon) = \mathcal{O}\left(\frac{H^2}{\epsilon^2}(\mathrm{Pdim}(\mathcal{F}) + \log(1/\delta))\right)$, with probability at least $1 - \delta$ over the draw of $S = (\boldsymbol{x}_1, \ldots, \boldsymbol{x}_m) \sim \mathcal{D}^m$, we have*

$$\mathbb{E}_{\boldsymbol{x} \sim \mathcal{D}}[\hat{u}_S(\boldsymbol{x})] \geq \sup_{u \in \mathcal{U}} \mathbb{E}_{\boldsymbol{x} \sim \mathcal{D}}[u(\boldsymbol{x})] - \epsilon.$$

*Here $\hat{u}_S \in \arg\max_{u \in \mathcal{U}} \frac{1}{m} \sum_{i=1}^{m} u(\boldsymbol{x}_i)$.*

**Online learning.**  In the online learning setting, there is a sequence of utility functions $u(\boldsymbol{x}_1, \cdot), \ldots, u(\boldsymbol{x}_T, \cdot)$ corresponding to a sequence of problem instances $(\boldsymbol{x}_1, \ldots, \boldsymbol{x}_T)$, coming over $T$ rounds. The task is to design a sequence of hyperparameters $(\boldsymbol{a}_1, \ldots, \boldsymbol{a}_T)$ for the algorithm so that the regret (w.r.t. the best hyperparameter in hindsight) is small

$$\mathrm{Regret}_T = \max_{\boldsymbol{a} \in \mathcal{A}} \sum_{t=1}^{T} u(\boldsymbol{x}_t, \boldsymbol{a}) - \sum_{t=1}^{T} u(\boldsymbol{x}_t, \boldsymbol{a}_t).$$

Our goal is to design a sequence of hyperparameters $\boldsymbol{a}_1, \ldots, \boldsymbol{a}_T$ that achieve sub-linear regret.

## 4   Pfaffian GJ framework for data-driven algorithm design

In a classical work, Goldberg & Jerrum (1993) introduced a comprehensive framework for bounding the VC-dimension (or pseudo-dimension) of parameterized function classes exhibiting a specific property. They proposed that if any function within a given class can be computed via a specific type of computation, named a GJ algorithm, consisting of fundamental operators such as addition, subtraction, multiplication, division, and conditional statements, then the pseudo-dimension of such a function class can be effectively upper bounded. The bound depends on the running time (number of operations) of the algorithm, offering a convenient approach to reduce the task of bounding the complexity of a function class into the more manageable task of counting the number of operations.

However, a bound based on running time can often be overly conservative. Recently, Bartlett et al. (2022) instantiated a refinement for the GJ framework. Noting that any intermediate values the GJ algorithm computes are rational functions of parameters, Bartlett et al. proposed the relevant complexity measures of the GJ framework, namely the *predicate complexity* and the *degree* of the GJ algorithm. Informally, the predicate complexity and the degree are the number of distinct rational functions in conditional statements and the highest order of those rational functions, respectively. Remarkably, based on the GJ framework's complexity measures, Bartlett et al. showed an improved bound, demonstrating its efficacy in various cases, including applications on data-driven algorithm design for numerical linear algebra.

It is worth noting that the GJ algorithm has limitations as it can only accommodate scenarios where intermediate values are rational functions. In other words, it does not capture more prevalent classes of functions, such as the exponential function. Building upon the insights gained from the refined GJ framework, we introduce an extended framework called *the Pfaffian GJ Framework*. Our framework can be used to bound the pseudo-dimension of function classes that can be computed not only by fundamental operators and conditional statements but also by a broad class of functions called Pfaffian functions, which includes exponential and logarithmic functions. Technically, our result is a refinement of the analytical approach introduced by Khovanski (1991); Karpinski & Macintyre (1997); Milnor & Weaver (1997), which is directly applicable to data-driven algorithm design. An important part of our contribution is a careful instantiation of our main result for several important algorithmic problems, as a naive application could result in significantly looser bounds on the sample complexity.

### 4.1   Pfaffian functions

We present the foundational concepts of Pfaffian chains, Pfaffian functions, and their associated complexity measures. Introduced by Khovanski (1991), Pfaffian function analysis is a tool for analyzing the properties

of solution sets of Pfaffian equations. We note that these techniques have been previously used to derive an upper bound on the VC-dimension of sigmoidal neural networks (Karpinski & Macintyre, 1997).

We first introduce the notion of a *Pfaffian chain.* Intuitively, a Pfaffian chain consists of an ordered sequence of functions, in which the derivative of each function can be represented as a polynomial of the variables and previous functions in the sequence.

**Definition 2** (Pfaffian Chain, Khovanski, 1991)**.** A finite sequence of continuously differentiable functions $\eta_1, \ldots, \eta_q : \mathbb{R}^d \to \mathbb{R}$ and variables $\boldsymbol{a} = (a_1, \ldots, a_d) \in \mathbb{R}^d$ form a Pfaffian chain $\mathcal{C}(\boldsymbol{a}, \eta_1, \ldots, \eta_q)$ if there are real polynomials $P_{i,j}(\boldsymbol{a}, \eta_1, \ldots, \eta_j)$ in $a_1, \ldots, a_d, \eta_1, \ldots, \eta_j$, for for all $i \in [d]$ and $j \in [q]$, such that

$$\frac{\partial \eta_j}{\partial a_i} = P_{i,j}(\boldsymbol{a}, \eta_1, \ldots, \eta_q).$$

Here, we emphasize again that $P_{i,j}(\boldsymbol{a}, \eta_1, \ldots, \eta_j)$ is a polynomial in $\boldsymbol{a}$ and the functions $\eta_1(\boldsymbol{a}), \ldots, \eta_j(\boldsymbol{a})$ of $\boldsymbol{a}$. We now define two complexity notations for Pfaffian chains, termed the *length* and *Pfaffian degree*, that dictate the complexity of a Pfaffian chain. The length of a Pfaffian chain is the number of functions that appear on that chain, while the Pfaffian degree of a chain is the maximum degree of polynomials that can be used to express the partial derivative of functions on that chain. Formal definitions of Pfaffian chain length and Pfaffian degree are mentioned in Definition 3.

**Definition 3** (Complexity of Pfaffian chain)**.** Given a Pfaffian chain $\mathcal{C}(\boldsymbol{a}, \eta_1, \ldots, \eta_q)$, as defined in Definition 2, we say that the length of $\mathcal{C}$ is $q$, and Pfaffian degree of $\mathcal{C}$ is $\max_{i,j} \deg(P_{i,j})$.

Given a Pfaffian chain, one can define the Pfaffian function, which is simply a polynomial of variables and functions on that chain.

**Definition 4** (Pfaffian functions, Khovanski, 1991)**.** Given a Pfaffian chain $\mathcal{C}(\boldsymbol{a}, \eta_1, \ldots, \eta_q)$, as defined in Definition 2, a Pfaffian function over the chain $\mathcal{C}$ is a function of the form $g(\boldsymbol{a}) = Q(\boldsymbol{a}, \eta_1, \ldots, \eta_q)$, where $Q$ is a polynomial in variables $\boldsymbol{a}$ and functions $\eta_1, \ldots, \eta_q$ in the chain $\mathcal{C}$. The degree $\Delta$ of the Pfaffian function $g(\boldsymbol{a})$ is the degree of the polynomial $Q(\boldsymbol{a}, \eta_1, \ldots, \eta_q)$.

The concepts of the Pfaffian chain, functions, and complexities may be a bit abstract to unfamiliar readers. To help readers better grasp the concepts of Pfaffian chains and Pfaffian functions, here are some simple examples.

**Example 1.** Consider the chain $\mathcal{C}(a, e^a)$ consisting of the variable $a$ and the function $e^a$, where $a \in \mathbb{R}$. Then $\mathcal{C}$ is a Pfaffian chain since $\frac{d}{da} e^a = e^a$, which is a polynomial of degree 1 in $e^a$. Hence, the chain $\mathcal{C}$ has length $q = 1$ and Pfaffian degree $M = 1$. Now, consider the function $f(a) = (e^a)^2 + a^3$. We observe that $f(a)$ is a polynomial in $a$ and $e^a$ of degree 3. Therefore, $f(a)$ is a Pfaffian function of degree $\Delta = 3$ of the Pfaffian chain $\mathcal{C}$.

**Example 2.** The following example is useful when analyzing the learnability of clustering algorithms (we present only the variables and functions here, their interpretation is deferred to Section 6.1). Let $\boldsymbol{\beta} = (\beta_1, \ldots, \beta_k) \in \mathbb{R}^k_{\geq 0}$ and $\alpha \in \mathbb{R}$ be variables, and let $d(\boldsymbol{\beta}) = \sum_{i=1}^{k} d_i \beta_i$, where $d_i \in \mathbb{R}_+$ are some fixed positive real coefficients for $i = 1, \ldots, k$. Consider the functions $f(\alpha, \boldsymbol{\beta}) := \frac{1}{d(\boldsymbol{\beta})}$, $g(\alpha, \boldsymbol{\beta}) := \ln d(\boldsymbol{\beta})$, and $h(\alpha, \boldsymbol{\beta}) := d(\boldsymbol{\beta})^\alpha$. Then $f$, $g$, and $h$ are Pfaffian functions, all of degree $\Delta = 1$, from the chain $\mathcal{C}(\alpha, \boldsymbol{\beta}, f, g, h)$ of length 3 and Pfaffian degree 2, since

$$\frac{\partial f}{\partial \alpha} = 0, \qquad \frac{\partial f}{\partial \beta_i} = -d_i \cdot f^2,$$

$$\frac{\partial g}{\partial \alpha} = 0, \qquad \frac{\partial g}{\partial \beta_i} = d_i \cdot f,$$

$$\frac{\partial h}{\partial \alpha} = g \cdot h, \quad \frac{\partial h}{\partial \beta_i} = d_i \cdot f \cdot h.$$

### 4.2 Pfaffian GJ Algorithm

We now present a formal definition of the *Pfaffian GJ algorithm*, which shares similarities with the GJ algorithm but extends its capabilities to compute Pfaffian functions as intermediate values, in addition to basic operators and conditional statements. This improvement makes the Pfaffian GJ algorithm significantly more versatile compared to the classical GJ framework.

**Definition 5** (Pfaffian GJ algorithm)**.** A Pfaffian GJ algorithm $\Gamma$ operates on real-valued inputs $\boldsymbol{a} = (a_1, \ldots, a_d) \in \mathcal{A} \subseteq \mathbb{R}^d$, and can perform three types of operations:

- Arithmetic operators of the form $v'' = v \odot v'$, where $\odot \in \{+, -, \times, \div\}$.

- Pfaffian operators of the form $v'' = \eta(v)$, where $\eta : \mathbb{R} \to \mathbb{R}$ is a Pfaffian function.

- Conditional statements of the form "if $v \geq 0 \ldots$ else $\ldots$".

Here $v$ and $v'$ are either inputs $(a_1, \ldots,$ or $a_d)$ or (intermediate) values previously computed by the algorithm.

The main difference between the classical GJ algorithm (Goldberg & Jerrum, 1993; Bartlett et al., 2022) and our Pfaffian GJ algorithm is that we allow unary Pfaffian operators in the algorithmic computation. By leveraging the fundamental properties of Pfaffian chains and functions, we can easily show that all intermediate functions computed by a specific Pfaffian GJ algorithm come from the same Pfaffian chain (see Appendix B.1 for details). Therefore, for a Pfaffian GJ algorithm $\Gamma$, we can define a Pfaffian chain $\mathcal{C}$ corresponding to $\Gamma$, as formalized below.

**Definition 6** (Pfaffian chain associated with Pfaffian GJ algorithm)**.** Given a Pfaffian GJ algorithm $\Gamma$ operating on real-valued inputs $\boldsymbol{a} \in \mathcal{A} \subseteq \mathbb{R}^d$, we say that a Pfaffian chain $\mathcal{C}$ is associated with $\Gamma$ if all the intermediate values computed by $\Gamma$ are Pfaffian functions from the chain $\mathcal{C}$.

This remarkable property enables us to control the complexity of the Pfaffian GJ algorithm $\Gamma$ by controlling the complexity of the corresponding Pfaffian chain $\mathcal{C}$. We formalize this claim in the following lemma.

**Lemma 4.1.** *For any Pfaffian GJ algorithm $\Gamma$ involving a finite number of operations, there is a Pfaffian chain $\mathcal{C}$ of finite length associated with $\Gamma$.*

*Proof Sketch.* The Pfaffian chain $\mathcal{C}$ can be constructed recursively as follows. Initially, we create a Pfaffian chain of variables $\boldsymbol{a}$ with length 0. Using the basic properties of Pfaffian functions discussed in Appendix B.1, each time $\Gamma$ computes a new value $v$, one of the following cases arises: (1) $v$ is a Pfaffian function on the current chain $\mathcal{C}$, or (2) we can extend the chain $\mathcal{C}$ by adding new functions, increasing its length (but still finite), such that $v$ becomes a Pfaffian function on the modified chain $\mathcal{C}$. $\qquad\square$

**Remark 1.** We note that each Pfaffian GJ algorithm $\Gamma$ can be associated with various Pfaffian chains, with different complexities. For example, if $\mathcal{C}(\boldsymbol{a}, \eta_1, \ldots, \eta_q)$ is a Pfaffian chain corresponding to $\Gamma$, then $\mathcal{C}'(\boldsymbol{a}, \eta_1, \ldots, \eta_q, e^{a_1})$ is also a Pfaffian chain corresponding to $\Gamma$, but with a greater length compared to $\mathcal{C}$. Therefore, in any specific application, designing the corresponding Pfaffian chain $\mathcal{C}$ for $\Gamma$ with a small complexity is a crucial task that requires careful analysis.

We now present our main technical tool, which can be used to bound the pseudo-dimension of a function class by expressing the function computation as a Pfaffian GJ algorithm and giving a bound on the complexity of the associated Pfaffian chain. Technical background for the proof is located in Appendix B.2.

**Theorem 4.2.** *Consider a real-valued function class $\mathcal{U} = \{u_{\boldsymbol{a}} : \mathcal{X} \to \mathbb{R} \mid \boldsymbol{a} \in \mathcal{A}\}$ with domain $\mathcal{X}$, of which each function $u_{\boldsymbol{a}} \in \mathcal{U}$ is parameterized by $\boldsymbol{a} \in \mathcal{A} \subseteq \mathbb{R}^d$. Suppose that for every $\boldsymbol{x} \in \mathcal{X}$ and $r \in \mathbb{R}$, there is a Pfaffian GJ algorithm $\Gamma_{\boldsymbol{x},r}$, with associated Pfaffian chain $\mathcal{C}_{\boldsymbol{x},r}$ of length at most $q$ and Pfaffian degree at most $M$, that given $u_{\boldsymbol{a}} \in \mathcal{U}$, outputs $\mathrm{sign}(u_{\boldsymbol{a}}(\boldsymbol{x}) - r)$. Moreover, assume that values computed at intermediate steps of $\Gamma_{\boldsymbol{x},r}$ are from the Pfaffian chain $\mathcal{C}_{\boldsymbol{x},r}$, each of degree at most $\Delta$; and the functions computed in the conditional statements are of at most $K$ Pfaffian functions. Then $\mathrm{Pdim}(\mathcal{U}) \leq d^2 q^2 + 2dq \log(\Delta + M) + 4dq \log d + 2d \log(\Delta K) + 16d$.*

*Proof.* To bound $\mathrm{Pdim}(\mathcal{U})$, the overall idea is that given any $N$ input problem instances $\boldsymbol{x}_1, \ldots, \boldsymbol{x}_N$ and $N$ thresholds $r_1, \ldots, r_N$, we bound $\Pi_{\mathcal{U}}(N)$, the number of distinct sign patterns

$$(\mathrm{sign}(u_{\boldsymbol{a}}(\boldsymbol{x}_1) - r_1), \ldots, \mathrm{sign}(u_{\boldsymbol{a}}(\boldsymbol{x}_N) - r_N)) = (\mathrm{sign}(u^*_{\boldsymbol{x}_N}(\boldsymbol{a}) - r_1), \ldots, \mathrm{sign}(u^*_{\boldsymbol{x}_N}(\boldsymbol{a}) - r_N))$$

obtained by varying $\boldsymbol{a} \in \mathcal{A}$. Then we solve the inequality $2^N \leq \Pi_{\mathcal{L}}(N)$ to obtain an upper bound for $\mathrm{Pdim}(\mathcal{U})$.

From the assumption, for each $\boldsymbol{x_i}$ and a threshold value $r_i$, the value of $\mathrm{sign}(u_{\boldsymbol{a}}(\boldsymbol{x}_i) - r_i) = \mathrm{sign}(u^*_{\boldsymbol{x}_i}(\boldsymbol{a}) - r_i)$ can be computed by a Pfaffian GJ algorithm $\Gamma_{\boldsymbol{x}_i, r_i}$ that takes input as $\boldsymbol{a}$ and return if $u_{\boldsymbol{a}}(\boldsymbol{x}_i) > r_i$.

Again, from the assumption, the Pfaffian GJ algorithm $\Gamma_{\boldsymbol{x}_i, r_i}$ has at most $K$ conditional statements, each determines if $\tau_{\boldsymbol{x}_i, j}(\boldsymbol{a}) \geq 0$ $(j = 1, \ldots, K)$, where $\tau_{\boldsymbol{x}_i, r_i}(\boldsymbol{a})$ is a Pfaffian function from the Pfaffian chain $\mathcal{C}_{\boldsymbol{x}_i, r_i}$ of length at most $q$ and Pfaffian degree at most $M$ corresponding to $\Gamma_{\boldsymbol{x}, i}$.

Therefore, each binary value $\mathrm{sign}(u^*_{\boldsymbol{x}_i}(\boldsymbol{a}) - r_i)$ is determined by at most $K$ other values in the form $\mathrm{sign}(\tau_{\boldsymbol{x}_i, j}(\boldsymbol{a}))$ for $j = 1, \ldots, K$, meaning that the number of patterns

$$(\mathrm{sign}(u^*_{\boldsymbol{x}_1}(\boldsymbol{a}) - r_1), \ldots, \mathrm{sign}(u^*_{\boldsymbol{x}_N}(\boldsymbol{a}) - r_N))$$

is upper bounded by the number of sign patterns

$$(\mathrm{sign}(\tau_{\boldsymbol{x}_1, 1}(\boldsymbol{a})), \ldots, \mathrm{sign}(\tau_{\boldsymbol{x}_1, K}(\boldsymbol{a})), \ldots, \mathrm{sign}(\tau_{\boldsymbol{x}_N, 1}(\boldsymbol{a})), \ldots, \mathrm{sign}(\tau_{\boldsymbol{x}_N, K}(\boldsymbol{a})))$$

obtained by varying $\boldsymbol{a}$. Moreover, we can construct a Pfaffian chain $\mathcal{C}'$ of length at most $qN$ and Pfaffian degree at most $\Delta$ by merging all the Pfaffian chain $\mathcal{C}_{\boldsymbol{x}_i}$, such that any function $\tau_{\boldsymbol{x}_i, j}(\boldsymbol{a})$ above is a Pfaffian function from the Pfaffian chain $\mathcal{C}'$, and of degree at most $\Delta$.

Finally, the number of sign patterns $(\mathrm{sign}(\tau_{\boldsymbol{x}_1, 1}(\boldsymbol{a})), \ldots, \mathrm{sign}(\tau_{\boldsymbol{x}_N, K}(\boldsymbol{a})))$ can be upper bounded by the number of connected components of $\mathbb{R}^d - \cup_{i=1}^{N} \cup_{j=1}^{K} \{\boldsymbol{a} : \tau_{\boldsymbol{x}_i, j}(\boldsymbol{a}) = 0\}$, where $\tau_{\boldsymbol{x}_i, j}(\boldsymbol{a}) = 0$ is a Pfaffian hypersurface. Using results by Khovanski (1991) and Karpinski & Macintyre (1997) (see Appendix B.2 for background), we can establish Lemma B.6, which leads to the claimed bound $\mathrm{Pdim}(\mathcal{U}) \leq d^2 q^2 + 2dq \log(\Delta + M) + 4dq \log d + 2d \log \Delta K + 16d$.

$\square$

**Remark 2.** For the case $q = 0$, meaning that the functions computed in the conditional statements are merely rational functions in $\boldsymbol{a}$, Theorem 4.2 gives an upper bound of $\mathcal{O}(d \log(\Delta K))$, which matches the rate of GJ algorithm instantiated by Bartlett et al. (2022).

## 5 Pfaffian piecewise structure for data-driven statistical learning

In this section, we analyze function classes for which the duals have a Pfaffian piecewise structure, a special case of the piecewise decomposable structure introduced by Balcan et al. (2021a). Compared to their piecewise decomposable structure, our proposed Pfaffian piecewise structure incorporates additional information about the Pfaffian structures of piece and boundary functions of the dual function classes, as well as the maximum number of forms that the piece functions can take. We argue that the additional information can be derived as a by-product in many data-driven algorithm design problems, but would be ignored if naively using the framework by Balcan et al. (2021a). We then show that if the dual utility function class $\mathcal{U}^*$ of a parameterized algorithm admits the Pfaffian piecewise structure, then we can directly establish learning guarantees for the utility function class $\mathcal{U}$. We note that the advantage of our framework compared to the framework by Balcan et al. (2021a) is two-fold: (1) our approach offers a concrete and alternative way of analyzing the pseudo-dimension for the utility function class $\mathcal{U}$, avoiding the non-trivial combinatorial tasks of analyzing the pseudo/VC-dimension of the dual piece and boundary function classes if using the framework by Balcan et al. (2021a), and (2) directly and naively applying the framework by Balcan et al. (2021a) can potentially lead to loose or vacuous bounds (see e.g. Balcan et al., 2020a, Lemma 7 and Bartlett et al., 2022, Appendix E.3). Additionally, we propose a further refined argument for the case where all dual utility functions share the same boundary structures, which leads to further improved learning guarantees.

### 5.1 Prior generalization framework for piecewise structured utility functions in data-driven algorithm design and its limitations

In this section, we discuss a prior general framework for providing learning guarantees for data-driven algorithm design problems by Balcan et al. (2021a). Many parameterized algorithms, such as combinatorial algorithms and integer/mixed-integer programming (Balcan et al., 2018a; 2017), exhibit volatile utility functions with respect to their parameters. In other words, even minor changes in parameters can lead to significant alterations in the behavior of the utility function. Analyzing such volatile utility function classes poses a significant technical challenge.

Fortunately, many data-driven algorithm design problems still possess a certain degree of structure. Prior studies (Balcan et al., 2017; 2018a; 2022a; 2023b) have demonstrated that the dual utility functions associated with data-driven algorithm design problems often exhibit a piecewise structure. In other words, the parameter space of the dual utility function can be partitioned into regions, each separated by distinct *boundary functions*. Within each region, the dual utility function corresponds to a *piece function* that exhibits well-behaved properties, such as being linear, rational, or Pfaffian in nature. Building upon this insight, prior work by Balcan et al. (2021a) proposed a formal definition of a piecewise-structured dual utility function class and established a generalization guarantee applicable to any data-driven algorithm design problem conforming to such structures.

Formally, let us recall the definition of the utility function class $\mathcal{U} = \{u_{\boldsymbol{a}} : \mathcal{X} \to [0, H] \mid \boldsymbol{a} \in \mathcal{A}\}$ for a parameterized algorithm, where $\mathcal{A} \subseteq \mathbb{R}^d$. This class represents functions that evaluate the performance of the algorithm, with $u_{\boldsymbol{a}} : \mathcal{X} \to [0, H]$ denoting the utility function corresponding to parameter $\boldsymbol{a}$. For a given input problem instance $\boldsymbol{x} \in \mathcal{X}$, $u_{\boldsymbol{a}}(\boldsymbol{x})$ yields the performance evaluation of the algorithm on $\boldsymbol{x}$. Notably, for each input $\boldsymbol{x} \in \mathcal{X}$, we can define the *dual utility function corresponding to $\boldsymbol{x}$* as $u_{\boldsymbol{x}}^* : \mathcal{A} \to [0, H]$, where $u_{\boldsymbol{x}}^*(\boldsymbol{a}) := u_{\boldsymbol{a}}(\boldsymbol{x})$ measures the performance for a specific problem instance $\boldsymbol{x}$ as a function of the parameter $\boldsymbol{a}$. Consequently, we can also define the dual utility function class $\mathcal{U}^* = \{u_{\boldsymbol{x}}^* : \mathcal{A} \to \mathbb{R} \mid \boldsymbol{x} \in \mathcal{X}\}$. It was shown in prior work (Balcan et al., 2021a; 2022a;c; 2023b) that in many data-driven algorithm design problems, every function in the dual function class $\mathcal{U}^*$ adheres to a specific piecewise structure, which can be precisely defined as follows:

**Definition 7** (Piecewise decomposable, Balcan et al., 2021a)**.** A function class $\mathcal{H} \subseteq \mathbb{R}^{\mathcal{A}}$ that maps a domain $\mathcal{A}$ to $\mathbb{R}$ is $(\mathcal{F}, \mathcal{G}, k)$-piecewise decomposable for a class $\mathcal{G} \subseteq \{0, 1\}^{\mathcal{A}}$ and a class $\mathcal{F} \subseteq \mathbb{R}^{\mathcal{A}}$ of piece functions if the following holds: for every $h \in \mathcal{H}$, there are $k$ boundary functions[2] $\mathbb{I}(g^{(1)}(\boldsymbol{a}) \geq 0), \ldots, \mathbb{I}(g^{(k)}(\boldsymbol{a}) \geq 0) \in \mathcal{G}$ and a piece function $f_{\mathbf{b}} \in \mathcal{F}$ for each bit vector $\boldsymbol{b} \in \{0, 1\}^k$ such that for all $\boldsymbol{a} \in \mathcal{A}$, $h(\boldsymbol{a}) = f_{\mathbf{b}_y}(\boldsymbol{a})$ where $\mathbf{b}_{\boldsymbol{a}} = (g^{(1)}(\boldsymbol{a}), \ldots, g^{(k)}(\boldsymbol{a})) \in \{0, 1\}^k$.

An intuitive illustration of the piecewise structure can be found in Figure 3. Roughly speaking, if a function class satisfies the piecewise structure as defined in Definition 7, then for each function in such a class, the input domain is partitioned into multiple regions by $k$ boundary functions. Within each region, which corresponds to a $k$-element binary vector indicating its position relative to the $k$ boundary functions, the utility function is a well-behaved piece function. Based on this observation, Balcan et al. (2021a) showed that for any algorithm, if the dual utility function class satisfies the piecewise structure as defined in Definition 7, then the pseudo-dimension of the utility function class is bounded.

**Theorem 5.1** (Balcan et al., 2021a)**.** *Consider the utility function class $\mathcal{U} = \{u_{\boldsymbol{a}} : \mathcal{X} \to [0, H] \mid \boldsymbol{a} \in \mathcal{A}\}$. Suppose that the dual function class $\mathcal{U}^*$ is $(\mathcal{F}, \mathcal{G}, k)$-piecewise decomposable with boundary functions $\mathcal{G} \subseteq \{0, 1\}^{\mathcal{A}}$ and piece functions $\mathcal{F} \subseteq \mathbb{R}^{\mathcal{A}}$. Then the pseudo-dimension of $\mathcal{U}$ is bounded as follows*

$$\text{Pdim}(\mathcal{U}) = \mathcal{O}((\text{Pdim}(\mathcal{F}^*) + \text{VCdim}(\mathcal{G}^*)) \log(\text{Pdim}(\mathcal{F}^*) + \text{VCdim}(\mathcal{G}^*)) + \text{VCdim}(\mathcal{G}^*) \log k),$$

*where $\mathcal{F}^*$ and $\mathcal{G}^*$ is the dual class of $\mathcal{F}$ and $\mathcal{G}$, respectively.*

Intuitively, Theorem 5.1 allows us to bound the pseudo-dimension of the utility function class $\mathcal{U}$, which is not well-behaved, by alternatively analyzing the dual boundary and piece function classes $\mathcal{F}^*$ and $\mathcal{G}^*$. However,

---

[2]At the risk of notation abuse, we sometimes use the term *boundary functions* to refer to $g^{(i)}(\boldsymbol{a})$ rather than $\mathbb{I}(g^{(i)}(\boldsymbol{a}) \geq 0)$. We will ensure the context is clear when employing this shorthand.

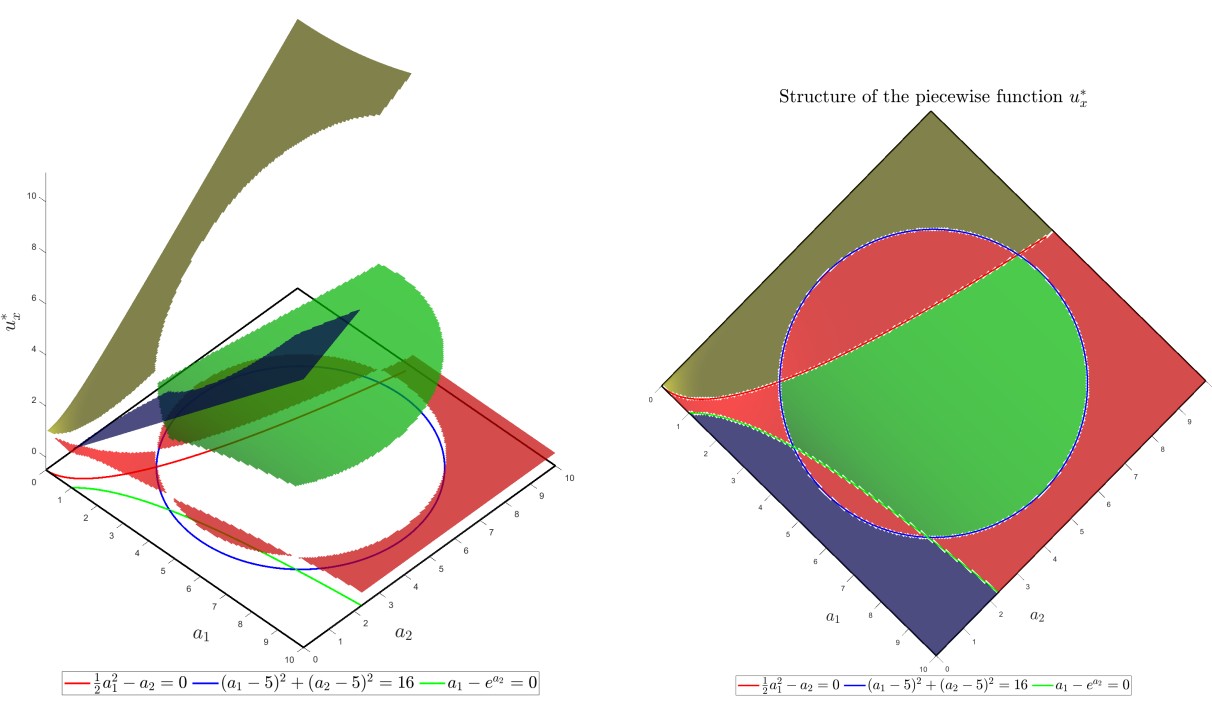

(a) A demonstration of piecewise structure of $u_{\boldsymbol{x}}^*$ in sheer view. This has been shown in Figure 2.

(b) The corresponding piecewise structure of $u_{\boldsymbol{x}}^*$ in top view.

Figure 3: An example of the original piecewise structure (Definition 7) and our proposed Pfaffian piecewise structure (Definition 8). Here, (a) demonstrates the sheer view of the piecewise structure of a specific dual utility function $u_{\boldsymbol{x}}^*$, while (b) shows the corresponding top view for better illustration of regions and their boundaries. As can be seen, there are three boundary functions $g_{\boldsymbol{x},1}(\boldsymbol{a}) = \frac{1}{2}a_1^2 - a_2$, $g_{\boldsymbol{x},2}(\boldsymbol{a}) = (a_1 - 5)^2 + (a_2 - 5)^2 - 16$, and $g_{\boldsymbol{x},3}(\boldsymbol{a}) = a_1 - e^{a_2}$, partitioning the domain $\mathcal{A}$ into 7 regions. In each region, the function $u_{\boldsymbol{x}}^*(\boldsymbol{a})$ takes the form of a Pfaffian function. What is not captured by the original piecewise structure Definition 7 is that, in this example, there are only 4 forms that $u_{\boldsymbol{x}}^*(\boldsymbol{a})$ can take, which is either $a_1 + \frac{a_2}{2}$ (blue region), $e^{-0.2(a_1^2+a_2^2)}$ (red regions), $\log(a_2) + 2$ (green region), and $\sqrt{a_1^2 + a_2^2 + \exp(0.1\sqrt{a_1})}$, (yellow region). It can be verified that all the piece and boundary functions are Pfaffian function from the Pfaffian chain $\mathcal{C}_{\boldsymbol{x}}(\boldsymbol{a}, e^{a_2}, e^{-0.2(a_1^2+a_2^2)}, \frac{1}{\sqrt{a_1}}, e^{0.1\sqrt{a_1}}, \frac{1}{\sqrt{a_1^2+a_2^2+\exp(0.1\sqrt{a_1})}})$.

Bartlett et al. (2022) demonstrate that for certain problems in which the piecewise structure involves only rational functions, naively applying Theorem 5.1 can yield looser bounds compared to the approach based on the GJ framework (Goldberg & Jerrum, 1993). Besides, applying Theorem 5.1 requires the non-trivial task of analyzing the dual classes $\mathcal{F}^*$ and $\mathcal{G}^*$ of piece and boundary functions. This might cause trouble, such as leading to loose bounds (see e.g. Balcan et al., 2020a, Lemma 7) or vacuous bounds (e.g. Bartlett et al., 2022, Appendix E.3). This situation arises when the piece and boundary functions involve Pfaffian functions, motivating the need to design a refined approach.

## 5.2 A refined piecewise structure for data-driven algorithm design

In this section, we propose a more refined and concrete approach to derive learning guarantees for data-driven algorithm design problems where the utility functions exhibit a Pfaffian piecewise structure. The key difference between our proposed frameworks and the framework by Balcan et al. (2021a) is that we consider the following additional factors: (1) Both piece and boundary functions are Pfaffian functions from the same Pfaffian chain, and (2) the maximum number of the distinct forms that the piece function $h_{\boldsymbol{x},i}(\boldsymbol{a})$ can admit.

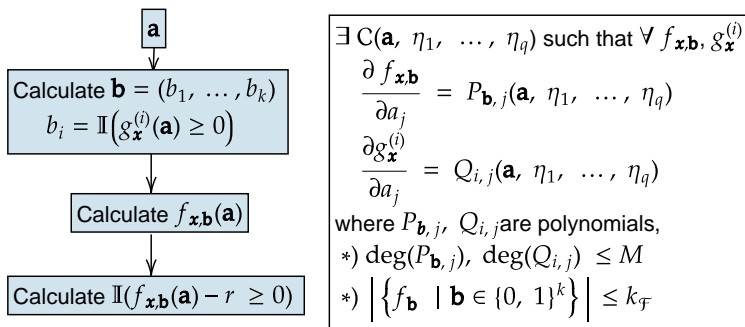

Figure 4: A demonstration of how the computation of a dual utility function satisfying Pfaffian piecewise structure can be described by the Pfaffian GJ algorithm. Given an input $\boldsymbol{x} \in \mathcal{X}$ and a threshold $r \in \mathbb{R}$, the function $u_{\boldsymbol{x}}^*$ is piecewise structured with boundary functions $g_{\boldsymbol{x}}^{(i)}$ (for $i = 1, \ldots k$), and piece functions $f_{h,\mathbf{b}}$ ($\mathbf{b} \in \{0,1\}^k$). Note that, the piece functions $f_{h,\mathbf{b}}$ can take at most $k_{\mathcal{F}}$ forms and all the piece and boundary functions are Pfaffian functions from the chain $\mathcal{C}$.

Later, we will argue that by leveraging those extra structures, we can get a better pseudo-dimension upper bound by a logarithmic factor, compared to using the framework by Balcan et al. (2021a). The Pfaffian piecewise structure is formalized as below.

**Definition 8** (Pfaffian piecewise structure). A function class $\mathcal{H} \subseteq \mathbb{R}^{\mathcal{A}}$ that maps domain $\mathcal{A} \subseteq \mathbb{R}^d$ to $\mathbb{R}$ is said to be $(k_{\mathcal{F}}, k_{\mathcal{G}}, q, M, \Delta, d)$ piecewise structured if the following holds: for every $h \in \mathcal{H}$, there are at most $k_{\mathcal{G}}$ boundary functions of the forms $\mathbb{I}(g_h^{(1)}(\boldsymbol{a}) \geq 0), \ldots, \mathbb{I}(g_h^{(k)}(\boldsymbol{a}) \geq 0)$, where $k \leq k_{\mathcal{G}}$, and a piece function $f_{h,\mathbf{b}}$ for each binary vector $\mathbf{b} \in \{0,1\}^k$ such that for all $\boldsymbol{a} \in \mathcal{A}$, $h(\boldsymbol{a}) = f_{h,\mathbf{b}_{\boldsymbol{a}}}(\boldsymbol{a})$ where $\mathbf{b}_{\boldsymbol{a}} = (\mathbb{I}(g_h^{(1)}(\boldsymbol{a}) \geq 0), \ldots, \mathbb{I}(g_h^{(k)}(\boldsymbol{a}) \geq 0)) \in \{0,1\}^k$. Moreover, the piece functions $f_{h,\mathbf{b}}$ can take on of at most $k_{\mathcal{F}}$ forms, i.e. $\left| \{ f_{h,\mathbf{b}} \mid \mathbf{b} \in \{0,1\}^k \} \right| \leq k_{\mathcal{F}}$, and all the piece and boundary functions $f_{h,\mathbf{b}}, g_h^{(i)}$ are Pfaffian functions of degree at most $\Delta$ over a Pfaffian chain $\mathcal{C}_h$ of length at most $q$ and Pfaffian degree at most $M$.

An intuitive illustration of Pfaffian piecewise structure and its comparison with the piecewise structure by Balcan et al. (2021a) can be found in Figure 3. For a data-driven algorithm design problem with corresponding utility function class $\mathcal{U} = \{ u_{\boldsymbol{a}} : \mathcal{X} \to [0, H] \mid \boldsymbol{a} \in \mathcal{A} \}$ where $\mathcal{A} \subseteq \mathbb{R}^d$, we can see that if its dual utility function class $\mathcal{U}^* = \{ u_{\boldsymbol{x}}^* : \mathcal{A} \to \mathbb{R} \mid \boldsymbol{x} \in \mathcal{X} \}$ admits the Pfaffian piecewise structure as in Definition 8, then it can be computed using the Pfaffian GJ algorithm (see Figure 4 for a visualization). Therefore, we can use the established results for the Pfaffian GJ algorithm (Theorem 4.2) to derive learning guarantees for such problems. We formalize this claim in the following theorem.

**Theorem 5.2.** *Consider the utility function class* $\mathcal{U} = \{ u_{\boldsymbol{a}} : \mathcal{X} \to [0, H] \mid \boldsymbol{a} \in \mathcal{A} \}$ *where* $\mathcal{A} \subseteq \mathbb{R}^d$. *Suppose that the dual function class* $\mathcal{U}^* = \{ u_{\boldsymbol{x}}^* : \mathcal{A} \to [0, H] \mid \boldsymbol{x} \in \mathcal{X} \}$ *is* $(k_{\mathcal{F}}, k_{\mathcal{G}}, q, M, \Delta, d)$*-Pfaffian piecewise structured, then the pseudo-dimension of* $\mathcal{U}$ *is bounded as follows*

$$\mathrm{Pdim}(\mathcal{U}) \leq d^2 q^2 + 2dq \log(\Delta + M) + 4dq \log d + 2d \log \Delta (k_{\mathcal{F}} + k_{\mathcal{G}}) + 16d.$$

*Proof.* An intuitive explanation of this theorem can be found in Figure 4. Since $\mathcal{U}^*$ admits $(k_{\mathcal{F}}, k_{\mathcal{G}}, q, M, \Delta, d)$-Pfaffian piecewise structure, then for any problem instance $\boldsymbol{x} \in \mathcal{X}$ corresponding to the dual utility function $u_{\boldsymbol{x}}^*$, there are $k$ boundary functions $\mathbb{I}(g_{\boldsymbol{x}}^{(1)} \geq 0), \ldots, \mathbb{I}(g_{\boldsymbol{x}}^{(k)} \geq 0)$ where $k \leq k_{\mathcal{G}}$, and a piece function $f_{\boldsymbol{x},\boldsymbol{b}}$ for each binary vector $\boldsymbol{b} \in \{0,1\}^k$ such that for all $\boldsymbol{a} \in \mathcal{A}$, $u_{\boldsymbol{x}}^*(\boldsymbol{a}) = f_{\boldsymbol{x},\boldsymbol{b}_{\boldsymbol{a}}}(\boldsymbol{a})$, where $\boldsymbol{b}_{\boldsymbol{a}} = (\mathbb{I}(g_{\boldsymbol{x}}^{(1)}(\boldsymbol{a}) \geq 0), \ldots, \mathbb{I}(g_{\boldsymbol{x}}^{(k)}(\boldsymbol{a}) \geq 0))$. Therefore, for any problem instance $\boldsymbol{x}$ and real threshold $r \in \mathbb{R}$,

the boolean value of $u_{\boldsymbol{x}}^*(\boldsymbol{a}) - r \geq 0$ for any $\boldsymbol{a} \in \mathcal{A}$ can be calculated by an algorithm $\Gamma_{\boldsymbol{x},r}$ described as follow: first calculating the boolean vector $\boldsymbol{b_a}$, and then calculate the boolean value $f_{\boldsymbol{x},\boldsymbol{b_a}}(\boldsymbol{a}) - r \geq 0$.

From assumption, note that $g_{\boldsymbol{x}}^{(i)}$ and $f_{\boldsymbol{x},\boldsymbol{b}}$ are Pfaffian functions of degree at most $\Delta$ from Pfaffian chain $\mathcal{C}_{\boldsymbol{x}}$ of length at most $q$ and Pfaffian degree at most $M$. This means that $\Gamma_{\boldsymbol{x},r}$ is a Pfaffian GJ algorithm. Therefore, combining with Theorem 4.2, we have the above claim. $\qquad\square$

**Remark 3.** The details of the differences between our refined Pfaffian piecewise structure and the piecewise structure by Balcan et al. (2021a) can be found in Appendix C.1. In short, compared to the framework by Balcan et al. (2021a), our framework offers: (1) an improved upper bound on the pseudo-dimension by a logarithmic factor (Theorem C.1) compared to naively applying (Balcan et al., 2021a), and (2) a more concrete method for problems admitting a Pfaffian piecewise structure, without invoking dual piece and boundary function classes, which are non-trivial to analyze. This might lead to loose bounds (see e.g. Balcan et al. (2020a), Lemma 7, and Bartlett et al. (2022), Appendix E.3).

### 5.3 Special case: discontinuity-homogeneous function class

We now consider the case where all the dual utility functions $v^* \in \mathcal{V}^*$ share the same discontinuity structure, which can be used to establish an improved bound for the utility function class $\mathcal{V}$. We then argue that this analysis is particularly useful in some cases (e.g. Section 5.3.1).

Concretely, consider a utility function class $\mathcal{V} = \{v_{\boldsymbol{a}} : \mathcal{X} \to [0, H] \mid \boldsymbol{a} \in \mathcal{A}\}$ where $\mathcal{A} \subseteq \mathbb{R}^d$, with the corresponding dual function class $\mathcal{V} = \{v_{\boldsymbol{x}}^* : \mathcal{A} \to [0, H] \mid \boldsymbol{x} \in \mathcal{X}\}$. Different from Section 5.2, here all the dual utility function $v_{\boldsymbol{x}}^*$ shares the same discontinuity structure: that is, there is a partition $\mathcal{P} = \{\mathcal{A}_1, \ldots, \mathcal{A}_n\}$ of the parameter space $\mathcal{A}$ such that for any problem instance $\boldsymbol{x} \in \mathcal{X}$, the dual function $v_{\boldsymbol{x}}^*$ restricted to any region $\mathcal{A}_i$ is a Pfaffian function. In this case, we have the following refined bound for the utility function class $\mathcal{V}$.

**Corollary 5.3.** *Consider a function class $\mathcal{V} = \{v_{\boldsymbol{a}} : \mathcal{X} \to [0, H] \mid \boldsymbol{a} \in \mathcal{A}\}$ where $\mathcal{A} \subseteq \mathbb{R}^d$. Assume there is a partition $\mathcal{P} = \{\mathcal{A}_1, \ldots, \mathcal{A}_n\}$ of the parameter space $\mathcal{A}$ such that for any problem instance $\boldsymbol{x} \in \mathcal{X}$, the dual function $v_{\boldsymbol{x}}^*$ is a Pfaffian function of degree at most $\Delta$ in region $\mathcal{A}_i$ from a Pfaffian chain $\mathcal{C}_{\boldsymbol{x},\mathcal{A}_i}$ of length at most $q$ and Pfaffian degree $M$. Then the pseudo-dimension of $\mathcal{V}$ is upper bounded as follows*

$$\mathrm{Pdim}(\mathcal{V}) = \mathcal{O}(q^2 d^2 + qd \log(\Delta + M) + qd \log d + \log n).$$

**Remark 4.** The detailed proof is provided in Appendix C.2. Essentially, Lemma 5.3 simplifies the analysis by restricting the complexity of the Pfaffian chain to individual regions $\mathcal{A}_i$, rather than across the entire partition $\mathcal{P}$. The insight here is that, compared to the Pfaffian piecewise structure (Theorem 5.1), Lemma 5.3 makes use of the fact that all the dual functions $v_{\boldsymbol{x}}^*$ share the same discontinuity structure dictated by a fixed partition $\mathcal{P}$ of $\mathcal{A}$. This shared structure significantly reduces the length of the Pfaffian chain, which is typically the dominant term in the upper bound.

#### 5.3.1 Use case: analyzing via surrogate function class

In many applications, we might want to analyze the utility function class $\mathcal{U}$ indirectly by studying a surrogate utility function class $\mathcal{V}$, of which the dual function class $\mathcal{V}^*$ is "sufficiently close" to $\mathcal{U}^*$. This indirect approach offers several advantages. First, even the dual utility function $\mathcal{U}^*$ may lack a clear structure or prove difficult to analyze, making it impractical to establish learning guarantees for $\mathcal{U}$ by examining $\mathcal{U}^*$ (Balcan et al., 2023b). Second, when $\mathcal{U}^*$ is excessively complex, deriving learning guarantees for $\mathcal{U}$ through its analysis may lead to sub-optimal bounds. In such cases, analyzing a simpler surrogate function class $\mathcal{V}$ can yield tighter bounds (Balcan et al., 2020c).

Formally, consider the utility function class $\mathcal{U} = \{u_{\boldsymbol{a}} : \mathcal{X} \to [0, H] \mid \boldsymbol{a} \in \mathcal{A}\}$ with the corresponding dual utility function class $\mathcal{U}^* = \{u_{\boldsymbol{x}}^* : \mathcal{A} \to [0, H] \mid \boldsymbol{x} \in \mathcal{X}\}$ over instance space $\mathcal{X}$ and parameter space $\mathcal{A} \subseteq \mathbb{R}^d$. Assume that for any dual utility function $u_{\boldsymbol{x}}^* \in \mathcal{U}^*$, there is a function $v_{\boldsymbol{x}}^* : \mathcal{A} \to [0, H]$ such that $\|v_{\boldsymbol{x}}^* - u_{\boldsymbol{x}}^*\|_\infty \leq \delta$. We then construct the surrogate dual function class $\mathcal{V}^* = \{v_{\boldsymbol{x}}^* : \mathcal{A} \to [0, H] \mid \boldsymbol{x} \in \mathcal{X}\}$, and the surrogate utility function class $\mathcal{V} = \{v_{\boldsymbol{a}} : \mathcal{X} \to [0, H] \mid \boldsymbol{a} \in \mathcal{A}\}$.

In this section, we consider a scenario where we approximate the parameterized utility function over a predefined partition $\mathcal{A}_1, \ldots, \mathcal{A}_n$ of the parameter space $\mathcal{A}$. Formally, a partition $\mathcal{P}$ of a set $\mathcal{A}$ is a collection $\{\mathcal{A}_1, \ldots, \mathcal{A}_n\}$ of non-empty subsets $\mathcal{A}_i$ of $\mathcal{A}$ that are pairwise disjoint and whose union is the entire set $\mathcal{A}$. For any problem instance $\boldsymbol{x}$, the function $v_{\boldsymbol{x}}^*(\boldsymbol{a})$ restricted to $\mathcal{A}_i$ (for any $i = 1, \ldots, n$) is a Pfaffian function from some Pfaffian chain $\mathcal{C}_{\boldsymbol{x}}$. This is a special case, as for every problem instance $\boldsymbol{x}$, the dual function $v_{\boldsymbol{x}}^*$ exhibits the same discontinuity structure, which can be leveraged to obtain a tighter bound. Our goal is for this property to recover the learning guarantee for the utility function class $\mathcal{U}$. To do that, we proceed with the following steps: (1) derive learning guarantee for $\mathcal{V}$ using the property of $\mathcal{V}^*$, (2) using the relation between $\mathcal{U}^*$ and $\mathcal{V}^*$, derive learning guarantee for $\mathcal{U}$ via $\mathcal{V}$. Step (1) is resolved by Lemma 5.3 above, and to proceed step (2), we first recall the following useful results.

**Lemma 5.4** (Balcan et al., 2020c). *Let* $\mathcal{F} = \{f_{\boldsymbol{a}} : \mathcal{X} \to [0, H] \mid \boldsymbol{a} \in \mathcal{A}\}$ *and* $\mathcal{G} = \{g_{\boldsymbol{a}} : \mathcal{X} \to [0, H] \mid \boldsymbol{a} \in \mathcal{A}\}$ *be two function classes parameterized by* $\boldsymbol{a} \in \mathcal{A}$ *and consist of functions mapping from* $\mathcal{X}$ *to* $[0, H]$. *For any set of problem instances* $S \in \mathcal{X}^N$, *we have*

$$\hat{\mathscr{R}}_S(\mathcal{F}) \leq \hat{\mathscr{R}}_S(\mathcal{G}) + \frac{1}{|S|} \sum_{\boldsymbol{x} \in S} \|f_{\boldsymbol{x}}^* - g_{\boldsymbol{x}}^*\|_\infty.$$

*Here* $\hat{\mathscr{R}}_S(\mathcal{F})$ *is the empirical Rademacher complexity of* $\mathcal{F}$ *corresponding to the set of inputs* $S = \{\boldsymbol{x}_1, \ldots, \boldsymbol{x}_N\}$, *i.e.*

$$\hat{\mathscr{R}}_S(\mathcal{F}) = \mathbb{E}_{\xi_1, \ldots, \xi_N \ i.i.d \ from \ unif. \ \{-1, 1\}} \left[ \frac{1}{N} \sup_{f \in \mathcal{F}} \sum_{i=1}^{N} \xi_i f(\boldsymbol{x}_i) \right].$$

We also recall a standard result in learning theory, which draws a connection between empirical Rademacher complexity and the pseudo-dimension.

**Lemma 5.5** (Wainwright, 2019). *Let* $\mathcal{F}$ *be a function class consisting of functions with bounded range* $[0, H]$. *Then* $\mathscr{R}_N(\mathcal{F}) = \mathcal{O}\left(H\sqrt{\frac{\text{Pdim}(\mathcal{F})}{N}}\right)$. *Here* $\mathscr{R}_N(\mathcal{F}) = \sup_{S \in \mathcal{X}^N} \hat{\mathscr{R}}_S(\mathcal{F})$.

We are now ready to present the main result in this section, which allows us to establish learning guarantees for the utility function class $\mathcal{U}$ via the surrogate function class $\mathcal{V}$ satisfying a very specific piecewise Pfaffian structure.

**Lemma 5.6.** *Consider the utility function class* $\mathcal{U} = \{u_{\boldsymbol{a}} : \mathcal{X} \to [0, H] \mid \boldsymbol{a} \in \mathcal{A}\}$ *where* $\mathcal{A} \subseteq \mathbb{R}^d$. *Assume that there exists a function class* $\mathcal{V} = \{v_{\boldsymbol{a}} : \mathcal{X} \to \mathbb{R} \mid \boldsymbol{a} \in \mathcal{A}\}$ *such that:*

1. *For any* $\boldsymbol{x}$, *we have* $\|u_{\boldsymbol{x}}^* - v_{\boldsymbol{x}}^*\|_\infty \leq \xi$, *and*

2. *There is a partition* $\mathcal{P} = \{\mathcal{A}_1, \ldots, \mathcal{A}_n\}$ *of* $\mathcal{A}$ *such that for any problem instance* $\boldsymbol{x} \in \mathcal{X}$, *the dual function* $v_{\boldsymbol{x}}^*$ *is a Pfaffian function of degree at most* $\Delta$ *when restricted to region* $\mathcal{A}_i$, *from a Pfaffian chain* $\mathcal{C}_{\boldsymbol{x}, \mathcal{A}_i}$ *of length at most* $q$ *and Pfaffian degree* $M$.

*Then, for any* $\delta \in (0, 1)$, *w.p. at least* $1 - \delta$ *over the draw of* $m$ *problem instances* $S = \{\boldsymbol{x}_1, \ldots, \boldsymbol{x}_m\} \sim \mathcal{D}^m$, *where* $\mathcal{D}$ *is any problem distribution over* $\mathcal{X}$, *we have*

$$\left| \sup_{\boldsymbol{a} \in \mathcal{A}} \mathbb{E}_{\boldsymbol{x} \sim \mathcal{D}} u_{\boldsymbol{a}}(\boldsymbol{x}) - \mathbb{E}_{\boldsymbol{x} \sim \mathcal{D}}[u_{\hat{\boldsymbol{a}}_S}(\boldsymbol{x})] \right| = \mathcal{O}\left( H\sqrt{\frac{q^2 d^2 + qd\log(\Delta + M) + qd\log d + \log n}{m}} + \xi + H\sqrt{\frac{\log(1/\delta)}{m}} \right).$$

*Here* $\hat{\boldsymbol{a}}_S \in \arg\max_{\boldsymbol{a} \in \mathcal{A}} \frac{1}{m} \sum_{i=1}^{m} u_{\boldsymbol{a}}(\boldsymbol{x})$.

*Proof.* From Lemma 5.3, we have $\text{Pdim}(\mathcal{V}) = \mathcal{O}(q^2 d^2 + qd\log(\Delta + M) + qd\log d + \log n)$, which implies

$$\mathscr{R}_m(\mathcal{V}) = \mathcal{O}\left( H\sqrt{\frac{q^2 d^2 + qd\log(\Delta + M) + qd\log d + \log n}{m}} \right).$$

From Lemma 5.4, we have

$$\mathscr{R}_m(\mathcal{U}) = \mathcal{O}\left(H\sqrt{\frac{q^2 d^2 + qd\log(\Delta + M) + qd\log d + \log n}{m}} + \xi\right).$$

A standard learning theory result implies the final claim. $\qquad\square$

A concrete application of the above general result to tuning the regularization parameter in regularized logistic regression appears in Section 6.3.

# 6 Applications of the Pfaffian piecewise structure framework

In this section, we demonstrate how to leverage our proposed framework to establish new statistical learning guarantees for under-explored data-driven algorithm design problems. For convenience, the notation used in this section might be redefined for each application.

## 6.1 Data-driven agglomerative hierarchical clustering

Agglomerative hierarchical clustering (Murtagh & Contreras, 2012), or AHC, is a versatile, two-stage clustering approach widely employed across various domains. In the first stage, data is organized into a hierarchical clustering tree, determining the order in which data points are merged into clusters. Subsequently, in the second stage, the cluster tree is pruned according to a specific objective function, for which some common choices are $k$-mean, $k$-median, or $k$-center (Lloyd, 1982; Xu & Wunsch, 2005), to obtain the final clusters.

The first stage of AHC involves carefully designing linkage functions, which measure the similarity between clusters and determine the pair of clusters to be merged at each step. These linkage functions require a pairwise distance function $\delta$ between data points and calculate the distance between clusters based on the distances of their constituent points in a specific manner. Common linkage functions include single-linkage, complete-linkage, and average-linkage, and it is possible to generate infinite families of linkage functions by interpolating them (Balcan et al., 2017), which can lead to different merge sequences when building the hierarchical clustering tree. Note that if two linkage functions generate the same hierarchical cluster tree, they will yield the same final clusters, irrespective of the objective function used in the subsequent pruning stage.

Although linkage functions are a crucial component of AHC, they are generally chosen heuristically without any theoretical guarantees. Recently, Balcan et al. (2017) proposed a principled data-driven approach for designing linkage functions. Similar to other data-driven settings, their analysis operates under the assumption that there exists an unknown, application-specific distribution for clustering instances. They then provide learning guarantees for some simple families of linkage functions, parameterized by a single parameter, that interpolates among single, complete, and average linkage. However, they assume that the pairwise distance function $\delta$ is fixed, whereas in practice, multiple distance functions, each with distinct properties and benefits, may be combined to achieve better performance. Subsequent work by Balcan et al. (2020a) proposes combining multiple pairwise distance functions by jointly learning their weights and the parameters of the linkage function. However, their analysis does not apply to all the families studied by Balcan et al. (2017).

**Contributions.** In this section, we instantiate theoretical guarantees for novel data-driven AHC settings, where we near-optimally learn the parameter of the merge function family and the combination of multiple distance functions. Compared to prior work, the setting we consider is more general—in prior work a combination of multiple distance metrics was only handled for simpler linear interpolations between single and complete linkage, while our techniques handle more sophisticated geometric interpolations as well that include average linkage.

### 6.1.1 Problem setting

**Distance function, linkage function, and linkage family.** Given a set of $n$ points $S \in \mathcal{X}^n$, where $\mathcal{X}$ denotes the data domain, and a distance function $\delta : \mathcal{X} \times \mathcal{X} \to \mathbb{R}_{\geq 0}$, the overall goal of clustering is to

partition $S$ into clusters such that some notion of intra-cluster distance is minimized, and/or some notion of inter-cluster distance is maximized. In the AHC approach, we first design a linkage function $m_\delta$ based on $\delta$, where $m_\delta(A, B)$ specifies the distance between two clusters $A$ and $B$. The cluster tree construction algorithm starts with $n$ singleton clusters, each containing a single data point, and successively merges the pair of clusters $A, B$ that minimizes the cluster-wise distance $m_\delta(A, B)$. This sequence of merges yields a hierarchical cluster tree, with the root corresponding to the entire set $S$, leaf nodes corresponding to individual points in $S$, and each internal node representing a subset of points in $S$ obtained by merging the point sets corresponding to its two child nodes. Subsequently, the cluster tree is pruned to obtain the final clusters via a dynamic programming procedure that optimizes a chosen objective function. Common objective functions include $k$-means, $k$-medians, and $k$-centers. Importantly, given a fixed objective function, if two linkage functions generate the same cluster tree for a given dataset, they will yield the same final clusters.

As discussed previously, the point-wise distance function $\delta$ can be a convex combination of several distance functions chosen from a given set of distance functions $\boldsymbol{\delta} = \{\delta_1, \ldots, \delta_L\}$, i.e., $\delta = \delta_{\boldsymbol{\beta}} = \sum_{i=1}^{L} \beta_i \delta_i$ for some $\boldsymbol{\beta} = (\beta_1, \ldots, \beta_L) \in \Delta(L)$. Here $\Delta(L) = \{\boldsymbol{\beta} \in \mathbb{R}^L \mid \beta_i \geq 0, \sum_{i=1}^{L} \beta_i = 1\}$ denotes the $(L-1)$-dimensional probability simplex. The combined distance function $\delta_{\boldsymbol{\beta}}$ is then used in the linkage function. In this work, we consider the following parameterized families of linkage functions:

$$
\mathcal{M}_1 = \left\{ m_{\delta_{\boldsymbol{\beta}}}^{1,\alpha} : (A, B) \mapsto \left( \min_{a \in A, b \in B} (\delta_{\boldsymbol{\beta}}(a, b))^\alpha + \max_{a \in A, b \in B} (\delta_{\boldsymbol{\beta}}(a, b))^\alpha \right)^{1/\alpha}, \alpha \in \mathbb{R} \cup \{-\infty, \infty\} \setminus \{0\} \right\},
$$

$$
\mathcal{M}_2 = \left\{ m_{\delta_{\boldsymbol{\beta}}}^{2,\alpha} : (A, B) \mapsto \left( \frac{1}{|A||B|} \sum_{a \in A, b \in B} (\delta_{\boldsymbol{\beta}}(a, b))^\alpha \right)^{1/\alpha}, \alpha \in \mathbb{R} \cup \{-\infty, \infty\} \setminus \{0\} \right\},
$$

$$
\mathcal{M}_3 = \left\{ m_{\boldsymbol{\delta}}^{3,\alpha} : (A, B) \mapsto \left( \frac{1}{|A||B|} \sum_{a \in A, b \in B} \Pi_{i \in [L]} (\delta_i(a, b))^{\alpha_i} \right)^{1/\sum_i \alpha_i}, \alpha_i \in \mathbb{R} \cup \{-\infty, \infty\} \setminus \{0\} \right\}.
$$

The linkage function family $\mathcal{M}_1$ interpolates between single and complete linkage. The linkage function family $\mathcal{M}_2$ is called the versatile linkage (Fernández & Gómez, 2020), which interpolates among single, complete, and average linkage. The family $\mathcal{M}_3$ is another generalization of single, complete, and average linkages that incorporates multiple pairwise distance functions. Note that in $\mathcal{M}_3$, we do not combine all the given distance functions $\boldsymbol{\delta} = \{\delta_1, \ldots, \delta_L\}$ into one but treat them separately. Precisely, if we set $\alpha_i = 1$, and $\alpha_j = 0$ for all $j \neq i$, we have average linkage with respect to the distance function $\delta_i$; if we set $\alpha_i = \infty$, and $\alpha_j = 0$ for all $j \neq i$, we have complete linkage with respect to the distance function $\delta_i$; and if we set $\alpha_i = -\infty$, and $\alpha_j = 0$ for $j \neq i$, we have the well-known single linkage with respect to $\delta_i$.

**Data-driven AHC.** In the data-driven setting, we are given multiple problem instances $P_1, \ldots, P_N$, where each problem instance $P_i = (S_i, \boldsymbol{\delta})$ consists of a set of points $S_i$ that need to be clustered and a fixed set of distance functions $\boldsymbol{\delta}$ that is shared across problem instances. Assuming that there exists an underlying problem distribution that represents a specific application domain, we aim to determine how many problem instances are sufficient to learn the parameters $\alpha$ of the linkage function families and the weights $\boldsymbol{\beta}$ of the distance functions (for the families $\mathcal{M}_1$ and $\mathcal{M}_2$). These questions are equivalent to analyzing the pseudo-dimension of the following classes of utility functions.

For $i \in \{1, 2\}$, let $A_i^{\alpha,\beta}$ denote the algorithm that takes a problem instance $P = (S, \boldsymbol{\delta})$ as input and returns a cluster tree $A_i^{\alpha,\beta}(S, \boldsymbol{\delta}) \in \mathcal{T}$, where $\mathcal{T}$ is the set of all possible cluster trees, by using the interpolated merge function $m_{\delta_{\boldsymbol{\beta}}}^{1,\alpha}$. Then given an objective function, for example, the $k$-means objective, the pruning function $\mu_k : \mathcal{T} \to \mathcal{S}_k$ takes as input a clustering tree, and returns a $k$-partition $\{\mathcal{P}_1, \ldots, \mathcal{P}_k\}$ of $S$ that minimizes such objective function. Then, given a target cluster $\mathcal{Y} = \{C_1, \ldots, C_k\}$, the performance of the algorithm $A_i^{\alpha,\beta}$ is measured by the Hamming distance between the produced clusters $\mu_k(A_i^{\alpha,\beta}(S, \boldsymbol{\delta})) = \{\mathcal{P}_1, \ldots, \mathcal{P}_k\}$

and the target clusters $\mathcal{Y} = \{C_1, \ldots, C_k\}$

$$u(\mu_k(A_i^{\alpha,\beta}(S,\boldsymbol{\delta})), \mathcal{Y}) = 1 - \min_{\sigma \in \mathbb{S}_k} \frac{1}{|S|} \sum_{i=1}^{k} |C_i \setminus P_{\sigma_i}|.$$

Here, the minimum is taken over the set of all permutations of $\mathbb{S}_k = \{1, \ldots, k\}$. We can clearly see that $\ell$ takes value in $\{0, \frac{1}{n}, \ldots, \frac{n-1}{n}, 1\}$. However, note that given an objective function, the cluster tree is equivalent to the produced clusters. Thus, the performance of the algorithm is completely determined by the cluster tree it produces, and for simplicity, we can express the performance of $A_i^{\alpha,\beta}$ as $u_i^{\alpha,\beta} : (S, \boldsymbol{\delta}) \mapsto v(A_i^{\alpha,\beta}(S,\boldsymbol{\delta}))$, where $v$ is a function that maps a cluster tree to a value in $[0,1]$.

Similarly, we denote $A_3^\alpha$ as the cluster tree building algorithm that takes $P = (S, \boldsymbol{\delta})$ as the input and returns a cluster tree $A_3^\alpha(S, \boldsymbol{\delta})$ by using the linkage function $m_{\boldsymbol{\delta}}^{3,\alpha}$. Again, we have that $u_3^\alpha : (S, \boldsymbol{\delta}) \mapsto v(A_3^\alpha(S, \boldsymbol{\delta}))$ represents the performance of the algorithm. We consider the following function classes that measure the performance of the above algorithm families:

$$\mathcal{H}_1 = \{u_1^{\alpha,\beta} : (S, \boldsymbol{\delta}) \mapsto u(A_1^{\alpha,\beta}(S, \boldsymbol{\delta})) \mid \alpha \in \mathbb{R} \cup \{-\infty, +\infty\}, \beta \in \Delta([L])\},$$
$$\mathcal{H}_2 = \{u_2^{\alpha,\beta} : (S, \boldsymbol{\delta}) \mapsto u(A_2^{\alpha,\beta}(S, \boldsymbol{\delta})) \mid \alpha \in \mathbb{R} \cup \{-\infty, +\infty\}, \beta \in \Delta([L])\},$$
$$\mathcal{H}_3 = \{u_3^\alpha : (S, \boldsymbol{\delta}) \mapsto u(A_3^\alpha(S, \boldsymbol{\delta})) \mid \alpha_i \in \mathbb{R} \cup \{-\infty, \infty\} \setminus \{0\}\}.$$

In the next section, we analyze the pseudo-dimension of the function classes described above, which provides insights into the number of problem instances required to learn near-optimal AHC parameters for a specific application domain.

### 6.1.2 Generalization guarantees for data-driven hierarchical clustering

In this section, we will leverage our proposed Pfaffian piecewise structure (Theorem 5.2) to establish the upper bounds for the pseudo-dimension of $\mathcal{H}_1$, $\mathcal{H}_2$, and $\mathcal{H}_3$ described above. First, we will instantiate a pseudo-dimension upper bound for $\mathcal{H}_1$, which is formalized as the following Theorem 6.1.

**Theorem 6.1.** *Let $\mathcal{H}_1$ be a class of functions*

$$\mathcal{H}_1 = \{u_1^{\alpha,\beta} : (S, \boldsymbol{\delta}) \mapsto u(A_1^{\alpha,\beta}(S, \boldsymbol{\delta})) \mid \alpha \in \mathbb{R} \cup \{-\infty, +\infty\}, \beta \in \Delta([L])\}$$

*mapping clustering instances $(S, \boldsymbol{\delta})$) to $[0,1]$ by using merge functions from class $\mathcal{M}_1$ and an arbitrary objective function. Then $\mathrm{Pdim}(\mathcal{H}_1) = \mathcal{O}(n^4 L^2)$.*

*Proof.* **Overview.** The high-level idea is to show that the dual utility function $u_{1,P}^* : (\alpha, \beta) \mapsto u(A_1^{\alpha,\beta}(S, \boldsymbol{\delta}))$ for any fixed problem instance $P = (S, \boldsymbol{\delta})$ exhibits a piecewise structure: its parameter space is partitioned by multiple boundary functions, and within each region, the cluster tree remains unchanged, implying that the utility function is constant. We then characterize the number and complexity of the boundary functions, which we show belong to a Pfaffian system. Subsequently, we can utilize our framework to obtain a bound on the pseudo-dimension of $\mathcal{H}_1$.

**Proof details.** Fix a problem instance $P = (S, \boldsymbol{\delta})$, and consider the dual utility function $u_{1,P}^* : (\alpha, \beta) \mapsto u(A_1^{\alpha,\beta}(S, \boldsymbol{\delta}))$. Suppose $A, B \subseteq S$ and $A', B' \subseteq S$ are two candidate clusters at some merge step of the algorithm. Then $A, B$ is preferred for merging over $A', B'$ iff

$$\left( \min_{a \in A, b \in B} (\delta_{\boldsymbol{\beta}}(a,b))^\alpha + \max_{a \in A, b \in B} (\delta_{\boldsymbol{\beta}}(a,b))^\alpha \right)^{1/\alpha} \leq \left( \min_{a \in A', b \in B'} (\delta_{\boldsymbol{\beta}}(a,b))^\alpha + \max_{a \in A', b \in B'} (\delta_{\boldsymbol{\beta}}(a,b))^\alpha \right)^{1/\alpha},$$

or equivalently,

$$(\delta_{\boldsymbol{\beta}}(a_1,b_1))^\alpha + (\delta_{\boldsymbol{\beta}}(a_2,b_2))^\alpha \leq (\delta_{\boldsymbol{\beta}}(a_1',b_1'))^\alpha + (\delta_{\boldsymbol{\beta}}(a_2',b_2'))^\alpha,$$

where $(a_1, b_1) \in \arg\min_{a \in A, b \in B}(\delta_{\boldsymbol{\beta}}(a,b))^\alpha$, $(a_2, b_2) \in \arg\max_{a \in A, b \in B}(\delta_{\boldsymbol{\beta}}(a,b))^\alpha$, $(a'_1, b'_1) \in \arg\min_{a \in A', b \in B'}(\delta_{\boldsymbol{\beta}}(a,b))^\alpha$, and $(a'_2, b'_2) \in \arg\max_{a \in A', b \in B'}(\delta_{\boldsymbol{\beta}}(a,b))^\alpha$. Each possible choice of the points $a_1, b_1, a_2, b_2, a'_1, b'_1, a'_2, b'_2$ corresponds to a boundary function in the form

$$\mathbb{I}((\delta_{\boldsymbol{\beta}}(a_1, b_1))^\alpha + (\delta_{\boldsymbol{\beta}}(a_2, b_2))^\alpha - (\delta_{\boldsymbol{\beta}}(a'_1, b'_1))^\alpha + (\delta_{\boldsymbol{\beta}}(a'_2, b'_2))^\alpha \geq 0).$$

Among all possible choices of the points $a_1, b_1, a_2, b_2, a'_1, b'_1, a'_2, b'_2$, we get at most $n^8$ distinct boundary conditions across which the merge decision at any step of the algorithm may change

We next show that the boundary functions constitute a Pfaffian system in $\alpha, \beta_1, \ldots, \beta_L$ and bound its complexity. For each pair of points $a, b \in S$, define $f_{a,b}(\alpha, \boldsymbol{\beta}) := \frac{1}{\delta_{\boldsymbol{\beta}}(a,b)}$, $g_{a,b}(\alpha, \boldsymbol{\beta}) := \ln \delta_{\boldsymbol{\beta}}(a,b)$ and $h_{a,b}(\alpha, \boldsymbol{\beta}) := \delta_{\boldsymbol{\beta}}(a,b)^\alpha$. Recall $\delta_{\boldsymbol{\beta}}(a,b) = \sum_{i=1}^L \beta_i \delta_i(a,b)$. Consider the chain $\mathcal{C}(\alpha, \boldsymbol{\beta}, f_{a,b}, g_{a,b}, h_{a,b})$ of length $q = 3n^2$, for $a, b \in S$. We will show that these functions form a Pfaffian chain of Pfaffian degree $M = 2$. Indeed, we have for each $a, b \in S$,

$$\frac{\partial f_{a,b}}{\partial \alpha} = 0, \qquad \frac{\partial f_{a,b}}{\partial \beta_i} = -\delta_i(a,b)f_{a,b}^2,$$

$$\frac{\partial g_{a,b}}{\partial \alpha} = 0, \qquad \frac{\partial g_{a,b}}{\partial \beta_i} = \delta_i(a,b)f_{a,b},$$

$$\frac{\partial h_{a,b}}{\partial \alpha} = g_{a,b}h_{a,b}, \qquad \frac{\partial h_{a,b}}{\partial \beta_i} = \delta_i(a,b)f_{a,b}h_{a,b},$$

which establishes the claim. The boundary conditions can clearly be all written in terms of the functions $\{h_{a,b}\}_{a,b \in S}$, meaning that the degree $\Delta = 1$. Note that the piece functions are constant and can only take value in $\{0, \frac{1}{n}, \ldots, 1\}$, meaning that $k_{\mathcal{F}} = n+1$. Then we conclude that $\mathcal{H}_1^*$ admits $(n+1, n^8, 3n^2, 2, 1, L+1)$-Pfaffian piecewise structure. Applying Theorem 5.2 now gives that $\mathrm{Pdim}(\mathcal{H}_1) = \mathcal{O}(n^4 L^2 + n^2 L \log L + L \log(n^8 + n + 1)) = \mathcal{O}(n^4 L^2)$. $\qquad \square$

Similarly, we also have the upper bound for the pseudo-dimension of $\mathcal{H}_2$, and $\mathcal{H}_3$.

**Theorem 6.2.** *Let $\mathcal{H}_2$ be a class of functions*

$$\mathcal{H}_2 = \{u_2^{\alpha, \boldsymbol{\beta}} : (S, \boldsymbol{\delta}) \mapsto u(A_2^{\alpha, \boldsymbol{\beta}}(S, \boldsymbol{\delta})) \mid \alpha \in \mathbb{R} \cup \{-\infty, +\infty\}, \boldsymbol{\beta} \in \Delta([L])\}$$

*mapping clustering instances $(S, \boldsymbol{\delta})$ to $[0,1]$ by using merge functions from class $\mathcal{M}_2$ and an arbitrary merge function. Then $\mathrm{Pdim}(\mathcal{H}_2) = \mathcal{O}(n^4 L^2)$.*

**Theorem 6.3.** *Let $\mathcal{H}_3$ be a class of functions*

$$\mathcal{H}_3 = \{u_3^\alpha : (S, \boldsymbol{\delta}) \mapsto u(A_3^\alpha(S, \boldsymbol{\delta})) \mid \alpha_i \in \mathbb{R} \cup \{-\infty, \infty\} \setminus \{0\}\}$$

*mapping clustering instances $(S, \boldsymbol{\delta})$ to $[0,1]$ by using merge functions from class $\mathcal{M}_3$. Then $\mathrm{Pdim}(\mathcal{H}_3) = \mathcal{O}(n^4 L^2)$.*

The detailed proofs of Theorem 6.2, 6.3 for the function classes $\mathcal{H}_2$, and $\mathcal{H}_3$ can be found in Appendix D.1. Although these function classes share the same pseudo-dimension asymptotic upper bound, their structures differ, necessitating separate analyses and leading to distinct Pfaffian piecewise structures. To show that the dual function classes of $\mathcal{H}_1$, $\mathcal{H}_2$, and $\mathcal{H}_3$ admit Pfaffian piecewise structure, we analyze the transition condition on the hyperparameters when the preference for merging two candidate clusters $A, B$ switches to merging a different pair of clusters $A', B'$ instead, at any merge step of the hierarchical clustering algorithm. The transition condition corresponds to an equality involving Pfaffian functions of the parameters $\alpha$ and $\boldsymbol{\beta}$. All of such equations corresponding to each tuple $(A, B, A', B') \subset S^4$ will divide the parameter space into regions, in each of which the AHC algorithm produces the same clustering tree, leading to the same performance. After this step, we construct the Pfaffian chains for each function in function classes $\mathcal{H}_1$, $\mathcal{H}_2$, and $\mathcal{H}_3$, where the difference naturally lies in the form of functions in those function classes. We then carefully analyze the complexities of the Pfaffian chain of those Pfaffian functions to obtain the above bounds.

## 6.2 Data-driven graph-based semi-supervised learning

Semi-supervised learning (Chapelle et al., 2010) is a learning paradigm designed for settings where labeled data is scarce due to expensive labeling processes. This paradigm leverages unlabeled data in addition to a small set of labeled samples for effective learning. A common semi-supervised learning approach is the graph-based method (Zhu & Goldberg, 2009; Chapelle et al., 2010), which captures relationships between labeled and unlabeled data using a graph. In this approach, nodes represent data points, and edges are constructed based on the similarity between data point pairs, measured by a given distance function. Optimizing a labeling function on this graph helps propagate labels from the labeled data to the unlabeled data.

A large body of research focuses on how to learn such labeling functions given the graph, including using *st*-mincuts (Blum & Chawla, 2001), optimizing harmonic objective with soft mincuts (Zhu et al., 2003), label propagation (Zhu & Ghahramani, 2002), among others. Recent work by Du et al. (2025) examines the problem of data-driven tuning of hyperparameters in label propagation based algorithms. However, it is crucial to design a good graph by setting the edge weights between the data point appropriately in order for these label optimization algorithms to work well (Zhu & Goldberg, 2009).

Despite its significance, the graph is usually considered given or constructed using heuristic methods without theoretical guarantees (Zhu, 2005; Zemel & Carreira-Perpiñán, 2004). Recently, Balcan & Sharma (2021) proposed a novel data-driven approach for constructing the graph, by learning the parameters of the graph kernel underlying the graph construction, from the problem instances at hand. Each problem instance $P$ consists of sets of labeled $\mathcal{L}$ and unlabeled data $\mathcal{U}$ and a distance metric $d$. Assuming that all problem instances are drawn from an underlying, potentially unknown distribution, they provide guarantees for learning near-optimal graph kernel parameters for such a distribution. Nonetheless, they consider only a single distance function, whereas in practical applications, combining multiple distance functions, each with its unique characteristics, can improve the graph quality and typically result in better outcomes compared to utilizing a single metric (Balcan et al., 2005).

**Contributions.** In this section, we consider a generalized and more practical setting for data-driven graph-based semi-supervised learning, where we learn the parameters of the commonly-used Gaussian RBF kernel $w_{\sigma,\boldsymbol{\beta}}(u,v) = \exp(-\delta(u,v)/\sigma^2)$ and the weights $\boldsymbol{\beta} \in \Delta(L) = \{\boldsymbol{\beta} \in \mathbb{R}^L \mid \beta_i \geq 0, \sum_{i=1}^{L} \beta_i = 1\}$ of $\delta = \sum_{i=1}^{L} \beta_i \delta_i$ which is a convex combination of multiple distance functions for constructing the graph.

### 6.2.1 Problem setting

**Graph-based semi-supervised learning with Gaussian RBF Kernel.** In the graph-based semi-supervised learning with Gaussian RBF kernel, we are given a few labeled samples $\mathcal{L} \subset \mathcal{X} \times \mathcal{Y}$, a large number of unlabeled points $\mathcal{U} \subset \mathcal{X}$, and a set of distance functions $\boldsymbol{\delta} = \{\delta_1, \ldots, \delta_L\}$, where $\delta_i : \mathcal{X} \times \mathcal{X} \to \mathbb{R} \geq 0$ for $i = 1, \ldots, L$. Here, $\mathcal{X}$ denotes the data space, and $\mathcal{Y} = \{0, 1\}$ denotes the binary classification label space. To extrapolate labels from $\mathcal{L}$ to $\mathcal{U}$, a graph $G^{\sigma,\boldsymbol{\beta}}$ is constructed with the node set $\mathcal{L} \cup \mathcal{U}$ and edges weighted by the Gaussian RBF graph kernel $w_{\sigma,\boldsymbol{\beta}}(x_1, x_2) = \exp(-\delta(x_1, x_2)/\sigma^2)$ for $x_1, x_2 \in \mathcal{L} \cup \mathcal{U}$, where $\sigma$ is the bandwidth parameter, and $\delta = \sum_{i=1}^{L} \beta_i \delta_i$ is a convex combination of the given distance functions. After constructing the graph $G^{\sigma,\boldsymbol{\beta}}$, a popular graph labeling algorithm called the harmonic method (Zhu et al., 2003) is employed. It assigns soft labels by minimizing the following quadratic objective:

$$\frac{1}{2} \sum_{u,v} w_{\sigma,\boldsymbol{\beta}}(x_1, x_2)(f(x_1) - f(x_2))^2 = f^T(D - W)f,$$

where $f \in [0,1]^n$ is the soft label vector that includes labels of labeled samples $f(x_{\mathcal{L}})$ ($x \in \mathcal{L}$) and the prediction variables of unlabeled labels $f(x_{\mathcal{U}})$ ($x_{\mathcal{U}} \in \mathcal{U}$) that we want to minimize over. Besides, $n = |\mathcal{U}| + |\mathcal{L}|$ is the total number of samples, $W$ denotes the graph adjacency matrix $W_{x_1 x_2} = w_{\alpha,\boldsymbol{\beta}}(x_1, x_2)$, and $D$ is the diagonal matrix with entries $D_{ii} = \sum_j W_{ij}$. The final predictions are obtained by rounding $f_{x_{\mathcal{U}}}$ for $x_{\mathcal{U}} \in \mathcal{U}$, i.e. predicting $\mathbb{I}\{f(x_{\mathcal{U}}) \geq \frac{1}{2}\}$, denoted by $G^{\sigma,\boldsymbol{\beta}}(\mathcal{L}, \mathcal{U}, \boldsymbol{\delta})$. Let $u^{\sigma,\boldsymbol{\beta}} : (\mathcal{L}, \mathcal{U}, \boldsymbol{\delta}) \mapsto [0, 1]$ denote the utility of the algorithm corresponding to $\sigma$ and $\boldsymbol{\beta}$, which is given by $u^{\sigma,\boldsymbol{\beta}} = 1 - l^{\sigma,\boldsymbol{\beta}}$, where $l^{\sigma,\boldsymbol{\beta}}$ is the average 0-1 binary classification loss of the predictions of the above algorithm when the graph is built with parameters $\sigma, \boldsymbol{\beta}$.

**Data-driven graph-based semi-supervised learning.** In the data-driven setting, we are given multiple problem instances $P_1, \ldots, P_i$, each $P_i$ is represented as a tuple $(\mathcal{L}_i, \mathcal{U}_i, \boldsymbol{\delta})$ of a set of labeled samples $\mathcal{L}_i$, a set of a unlabeled samples $\mathcal{U}_i$, and a set of distance functions $\boldsymbol{\delta}$ that is shared across problem instances. Assuming that there is an underlying (unknown) problem distribution that represents a specific application domain, we want to know how many problem instances are sufficient to learn the best parameters $\alpha, \boldsymbol{\beta}$ that are near-optimal for such a problem distribution. To do that, we want to analyze the pseudo-dimension of the following function class:

$$\mathcal{G} = \{u^{\sigma,\boldsymbol{\beta}} : (\mathcal{L}, \mathcal{U}, \boldsymbol{\delta}) \mapsto u(G^{\sigma,\boldsymbol{\beta}}(\mathcal{L}, \mathcal{U}, \boldsymbol{\delta})) \mid \sigma \in \mathbb{R} \setminus \{0\}, \boldsymbol{\beta} \in \Delta(L)\}.$$

### 6.2.2 Generalization guarantees for data-driven semi-supervised learning with Gaussian RBF kernel and multiple distance functions

We now instantiate the main result in this section, which establishes an upper bound for the pseudo-dimension of the function class $\mathcal{G}$ described above.

**Theorem 6.4.** *Let $\mathcal{G}$ be a class of functions mapping semi-supervised learning instances $(\mathcal{L}, \mathcal{U}, \boldsymbol{\delta})$ to $[0, 1]$. Then $\mathrm{Pdim}(\mathcal{G}) = \mathcal{O}(n^4 L^2)$, where $n = |\mathcal{L}| + |\mathcal{U}|$ is the total number of samples in each problem instance, and $L = |\boldsymbol{\delta}|$ is the number of distance functions.*

*Proof.* **Technical overview.** Fix a problem instance $P = (\mathcal{L}, \mathcal{U}, \boldsymbol{\delta})$, we will show that the dual utility function function $u_P^*(\sigma, \boldsymbol{\beta}) := u^{\sigma,\boldsymbol{\beta}}(\mathcal{L}, \mathcal{U}, \boldsymbol{\delta})$ is piecewise constant and characterize the number and complexity of the boundary functions which we will show belong to a Pfaffian system. This implies a bound on the pseudo-dimension of $\mathcal{G}$ following our Pfaffian piecewise structure Theorem 5.2.

**Proof details.** First, the quadratic objective minimization has a closed-form solution (Zhu et al., 2003), given by

$$f_{\mathcal{U}} = (D_{\mathcal{U}\mathcal{U}} - W_{\mathcal{U}\mathcal{U}})^{-1} W_{\mathcal{U}\mathcal{L}} f_{\mathcal{L}},$$

where $W$ denotes the graph adjacency matrix, $D$ is the diagonal matrix with entries $D_{x_i, x_i} = \sum_{j=1}^{n} w_{\sigma,\boldsymbol{\beta}}(x_i, x_j)$, and subscripts $\mathcal{U}, \mathcal{L}$ refer to the unlabeled and labeled points, respectively. Here $f_{\mathcal{L}} \in \{0, 1\}^{|\mathcal{L}|}$ is the ground truth label of samples in the labeled set $\mathcal{L}$.

The key challenge now is to analyze the formula $(D_{\mathcal{U}\mathcal{U}} - W_{\mathcal{U}\mathcal{U}})^{-1}$ of which the element will be shown to be Pfaffian functions of $\sigma, \boldsymbol{\beta}$. Recall that $w_{\sigma,\boldsymbol{\beta}}(x_i, x_j) = \exp(-\delta_{\boldsymbol{\beta}}(x_i, x_j)/\sigma^2)$ for $x_i, x_j \in \mathcal{L} \cup \mathcal{U}$. First, we recall the identity $A^{-1} = \frac{\mathrm{adj}(A)}{\det(A)}$, for any invertible matrix $A$, where $\mathrm{adj}(A)$ and $\det A$ are the adjoint and determinant of matrix $A$, respectively. Therefore, we can see that any element of $(D_{\mathcal{U}\mathcal{U}} - W_{\mathcal{U}\mathcal{U}})^{-1}$ is a rational function of $w_{\sigma,\boldsymbol{\beta}}(x_i, x_j)$ of degree at most $|\mathcal{U}|$ (i.e. a ratio of polynomial functions where both the numerator and the denominator have degrees at most $|\mathcal{U}|$).

Now, consider the Pfaffian chain $\mathcal{C}(\sigma, \boldsymbol{\beta}, \frac{1}{\sigma}, w_{11}, \ldots, w_{nn})$ with $L + 1$ variables $\sigma, \boldsymbol{\beta}$, and of length $q = n^2 + 1$. To see the Pfaffian degree of $\mathcal{C}$, note that for any pair of nodes $x_i, x_j \in \mathcal{U} \cup \mathcal{L}$, we have

$$\frac{\partial w_{\sigma,\boldsymbol{\beta}}(x_i, x_j)}{\partial \beta_k} = -\frac{\delta_k(x_i, x_j)}{\sigma^2} \exp\left(-\frac{\delta_{\boldsymbol{\beta}}(x_i, x_j)}{\sigma^2}\right) = -\delta_k(x_i, x_j) g^2 w_{\sigma,\boldsymbol{\beta}}(x_i, x_j), \quad \text{for } k = 1, \ldots, L,$$

$$\frac{\partial w_{\sigma,\boldsymbol{\beta}}(x_i, x_j)}{\partial \sigma} = \frac{2\delta_{\boldsymbol{\beta}}(x_i, x_j)}{\sigma^3} \exp\left(-\frac{\delta_{\boldsymbol{\beta}}(x_i, x_j)}{\sigma^2}\right) = 2\delta_{\boldsymbol{\beta}}(x_i, x_j) g^3 w_{\sigma,\boldsymbol{\beta}}(x_i, x_j).$$

Thus, the Pfaffian chain $\mathcal{C}$ has Pfaffian degree $M = 5$.

Now, consider the dual utility function $v_{\mathcal{L},\mathcal{U},\boldsymbol{\delta}}(\sigma, \boldsymbol{\beta})$. Note that

$$u_{\mathcal{L},\mathcal{U},\boldsymbol{\delta}}(\sigma, \boldsymbol{\beta}) = \frac{1}{|\mathcal{U}|} \sum_{x_{\mathcal{U}} \in \mathcal{U}} \mathbb{I}(\mathbb{I}(f(x_{\mathcal{U}}) \geq 1/2) = g(x_{\mathcal{U}})),$$

where $g(x_{\mathcal{U}})$ is the ground-truth label of unlabeled node $x_{\mathcal{U}} \in \mathcal{U}$, using for evaluation purpose. We can see that, for each sample $x_{\mathcal{U}} \in \mathcal{U}$, $\mathbb{I}(f(x_{\mathcal{U}}) \geq 1/2) = \mathbb{I}\left(\frac{f^{(1)}(x_{\mathcal{U}})}{f^{(2)}(x_{\mathcal{U}})} \geq 1/2\right) = \mathbb{I}\left(f^{(1)}(x_{\mathcal{U}}) \geq 1/2 f^{(2)}(x_{\mathcal{U}})\right)$ is a boundary function. Both functions $f^{(1)}(x_{\mathcal{U}}), f^{(2)}(x_{\mathcal{U}})$ are Pfaffian functions from chain $\mathcal{C}$ and of degree $|\mathcal{U}|$. In each region determined by the sign vector $b_{\mathcal{U}} = (b(x_{\mathcal{U}}))_{x_{\mathcal{U}} \in \mathcal{U}}$, where $b(x_{\mathcal{U}}) = \mathbb{I}(f(x_{\mathcal{U}}) \geq 1/2)$ for $x_{\mathcal{U}} \in \mathcal{U}$, the dual utility function $u_{\mathcal{L},\mathcal{U},\boldsymbol{\delta}}(\sigma, \boldsymbol{\beta})$ is a constant functions. We can see that there are at most $|\mathcal{U}| + 1$ such constant functions that $v_{\mathcal{L},\mathcal{U},\boldsymbol{\delta}}(\sigma, \boldsymbol{\beta})$ can take. Therefore, by Definition 8, the dual function class $\mathcal{G}^*$ is $(|\mathcal{U}| + 1, |\mathcal{U}| + 1, n^2 + 1, 5, |\mathcal{U}|)$-Pfaffain piecewise structured. We can apply Theorem 5.2 to get

$$\mathrm{Pdim}(\mathcal{G}) = \mathcal{O}(n^4 L^2 + n^2 L \log(|\mathcal{U}|) + n^2 L \log L + L \log |\mathcal{U}|).$$

Noting $|\mathcal{U}| \leq n$ and simplifying yields the claimed bound. $\qquad\square$

## 6.3 Data-driven regularized logistic regression

Logistic regression (James et al., 2013) is a fundamental statistical technique and widely used classification model with numerous applications across diverse domains, including healthcare screening (Armitage et al., 2008), market forecasting (Linoff & Berry, 2011), and engineering safety control (Palei & Das, 2009). To mitigate overfitting and enhance robustness, regularization for sparsity ($\ell_1$) and shrinkage ($\ell_2$) is commonly employed in logistic regression (Mohri et al., 2018; Murphy, 2012). In regularized logistic regression and regularized linear models in general, the regularization parameters, which control the magnitude of regularization, play a crucial role (Mohri et al., 2018; Murphy, 2012; James et al., 2013) and must be carefully chosen. Setting regularization parameters too high may lead to underfitting, while setting them too low may nullify the effectiveness of regularization. In practice, a common strategy for selecting regularization parameters is cross-validation, which is known to lack theoretical guarantees, in general (Zhang, 2009; Chichignoud et al., 2016).

Recently, Balcan et al. (2023b) proposed a data-driven approach for selecting regularization parameters in regularized logistic regression. Their methodology considers each regression problem, comprising training and validation sets, as a problem instance drawn from an underlying problem distribution. The objective is to leverage the available problem instances to determine regularization parameters that minimize the validation loss for any future problem instance sampled from the same distribution.

**Contribution.** In this section, we will demonstrate the applicability of Lemma 5.3 by recovering a result by Balcan et al. (2023b).

### 6.3.1 Problem setting

**Regularized logistic regression.** We closely follow the data-driven regularized logistic regression setting by Balcan et al. (2023b). A problem instance $P = (\boldsymbol{X}, \boldsymbol{y}, \boldsymbol{X}_{\mathrm{val}}, \boldsymbol{y}_{\mathrm{val}}) \in \mathcal{R}_{m,p,m'} = \mathbb{R}^{m \times p} \times \{\pm 1\}^m \times \mathbb{R}^{m' \times p} \times \{\pm 1\}^{m'}$, where $m' < m$, consists of a training set $(\boldsymbol{X}, \boldsymbol{y})$ of $m$ samples and a validation set $(\boldsymbol{X}_{\mathrm{val}}, \boldsymbol{y}_{\mathrm{val}})$ of $m'$ samples. Given a regularization parameter $\lambda \in [\lambda_{\min}, \lambda_{\max}]$, where $0 < \lambda_{\min} < \lambda_{\max}$, the regularized logistic regression solves for the coefficients $\hat{\boldsymbol{\beta}}_{(X,y)}$ that is the best fit for the training set $(\boldsymbol{X}, \boldsymbol{y})$, i.e.,

$$\hat{\boldsymbol{\beta}}_{(\boldsymbol{X},\boldsymbol{y})}(\lambda) = \arg \min_{\boldsymbol{\beta} \in \mathbb{R}^p} l(\boldsymbol{\beta}, (\boldsymbol{X}, \boldsymbol{y})) + \lambda R(\boldsymbol{\beta}),$$

where $l(\boldsymbol{\beta}, (\boldsymbol{X}, \boldsymbol{y})) = \frac{1}{m} \sum_{i=1}^m \log(1 + \exp(-y_i \boldsymbol{x}_i^\top \boldsymbol{\beta}))$ denotes the logistic loss, and $R(\boldsymbol{\beta})$ is either sparsity ($\|\boldsymbol{\beta}\|_1$) or shrinkage ($\|\boldsymbol{\beta}\|_2^2$) regularizer.

**Data-driven regularized logistic regression.** In the data-driven setting, we assume that there is an unknown, application specific problem distribution $\mathcal{D}$ over $\mathcal{R}_{m,p,m'}$. We then can define the optimal hyperparameter $\lambda^*$ for such problem distribution $\mathcal{D}$ as

$$\lambda^* \in \arg \min_{\lambda \in [\lambda_{\min}, \lambda_{\max}]} \mathbb{E}_{P \sim \mathcal{D}}[h_\lambda(P)],$$

where,

$$h_\lambda(P) = H - l(\hat{\boldsymbol{\beta}}_{\boldsymbol{X},\boldsymbol{y}}(\lambda), (\boldsymbol{X}_{\text{val}}, \boldsymbol{y}_{\text{val}})) = H - \frac{1}{m'} \sum_{i=1}^{m'} \log(1 + \exp(-y_i \boldsymbol{x}_i^\top \hat{\beta}_{(\boldsymbol{X},\boldsymbol{y})}(\lambda)))$$

$$= H - \frac{1}{m'} \log\left(\prod_{i=1}^{m'} (1 + \exp(-y_i \boldsymbol{x}_i^\top \hat{\beta}_{(\boldsymbol{X},\boldsymbol{y})}(\lambda)))\right),$$

which is the validation utility function for the problem instance $P$ corresponding to the regularization hyperparameter $\lambda$. Since the problem distribution $\mathcal{D}$ is unknown, we assume that we are given a set $S = \{P_1, \ldots, P_N\}$ of $N$ problem instances drawn i.i.d. from $\mathcal{D}$, and we set the regularization hyperparameter $\hat{\lambda}_S$ via ERM, i.e.,

$$\hat{\lambda}_S \in \underset{\lambda \in [\lambda_{\min}, \lambda_{\max}]}{\arg\max} \frac{1}{N} \sum_{i=1}^{N} h_\lambda(P_i).$$

Our goal is to determine how many problem instances $N$ do we need such that with high probability, the performance of the optimal hyperparameter $\lambda^*$ and the hyperparameter $\hat{\lambda}_S$ tuned using the set of problem instances $S = \{P_1, \ldots, P_N\}$ are sufficiently close. Define $\mathcal{H} = \{h_\lambda : \mathcal{R}_{m,p,m'} \to [0, H] \mid \lambda \in [\lambda_{\min}, \lambda_{\max}]\}$. The goal above is equivalent to analyzing the learning-theoretic complexity of the function class $\mathcal{H}$.

### 6.3.2 Generalization guarantee for data-driven regularized logistic regression

As discussed previously, the challenge for analyzing the learnability of $\mathcal{H}$ is the unknown structure of $h_\lambda(P)$ as a function of problem instance $P$. Even if we consider the dual function class $\mathcal{H}^* = \{h_P^* : [\lambda_{\min}, \lambda_{\max}] \to [0, H] \mid P \in \mathcal{R}_{m,p,m'}\}$, it is also hard to analyze the structure of $\mathcal{H}^*$ as we do not have the explicit form of $h_P^*(\lambda)$. Hence, the approach here is to construct a surrogate function class $\mathcal{V}_\epsilon^*$ that is sufficiently "close" to $\mathcal{H}^*$ and is more well-behaved, and then we will use the idea proposed in Section 5.3.1 to recover the learning guarantee for $\mathcal{H}$.

Using this approach, we recover the following result by Balcan et al. (2023b), which establishes learning guarantees for hyperparameter tuning for regularized logistic regression.

**Theorem 6.5.** *Consider the problem of tuning regularization parameter for regularized logistic regression with $\ell_1$ (or $\ell_2$) regularizer under data-driven setting. Consider the function class $\mathcal{H} = \{h_\lambda : \mathcal{R}_{m,p,m'} \to \mathbb{R} \mid \lambda \in [\lambda_{\min}, \lambda_{\max}]\}$, where $h_\lambda(P)$ is the validation utility for the problem instance $P$ and with regularization hyperparameter $\lambda$. Then for any $\delta \in (0, 1)$, with probability at least $1 - \delta$ over the draw of $m$ problem instances $S = \{P_1, \ldots, P_N\} \sim \mathcal{D}^N$, where $\mathcal{D}$ is some problem distribution over $\mathcal{R}_{m,p,m'}$, we have*

$$\mathcal{O}\left(\sqrt{\frac{m^2 + \log(1/\epsilon)}{N}} + \epsilon^2 + \sqrt{\frac{\log(1/\delta)}{N}}\right) + \mathbb{E}_{P \sim \mathcal{D}}[h_{\hat{\lambda}_S}(P)] \geq \sup_{\lambda \in [\lambda_{\min}, \lambda_{\max}]} \mathbb{E}_{p \sim \mathcal{D}}[h_\lambda(P)],$$

*where $\epsilon > 0$ is some sufficiently small value (satisfying Theorem D.1).*

*Proof.* **Technical overview.** The idea is to construct a surrogate dual function class $\mathcal{V}_\epsilon^* = \{v_{P,\epsilon}^* : [\lambda_{\min}, \lambda_{\max}] \to [0, H]\}$ that is sufficiently close to $\mathcal{H}^*$. In other words, given any sufficiently small $\epsilon$ (see Theorem D.1), we want to construct $\mathcal{V}_\epsilon^*$ such that for any problem instance $P \in \mathcal{R}_{m,p,m'}$, we have $\|v_{P,\epsilon}^* - h_P^*\|_\infty < \epsilon$. This can be done using Theorem D.1. Moreover, we can partition the space $[\lambda_{\min}, \lambda_{\max}]$ into $\frac{1}{\epsilon}(\lambda_{\max} - \lambda_{\min})$ intervals, over each of which the function $v_{P,\epsilon}^*$ is a Pfaffian function from a Pfaffian chain with bounded Pfaffian complexity. We then use our results from Section 5.3 (Theorem 5.3) to bound the statistical complexity of the surrogate primal function class $\mathcal{V}_\epsilon$, which then can convert to the statistical complexity of $\mathcal{H}$.

**Proof details.** We now go to the details of the proof, which consists of the following steps.

**Constructing the surrogate dual and primal function classes $\mathcal{V}_\epsilon^*$ and $\mathcal{V}_\epsilon$.** To construct such a surrogate dual function class $\mathcal{V}_\epsilon^*$ for $\mathcal{H}^*$, for any problem instance $P = (\boldsymbol{X}, \boldsymbol{y}, \boldsymbol{X}_{\text{val}}, \boldsymbol{y}_{\text{val}})$, Balcan et al. (2023b) first approximate the regularized logistic regression estimator $\hat{\beta}_{(\boldsymbol{X},\boldsymbol{y})}(\lambda)$ by $\beta_{(\boldsymbol{X},\boldsymbol{y})}^{(\epsilon)}(\lambda)$, using Algorithm 1 (see Appendix 6.3) if $R(\boldsymbol{\beta}) = \|\boldsymbol{\beta}\|_1$ (or Algorithm 2 if $R(\boldsymbol{\beta}) = \|\boldsymbol{\beta}\|_2^2$). Intuitively, for a sufficiently small $\epsilon$, the search space $[\lambda_{\min}, \lambda_{\max}]$ can be divided into $\frac{1}{\epsilon}(\lambda_{\max} - \lambda_{\min})$ intervals. Within each interval, $\beta_{(\boldsymbol{X},\boldsymbol{y})}^{(\epsilon)}(\lambda)$ is a linear function of $\lambda$, and there exists a uniform error bound of $O(\epsilon^2)$ for the approximation error $\|\beta_{(\boldsymbol{X},\boldsymbol{y})}^{(\epsilon)}(\lambda) - \hat{\beta}_{(\boldsymbol{X},\boldsymbol{y})}(\lambda)\|_2$ (formalized in Theorem D.1). The surrogate dual validation utility function class $\mathcal{V}_\epsilon^*$ can now be defined as $\mathcal{V}_\epsilon^* = \{v_{P,\epsilon}^* : [\lambda_{\min}, \lambda_{\max}] \to \mathbb{R} \mid P \in \mathcal{R}_{m,p,m'}\}$. Here

$$v_{P,\epsilon}^*(\lambda) = H - l(\beta_{(\boldsymbol{X},\boldsymbol{y})}^{(\epsilon)}(\lambda), (\boldsymbol{X}_{\text{val}}, \boldsymbol{y}_{\text{val}})) = H - \frac{1}{m'} \sum_{i=1}^{m'} \log(1 + \exp(-y_i x_i^\top \beta_{(\boldsymbol{X},\boldsymbol{y})}^{(\epsilon)}(\lambda)))$$

$$= H - \frac{1}{m'} \log \left( \prod_{i=1}^{m'} (1 + \exp(-y_i x_i^\top \beta_{(\boldsymbol{X},\boldsymbol{y})}^\epsilon(\lambda))) \right),$$

and $\beta_{(\boldsymbol{X},\boldsymbol{y})}^{(\epsilon)}(\lambda)$ is defined as in Theorem D.1. An important property of $\beta_{(\boldsymbol{X},\boldsymbol{y})}^{(\epsilon)}(\lambda)$ is its piecewise linear structure: we can partition $[\lambda_{\min}, \lambda_{\max}]$ into $\frac{1}{\epsilon}(\lambda_{\max} - \lambda_{\min})$ intervals, and in each interval $[\lambda_{\min} + t\epsilon, \lambda_{\min} + (t+1)\epsilon]$, $\beta_{(\boldsymbol{X},\boldsymbol{y})}^\epsilon(\lambda) = \lambda \boldsymbol{a}_{P,\epsilon,t} + \boldsymbol{b}_{P,\epsilon,t}$, where $\boldsymbol{a}_{P,\epsilon,t}, \boldsymbol{b}_{P,\epsilon,t}$ is defined as in Theorem D.1. This leads to the piecewise structure of $\mathcal{V}_\epsilon^*$, and we have successfully construct a surrogate function class for $\mathcal{H}^*$ that is: (1) admits piecewise structure, and (2) sufficiently close to $\mathcal{H}^*$.

We can then the define $\mathcal{V}_\epsilon = \{v_{\lambda,\epsilon} : \mathcal{R}_{m,p,m'} \to \mathbb{R} \mid \lambda \in [\lambda_{\min}, \lambda_{\max}]\}$, where $v_{\lambda,\epsilon}(P) := v_{P,\epsilon}^*(\lambda)$.

**Analyzing the pseudo-dimension of $\mathcal{V}_\epsilon$.** We can now proceed to analyze the pseudo-dimension of $\mathcal{V}_\epsilon$ using our proposed results from Section 5.3. First, notice that $f(z) = \frac{1}{m'} \log z$ is a monotonic function. Hence, we can simplify the analysis by analyzing the pseudo-dimension of $\mathcal{G}_\epsilon = \{g_{\lambda,\epsilon} : \mathcal{R}_{m,p,m'} \to \mathbb{R} \mid \lambda \in [\lambda_{\min}, \lambda_{\max}]\}$, where

$$g_{\lambda,\epsilon}(P) = \prod_{i=1}^{m'} (1 + \exp(-y_i x_i^\top \beta_{(\boldsymbol{X},\boldsymbol{y})}^{(\epsilon)}(\lambda))).$$

Again, we can define the dual function class $G_\epsilon^* = \{g_{P,\epsilon}^* : \lambda \to \mathbb{R} \mid P \in \mathcal{R}_{m,p,m'}\}$ of $G_\epsilon$, where $g_{P,\epsilon}^*(\lambda) := g_{\lambda,\epsilon}(P)$.

Fixing a problem instance $P$ and a threshold $\tau \in \mathbb{R}$, by the property of $\beta_{(\boldsymbol{X},\boldsymbol{y})}^{(\epsilon)}(\lambda)$, we know that the domain $[\lambda_{\min}, \lambda_{\max}]$ can be partitioned in to $\frac{1}{\epsilon}|\lambda_{\max} - \lambda_{\min}|$ intervals. In each interval $[\lambda_{\min} + t\epsilon, \lambda_{\min} + (t+1)\epsilon]$, $g_{P,\epsilon}^*(\lambda)$ takes the form

$$g_{P,\epsilon}^*(\lambda) = \prod_{i=1}^{m'} (1 + \exp(-y_i x_i^\top (\lambda a_t + b_t))).$$

Notice that from the problem assumption we have $m' < m$. Therefore, in each interval $[\lambda_{\min} + t\epsilon, \lambda_{\min} + (t+1)\epsilon]$, $g_{P,\epsilon}^*(\lambda)$ is a Pfaffian function of degree at most $m$ from a Pfaffian chain $\mathcal{C}$ of length at most $m$ and of Pfaffian degree at most $m$. From Lemma 5.3, we conclude that $\text{Pdim}(\mathcal{G}_\epsilon) = \mathcal{O}(m^2 + \log(1/\epsilon))$, which implies $\text{Pdim}(\mathcal{V}_\epsilon) = \mathcal{O}(m^2 + \log(1/\epsilon))$.

**Recovering the guarantee for $\mathcal{H}$.** Note that we just establish the upper bound for the learning-theoretic complexity of the surrogate primal function class $\mathcal{V}_\epsilon$. To recover the learning guarantee for $\mathcal{H}$, we need to leverage the approximation error guarantee between $\mathcal{H}$ and $\mathcal{H}^{(\epsilon)}$. Using Theorem 5.4, we then conclude that

$$\mathscr{R}_N(\mathcal{H}) = \mathcal{O}\left( \sqrt{\frac{m^2 + \log(1/\epsilon)}{N}} + \epsilon^2 \right).$$

Finally, classical learning theory results imply the claim. $\qquad\square$

**Remark 5.** Balcan et al. (2023b) introduced the result presented above (Theorem 6.5). For this specific application, the foundation of their approach and our framework (Section 5.3)) are similar. Concretely, our proof makes use of Corollary 5.3, which is similar to the proof in prior work. One can think of the novelty of results presented in Section 5.3 is in obtaining a generalization that is applicable to other problems.

# 7 Online learning

In this section, we introduce new general techniques for establishing no-regret learning guarantees for data-driven algorithms in the online learning setting with full-information[3] feedback when discontinuity of the piecewise Lipschitz dual loss or utility functions admit Pfaffian structure. Prior work (Balcan et al., 2020b) provided a tool for online learning when non-Lipschitzness (or discontinuity) occurs along roots of polynomials in one and two dimensions, and Balcan & Sharma (2021) extended the result to algebraic varieties in higher dimensions. Here, we generalize the results to cases where non-Lipschitzness occurs along Pfaffian hypersurfaces. For convenience and matching the frequently used notations in the online learning literature, we will present our results using *utility functions* instead of *loss functions*.

We note that although the online learning setting is more general than the statistical learning settings (in the sense that guarantees for online learning settings can be automatically translated into guarantees for statistical learning settings via online-to-batch conversion), the results in online learning settings typically require extra smoothness assumption compared to the statistical learning setting, as noted in prior work (Balcan et al., 2024a). Here, we consider the *dispersion property* as a sufficient condition for no-regret online learning in this setting and develop tools to verify whether the dispersion property holds when the discontinuities of the loss sequences involve Pfaffian functions. Roughly speaking, dispersion means that the discontinuities of the dual function do not concentrate in any small region of the parameter space. So, by online-to-batch conversion arguments, our online learning results imply sample complexity guarantees, but only if the distribution satisfies dispersion. In contrast, our sample complexity results in the statistical setting do not make such an assumption, and hold even if the discontinuities concentrate.

## 7.1 Overview of the dispersion property

We first start by giving a high-level overview of the dispersion property, which serves as a sufficient condition for online learning where the sequence of loss (or utility) functions is not continuous but is piecewise Lipschitz. While it is not always possible to obtain sub-linear regret for online learning with piecewise Lipschitz loss functions, Balcan et al. (2018b) introduce the dispersion condition under which online learning is possible. Informally, dispersion measures the number of discontinuities of the loss function that can occur in a ball of a given radius. A sufficient bound on the dispersion can lead to a no-regret learning guarantee. Concretely, we will use the $f$-point-dispersion (Balcan et al., 2020b) which is a slightly more generalized notion.

**Definition 9** ($f$-point-dispersed, Balcan et al., 2020b)**.** The sequence of loss functions $\ell_1, \ell_2, \ldots$ is $f$-point-dispersed for the Lipschitz constant $L$ and dispersion function $f : \mathbb{N} \times [0, \infty) \to \mathbb{R}$ if for all $T$ and for all $\epsilon > 0$, we have

$$\mathbb{E}[\max_{\rho, \rho'} |\{t = 1, \ldots, T : |\ell_t(\rho) - \ell_t(\rho')| > L\|\rho - \rho'\|_2\}|] \leq f(T, \epsilon),$$

where the max is taken over all $\rho, \rho' \in \mathcal{C} : \|\rho - \rho'\|_2 \leq \epsilon$.

The expectation here is over both the randomization of the algorithm[4] (e.g. Theorem 7.6 for logistic regression) for which the performance is measured by the loss functions $\ell_i$ and the "smoothed" adversary that generates the sequence of input instances on which the losses are measured (e.g. Theorem 7.4 for hierarchical clustering). In more detail, there are examples where randomization of the underlying algorithm is sufficient to ensure that Definition 9 is satisfied (Balcan et al., 2018b, Theorem 7.6). However, in other cases, one can establish $f$-point-dispersion by assuming that the input instances in the adversarial sequence are perturbed

---

[3]The loss function for the entire parameter domain is revealed to the learner in each round, after the learner has chosen the parameter for that round.

[4]Algorithm here refers to an algorithm from our parameterized family of algorithms under study, as opposed to the online learning algorithm used to learn its parameters.

slightly using a smoothed analysis (introduced by Spielman & Teng 2003, see Balcan et al., 2018b; 2022a; 2023a for some examples). Intuitively, the adversary can choose an adversarial problem instance, which is then slightly modified to avoid the concentration of a large number of loss functions in the sequence that have Lipschitzness violations (or discontinuities) in arbitrarily small regions of the parameter space. The exact nature of perturbations to the input instance needed to ensure Definition 9 is satisfied depends on the problem instance, e.g. for hierarchical clustering, it is sufficient to assume that pairwise distances between points in the clustering instance follow a bounded-density distribution (Theorem 7.4). In contrast to the statistical setting above, this smoothing does not need to be i.i.d., although in this work we will assume that the perturbations are independent (e.g. Theorem 7.3).

A continuous version of the classical multiplicative weights algorithm achieves the following no-regret guarantee provided the loss functions are dispersed.

**Theorem 7.1** (Balcan et al., 2020b). *Let $\mathcal{C} \subset \mathbb{R}^d$ be contained in a ball of radius $R$ and $\ell_1, \ell_2, \cdots : \mathcal{C} \to [0, 1]$ be piecewise $L$-Lipschitz functions that are $f$-point-dispersed with an $r_0$-interior minimizer. Then, for online learning under full-information feedback, i.e. $\ell_t(\rho)$ is revealed for all $\rho \in \mathcal{C}$ after the learner has picked $\rho_t$ for round $t$, there exists an algorithm that satisfies the regret bound*

$$\mathbb{E}\left[\sum_{t=1}^{T} \ell_t(\rho_t) - \ell_t(\rho^*)\right] \leq \mathcal{O}(\sqrt{dT\log(R/r)} + f(T, r) + TLr),$$

*for any $r \in (0, r_0]$.*

**Contributions.** Though the dispersion property provides a sufficient condition for no-regret learning piecewise Lipschitz loss functions, it is verifying the dispersion property that poses a challenge for Pfaffian structured functions. In this section, we provide a general tool for verifying dispersion property for the sequence of loss functions of which the discontinuities involve in Pfaffian hypersurfaces (Section 7.2). We then demonstrate the applicability of our tool by providing no-regret learning guarantees for under-explored data-driven algorithm design problems in the online learning setting (Section 7.3).

## 7.2 A general tool for verifying dispersion property involving Pfaffian discontinuity

We first introduce a useful notion of shattering which we use to establish a constant upper bound on the VC-dimension of a class of Pfaffian functions with bounded chain length and Pfaffian degree when labeled by axis-aligned segments (i.e., line segments parallel to some coordinate axis). This bound is crucial in extending the framework for establishing dispersion developed in prior work (Balcan et al., 2020b;d; Balcan & Sharma, 2021). We use tools from the theory of Pfaffian functions (included in Appendix E).

**Definition 10** (Hitting and shattering). Consider a Pfaffian chain $\mathcal{C}(\boldsymbol{x}, f_1, \ldots, f_q)$. Let $\mathcal{P} = \{P_1, \ldots, P_K\}$ denote a set of hypersurface, each defined by a Pfaffian function $P(\boldsymbol{x}, f_1(\boldsymbol{x}), \ldots, f_q(\boldsymbol{x})) = 0$ from the Pfaffian chain $\mathcal{C}$. We say that a subset $\mathcal{P}' \subseteq \mathcal{P}$ is hit by a line segment $v$ if, for any $P \in \mathcal{P}$, $v$ intersects with $P$ iff $P \in \mathcal{P}'$. A line $\Upsilon$ hits a subset $\mathcal{P}' \subseteq \mathcal{P}$, there is a line segment $v \in \Upsilon$ such that $v$ hit $\mathcal{P}'$. A collection $\mathcal{V}$ of line segments shatters $\mathcal{P}$ if for each subset $\mathcal{P}' \subseteq \mathbb{R}$, there exists a line segment $\mathcal{V}$ such that $v$ hits $\mathcal{P}'$.

We have the following key structural result which intuitively establishes a bound on the complexity of intersection of Pfaffian hypersurfaces.

**Theorem 7.2.** *There is a constant $K_{M,q,d,p}$ depending only on $M$, $q$, $d$, and $p$ such that axis-aligned line segments in $\mathbb{R}^p$ cannot shatter any collection of $K_{M,q,d,p}$ Pfaffian hypersurfaces consisting of functions from a Pfaffian $\mathcal{C}$ chain of length $q$, degree at most $d$ and Pfaffian degree at most $M$.*

*Proof.* Let $\mathcal{C}(\boldsymbol{x}, f_1, \ldots, f_q)$, where $\boldsymbol{x} \in \mathbb{R}^p$, is a Pfaffian chain of length $q$ and of Pfaffian degree at most $M$. Let $\mathcal{P} = \{P_1, \ldots, P_K\}$ denote a collection of $K$ Pfaffian hypersurfaces $P_i(\boldsymbol{x}, f_1(\boldsymbol{x}), \ldots, f_q(\boldsymbol{x})) = 0$ for $i = 1, \ldots, K$, where each $P_i$ is a Pfaffain function of degree at most $d$ from the Pfaffian chain $\mathcal{C}$. We seek to upper bound the number of subsets of $\mathcal{P}$ which may be hit by axis-aligned line segments. We will first consider shattering by line segments in a fixed axial direction $x = \boldsymbol{x}_j$ for $j \in [p]$.

Let $\Upsilon_j$ be a line along the axial direction $x_j$. The subsets of $\mathcal{P}$ which may be hit by segments along $\Upsilon_j$ are determined by the pattern of intersections of $\Upsilon_j$ with the hypersurfaces in $\mathcal{P}$. We can now use Theorem E.1 to bound the number of intersections between $\Upsilon_j$ and any hypersurface $P_i = 0$ for $P_i \in \mathcal{P}$ as

$$R := 2^{q(q-1)/2}d(M\min\{q,p\}+d)^q,$$

using the fact that the straight line $\Upsilon_j$ is given by $p-1$ equations $x_k = 0$ for $k \neq j$ which form a Pfaffian system with $P_i$ of chain length $q$, Pfaffian degree $M$, and degrees $\deg(P_i) = d$, $\deg(x_k) = 1$ for $k \neq j$. Therefore $\Upsilon_j$ intersects hypersurfaces in $\mathcal{P}$ at most $KR$ points, resulting in at most $KR + 1$ segments of $\Upsilon_j$. Thus, $\Upsilon_j$ may hit at most $\binom{KR+1}{2} = \mathcal{O}(K^2 2^{q(q-1)} d^2 (M\min\{q,p\}+d)^{2q})$ subsets of $\mathcal{P}$. We remark that these upper bounds correspond to an assumption that the Pfaffian hypersurfaces are in a general position, i.e. a small perturbation to the hypersurfaces does not change the number of intersections. Note that there can only be fewer subsets otherwise, so the upper bound still holds.

We will now bound the number of distinct subsets that may be hit as the axis-aligned $\Upsilon_j$ is varied (while it is still along the direction $x_j$). Define the equivalence relation $L_{x_1} \sim L_{x_2}$ if the same sequence of hypersurfaces in $\mathcal{P}$ intersect $L_{x_1}$ and $L_{x_2}$ (in the same order, including with multiplicities). To obtain these equivalence classes, we will project the hypersurfaces in $\mathcal{P}$ onto a hyperplane orthogonal to the $x_j$-direction. By the generalization of the Tarski-Seidenberg theorem to Pfaffian manifolds, we get a collection of semi-Pfaffian sets (van den Dries, 1986), and these sets form a cell complex with at most $\mathcal{O}(K^{(p!)^2(2p(2p+q))^p}(M+d)^{q^{\mathcal{O}(p^3)}})$ cells (Gabrielov & Vorobjov, 2001). This is also a bound on the number of equivalence classes for the relation $\sim$.

Putting together, for each axis-parallel direction $x_j$, the number of distinct subsets of $\mathcal{P}$ hit by any line segment along the direction $x_j$ is at most $\mathcal{O}(K^2 2^{q(q-1)} d^2 (M\min\{q,p\}+d)^{2q} K^{(p!)^2(2p(2p+q))^p}(M+d)^{q^{\mathcal{O}(p^3)}})$. Thus, for all axis-parallel directions, we get that the total number of subsets of $\mathcal{P}$ that may be hit is at most

$$C_K = O\left(p 2^{q(q-1)} d^2 (M\min\{q,p\}+d)^{2q} K^{(p!)^2(2p(2p+q))^p+2}(M+d)^{q^{\mathcal{O}(p^3)}}\right).$$

For fixed $M, q, d, p$, this grows sub-exponentially in $K$, and therefore there is an absolute constant $K_{M,q,d,p}$ such that $C_K < 2^K$ provided $K \geq K_{M,q,d,p}$. This implies that a collection of $K_{M,q,d,p}$ Pfaffian hypersurfaces cannot be shattered by axis-aligned line segments and establishes the desired claim. $\qquad\square$

We can now use the above theorem to establish the following general tool for verifying dispersion of piecewise-Lipschitz functions where the piece boundaries are given by Pfaffian hypersurfaces.

**Theorem 7.3.** *Let $\ell_1, \ldots, \ell_T : \mathbb{R}^p \to \mathbb{R}$ be independent piecewise $L$-Lipschitz functions, each having discontinuities specified by a collection of at most $K$ Pfaffian hypersurfaces of bounded degree $d$, Pfaffian degree $M$ and length of common Pfaffian chain $q$. Let $\mathcal{L}$ denote the set of axis-aligned paths between pairs of points in $\mathbb{R}^p$, and for each $s \in \mathcal{L}$ define $D(T,s) = |\{1 \leq t \leq T \mid \ell_t \text{ has a discontinuity along } s\}|$. Then we have that $\mathbb{E}[\sup_{s \in \mathcal{L}} D(T,s)] \leq \sup_{s \in \mathcal{L}} \mathbb{E}[D(T,s)] + \tilde{O}(\sqrt{T\log T})$, where the soft-O notation suppresses constants in $d, p, M, q$.*

*Proof.* **Technical overview**. We relate the number of ways line segments can label vectors of $K$ Pfaffian hypersurfaces of degree $d$, Pfaffian degree $M$ and common chain length $q$, to the VC-dimension of line segments (when labeling Pfaffian hypersurfaces), which from Theorem 7.2 is constant. To verify dispersion, we need a uniform-convergence bound on the number of Lipschitzness violations between the worst pair of points $\rho, \rho'$ at distance $\leq \epsilon$, but the definition allows us to bound the worst rate of discontinuities along any path between $\rho, \rho'$ of our choice. We can bound the VC-dimension of axis aligned segments against Pfaffian hypersurfaces of bounded complexity, which will allow us to establish dispersion by considering piecewise axis-aligned paths between points $\rho$ and $\rho'$.

**Proof details**. The proof is similar to the analogous results in (Balcan et al., 2020b; Balcan & Sharma, 2021). The main difference is that instead of relating the number of ways intervals can label vectors of discontinuity points to the VC-dimension of intervals, we instead relate the number of ways line segments

can label vectors of $K$ Pfaffian hypersurfaces of bounded complexity to the VC-dimension of line segments (when labeling Pfaffian hypersurfaces), which from Theorem 7.2 is constant. To verify dispersion, we need a uniform-convergence bound on the number of Lipschitz failures between the worst pair of points $\alpha, \alpha'$ at distance $\leq \epsilon$, but the definition allows us to bound the worst rate of discontinuities along any path between $\alpha, \alpha'$ of our choice. We can bound the VC dimension of axis aligned segments against bounded complexity Pfaffian hypersurfaces, which will allow us to establish dispersion by considering piecewise axis-aligned paths between points $\alpha$ and $\alpha'$.

Let $\mathcal{P}$ denote the set of all Pfaffian hypersurfaces of degree $d$, from a Pfaffian chain of length at most $q$ and Pfaffian degree at most $M$. For simplicity, we assume that every function has its discontinuities specified by a collection of exactly $K$ Pfaffian hypersurfaces ($K$ could be an upper bound, and we could simply duplicate hypersurfaces as needed which does not affect our argument below). For each function $\ell_t$, let $\pi^{(t)} \in \mathcal{P}^K$ denote the ordered tuple of Pfaffian hypersurfaces in $\mathcal{P}$ whose entries are the discontinuity locations of $\ell_t$. That is, $\ell_t$ has discontinuities along $(\pi_1^{(t)}, \ldots, \pi_K^{(t)})$, but is otherwise $L$-Lipschitz.

For any axis aligned path $s$, define the function $f_s : \mathcal{P}^K \to \{0, 1\}$ by

$$f_s(\pi) = \begin{cases} 1 & \text{if for some } i \in [K], \pi_i \text{ intersects } s, \\ 0 & \text{otherwise,} \end{cases}$$

where $\pi = (\pi_1, \ldots, \pi_K) \in \mathcal{P}^K$. The sum $\sum_{t=1}^{T} f_s(\pi^{(t)})$ counts the number of vectors $(\pi_1^{(t)}, \ldots, \pi_K^{(t)})$ that intersect $s$ or, equivalently, the number of functions $\ell_1, \ldots, \ell_T$ that are not $L$-Lipschitz on $s$. We will apply VC-dimension uniform convergence arguments to the class $\mathcal{F} = \{f_s : \mathcal{P}^K \to \{0, 1\} \mid s \text{ is an axis-aligned path}\}$. In particular, we will show that for an independent set of vectors $(\pi_1^{(t)}, \ldots, \pi_K^{(t)})$, with high probability we have that $\frac{1}{T} \sum_{t=1}^{T} f_s(\pi^{(t)})$ is close to $\mathbb{E}[\frac{1}{T} \sum_{t=1}^{T} f_s(\pi^{(t)})]$ for all paths $s$.

Now, Theorem 7.2 implies that the VC dimension of $\mathcal{F}$ is at most $K_{M,q,d,p}$. A standard VC-dimension uniform convergence argument for the class $\mathcal{F}$ imply that with probability at least $1 - \delta$, for all $f_s \in \mathcal{F}$

$$\left| \frac{1}{T} \sum_{t=1}^{T} f_s(\pi^{(t)}) - \mathbb{E}\left[ \frac{1}{T} \sum_{t=1}^{T} f_s(\pi^{(t)}) \right] \right| \leq O\left( \sqrt{\frac{K_{M,q,d,p} + \log(1/\delta)}{T}} \right), \text{ or}$$

$$\left| \sum_{t=1}^{T} f_s(\pi^{(t)}) - \mathbb{E}\left[ \sum_{t=1}^{T} f_s(\pi^{(t)}) \right] \right| \leq \tilde{O}\left( \sqrt{T \log(1/\delta)} \right).$$

Now since $D(T, s) = \sum_{t=1}^{T} f_s(\pi^{(t)})$, we have for all $s$ and $\delta$, with probability at least $1 - \delta$, $\sup_{s \in \mathcal{L}} D(T, s) \leq \sup_{s \in \mathcal{L}} \mathbb{E}[D(T, s)] + \tilde{O}(\sqrt{T \log(1/\delta)})$. Taking expectation and setting $\delta = 1/\sqrt{T}$ implies

$$\mathbb{E}[\sup_{s \in \mathcal{L}} D(T, s)] \leq (1 - \delta)\left( \sup_{s \in \mathcal{L}} \mathbb{E}[D(T, s)] + \tilde{O}(\sqrt{T \log(1/\delta)}) \right) + \delta \cdot T$$

$$\leq \sup_{s \in \mathcal{L}} \mathbb{E}[D(T, s)] + \tilde{O}(\sqrt{T \log(\sqrt{T})}) + \sqrt{T},$$

which implies the desired bound. Here we have upper bounded the expected regret by $T$ in the low probability failure event where the uniform convergence does not hold. $\qquad\square$

The above result reduces the problem of verifying dispersion to showing that the expected number of discontinuities $\mathbb{E}[D(T, s)]$ along any axis-aligned path $s$ is $\tilde{O}(\epsilon T)$, which together with Theorem 7.3 implies that the sequence of functions is $f$-point-dispersed with $f(T, \epsilon) = \tilde{O}(\epsilon T + \sqrt{T})$, which in turn implies no-regret online learning (Theorem 7.1).

### 7.3 Applications

We will now instantiate our general online learning results for concrete applications, including linkage clustering and regularized logistic regression. The key new challenge for establishing online learning guarantees is that we need to analyze the relative position of the Pfaffian structured discontinuities (in contrast, in the statistical learning setting, we only cared about the number of induced sign patterns).

### 7.3.1 Online learning for data-driven agglomerative hierarchical clustering

We first show how to analyze the dispersion property of Pfaffian structured loss functions in agglomerative hierarchical clustering. Technical lemmas needed for the proof are deferred to Appendix E.

**Theorem 7.4.** *Consider an adversary choosing a sequence of clustering instances where the $t$-th instance has a symmetric distance matrix $D^{(t)} \in [0, R]^{n \times n}$ and for all $i \leq j, d_{ij}^{(t)}$ follows a distribution of which the probability density function is bounded by some constant $\kappa$. For the sub-family of $\mathcal{M}_1$ with $\alpha > \alpha_{\min}$ for $0 < \alpha_{\min} < \infty$, we have that the corresponding sequence of utility functions $u_1^{(t)}$ are $f$-point-dispersed with $f(T, \epsilon) = \tilde{O}(\epsilon T + \sqrt{T})$.*

*Proof.* **Technical overview**. By using a generalization of the Descartes' rule of signs, we first show that there is at most one positive real solution for the equation

$$(d(a_1, b_1))^\alpha + (d(a_2, b_2))^\alpha = (d(a_1', b_1'))^\alpha + (d(a_2', b_2'))^\alpha.$$

We then use $\kappa$ boundedness of the distances $d_{ij}$ to show that the probability of having a zero $\alpha^* \in I$ in any interval of width $I$ is at most $\tilde{O}(\epsilon)$. Using that we have at most $n^8$ such boundary functions, we can conclude that $\mathbb{E}[D(T, s)] = \tilde{O}(n^8 \epsilon T)$ and use Theorem 7.3.

**Proof details**. As noted in the proof of Theorem 6.1, the boundaries of the piecewise-constant dual utility functions are given by

$$(d(a_1, b_1))^\alpha + (d(a_2, b_2))^\alpha = (d(a_1', b_1'))^\alpha + (d(a_2', b_2'))^\alpha, \tag{1}$$

for some (not necessarily distinct) points $a_1, b_1, a_2, b_2, a_1', b_1', a_2', b_2' \in S$.

We use the generalized Descartes' rule of signs (Jameson, 2006; Haukkanen & Tossavainen, 2011), which implies that the number of real zeros in $\alpha$ (for which boundary condition is satisfied) is at most the number of sign changes when the base of the exponents are arranged in an ascending order (since, the family $\{a^x \mid a \in \mathbb{R}_+\}$ is *Descartes admissible* on $\mathbb{R}$), to conclude that the boundary of the loss function occurs for at most one point $\alpha^* \in \mathbb{R}$ apart from $\alpha = 0$. We consider the following distinct cases (up to symmetry):

- Case $d_\beta(a_1, b_1) \geq d_\beta(a_2, b_2) \geq d_\beta(a_1', b_1') \geq d_\beta(a_2', b_2')$. The number of sign changes is one, and the conclusion is immediate.

- Case $d_\beta(a_1, b_1) \geq d_\beta(a_1', b_1') \geq d_\beta(a_2, b_2) \geq d_\beta(a_2', b_2')$. The only possibility is $d_\beta(a_1, b_1) = d_\beta(a_1', b_1')$ and $d_\beta(a_2, b_2) = d_\beta(a_2', b_2')$. But then, the condition holds for all $\alpha$ and we do not get a critical point. This corresponds to the special case of tie-breaking when merging clusters, and we assume ties are broken following some arbitrary but fixed ordering of the pair of points.

- Case $d_\beta(a_1', b_1') \geq d_\beta(a_1, b_1) \geq d_\beta(a_2, b_2) \geq d_\beta(a_2', b_2')$. The number of sign changes is two. Since $\alpha = 0$ is a solution, there is at most one $\alpha^* \in \mathbb{R} \in \backslash\{0\}$ corresponding to a critical point.

Now, let $\epsilon > 0$. Consider an interval $I = [\alpha_1, \alpha_2]$ with $\alpha_1 > \alpha_{\min}$ such that the width $\alpha_2 - \alpha_1 \leq \epsilon$. If equation (1) has a solution $\alpha^* \in I$ (guaranteed to be unique if it exists), we have that

$$d(a_1, b_1) = \left( (d(a_1', b_1'))^{\alpha^*} + (d(a_2', b_2'))^{\alpha^*} - (d(a_2, b_2))^{\alpha^*} \right)^{1/\alpha^*}$$

Let $p_1$ denote the density of $d(a_1, b_1)$ which is at most $\kappa$ by assumption, and let $f(\alpha^*) := (d(a_1', b_1'))^{\alpha^*} + (d(a_2', b_2'))^{\alpha^*} - (d(a_2, b_2))^{\alpha^*}$ and $d_1(\alpha^*) := (f(\alpha^*))^{1/\alpha^*}$. We seek to upper bound $\text{Pr}_{p_1}\{d_1(\alpha^*) \mid \alpha^* \in I, f(\alpha^*) \geq 0\}$ to get a bound on the probability that there is a critical point in $I$.

Now we consider two cases. If $f(\alpha) \leq \epsilon^\alpha$ for all $\alpha \in I$, we have that $d_1(\alpha) = f(\alpha)^{1/\alpha} \leq \epsilon$ for all $\alpha \in I$. Thus, the probability of having a critical point in $I$ is $\mathcal{O}(\kappa\epsilon)$ in this case.

Else, we have that $f(\alpha^*) > \epsilon^{\alpha^*} > 0$ for some $\alpha^* \in I$. Using Taylor series expansion for $d_1(\alpha)$ around $\alpha^*$, we have

$$d_1(\alpha^* + \epsilon) = d_1(\alpha^*) + d_1'(\alpha^*)\epsilon + \mathcal{O}(\epsilon^2).$$

Thus, the set $\{d_1(\alpha^*) \mid \alpha^* \in I, f(\alpha^*) \geq 0\}$ is contained in an interval of width at most $|d_1'(\alpha^*)|\mathcal{O}(\epsilon)$, implying a upper bound of $\kappa|d_1'(\alpha^*)|\mathcal{O}(\epsilon)$ on the probability of having a critical point in $I$.

Thus it is sufficient to give a bound on $|d_1'(\alpha^*)|$. We have

$$
\begin{aligned}
d_1'(\alpha^*) &= d_1(\alpha^*)\left(-\frac{1}{\alpha^{*2}}\ln f(\alpha^*) + \frac{1}{\alpha^*}\frac{g(\alpha^*)}{f(\alpha^*)}\right) \\
&= \frac{1}{\alpha^*}\left(\frac{g(\alpha^*)}{f(\alpha^*)} - d_1(\alpha^*)\ln d_1(\alpha^*)\right),
\end{aligned}
$$

where

$$g(\alpha) := \ln d(a_1', b_1')(d(a_1', b_1'))^{\alpha^*} + \ln d(a_2', b_2')(d(a_2', b_2'))^{\alpha^*} - \ln d(a_2, b_2)(d(a_2, b_2))^{\alpha^*}.$$

Thus,

$$
\begin{aligned}
|d_1'(\alpha^*)| &= \left|\frac{1}{\alpha^*}\left(\frac{g(\alpha^*)}{f(\alpha^*)} - d_1(\alpha^*)\ln d_1(\alpha^*)\right)\right| \\
&\leq \left|\frac{1}{\alpha^*}\right| \cdot \left(\left|\frac{g(\alpha^*)}{f(\alpha^*)}\right| + |d_1(\alpha^*)\ln d_1(\alpha^*)|\right) \\
&\leq \left|\frac{1}{\alpha_{\min}}\right| \cdot \left(\left|\frac{g(\alpha^*)}{f(\alpha^*)}\right| + R\ln R\right).
\end{aligned}
$$

Now using Lemma E.3, we get that

$$\left|\frac{g(\alpha^*)}{f(\alpha^*)}\right| \leq \frac{1}{\alpha^*} + \ln R,$$

and thus,

$$|d_1'(\alpha^*)| \leq O\left(\frac{R\ln R}{\alpha_{\min}^2}\right).$$

Using that we have at most $n^8$ boundary functions of the form (1), we can conclude that $\mathbb{E}[D(T, s)] = \tilde{O}(\frac{R\ln R}{\alpha_{\min}^2}\kappa n^8\epsilon T)$ and using Theorem 7.3 we can conclude that the sequence of utility functions are $f$-point-dispersed with $f(T, \epsilon) = \tilde{O}(\epsilon T + \sqrt{T})$. $\qquad\square$

We also show how to use our tools above to establish the dispersion property for the $\mathcal{M}_3$ linkage clustering algorithm family.

**Theorem 7.5.** *Consider an adversary choosing a sequence of clustering instances where the $t$-th instance has a symmetric distance matrix $D^{(t)} \in [0, R]^{n \times n}$ and for all $i \leq j, d_{ij}^{(t)}$ follows a distribution of which the probability density function is bounded by some constant $\kappa$. For the family of clustering algorithms $\mathcal{M}_3$, we have that the corresponding sequence of utility functions $u_3^{(t)}$ as a function of the parameter $\alpha = (\alpha_1, \ldots, \alpha_L)$ are $f$-point-dispersed with $f(T, \epsilon) = \tilde{O}(\epsilon T + \sqrt{T})$.*

*Proof.* **Technical overview**. It is sufficient to show dispersion for each $\alpha_i$ keeping the remaining parameters fixed. This is because the definition of dispersion allows us to consider intersections of discontinuities with axis-aligned paths. WLOG, assume $\alpha_2, \ldots, \alpha_L$ are fixed. The boundary functions are given by solutions of exponential equations in $\alpha_1$ of the form

$$\frac{1}{|A||B|} \sum_{a \in A, b \in B} \Pi_{i \in [L]} (d_i(a,b))^{\alpha_i} = \frac{1}{|A'||B'|} \sum_{a' \in A', b' \in B'} \Pi_{i \in [L]} (d_i(a',b'))^{\alpha_i},$$

for $A, B, A', B' \subseteq S$. We can now use Theorem 7.3 together with Theorem 25 of Balcan & Sharma (2021) to conclude that the discontinuities of $u_3^{(t)}$ are $f$-point-dispersed with $f(T, \epsilon) = \tilde{O}(\epsilon T + \sqrt{T})$.

**Proof details**. We will show dispersion for each $\alpha_i$ keeping the remaining parameters fixed. This is sufficient because, the definition of dispersion (Balcan et al., 2020b) allows us to consider discontinuities along axis-aligned paths between pairs of points $\alpha, \alpha'$ in the parameter space. WLOG, assume that $\alpha_2, \ldots, \alpha_L$ are fixed. The boundary functions are given by solutions of exponential equations in $\alpha_1$ of the form

$$\frac{1}{|A||B|} \sum_{a \in A, b \in B} \Pi_{i \in [L]} (d_i(a,b))^{\alpha_i} = \frac{1}{|A'||B'|} \sum_{a' \in A', b' \in B'} \Pi_{i \in [L]} (d_i(a',b'))^{\alpha_i},$$

for $A, B, A', B' \subseteq S$. We can rewrite this equation in the form $\sum_{j=1}^{n} a_j e^{b_j x} = 0$ where, $x = \alpha_1$, $b_j = \ln(d_1(a,b))$ for $a, b \in A \times B$ or $a, b \in A' \times B'$ and $a_j = \frac{1}{|A||B|} \Pi_{i \in \{2,\ldots,L\}} d_i(a,b)^{\alpha_i}$ for $a, b \in A \times B$ or $a_j = \frac{-1}{|A'||B'|} \Pi_{i \in \{2,\ldots,L\}} d_i(a',b')^{\alpha_i}$ for $a', b' \in A' \times B'$. The coefficients $a_j$ are real with magnitude at most $R^{L-1}$. By Lemma E.4, we have that $d_i(a,b)^{\alpha_i}$ is $\kappa'$-bounded for $\kappa' \leq \frac{\kappa}{\alpha_{\min}} \max\{1, R^{\frac{1}{\alpha_{\min}}-1}\}$, and therefore the coefficients $a_j$ have a $\kappa''$-bounded density for $\kappa'' \leq n^2 \kappa'^{L-1} = O\left(n^2 \left(\frac{\kappa R^{1/\alpha_{\min}}}{\alpha_{\min}}\right)^{L-1}\right)$ (using Lemma 8 from Balcan et al. (2018b)). Using Theorem E.5, the probability there is a discontinuity along any segment along the direction $\alpha_1$ of width $\epsilon$ is $p_1 = \tilde{O}(\epsilon)$. Thus, for any axis-aligned path $s$ between points $\alpha, \alpha' \in \mathbb{R}^L$, the expected number of discontinuities for any utility function $u_3^{(t)}$ ($t \in [T]$) is at most $Lp_1 = \tilde{O}(\epsilon)$. We can now apply Theorem 7.3 to get

$$\mathbb{E}[\sup_{s \in \mathcal{L}} D(T,s)] \leq \sup_{s \in \mathcal{L}} \mathbb{E}[D(T,s)] + \tilde{O}(\sqrt{T \log T})$$
$$= \tilde{O}(\epsilon T) + \tilde{O}(\sqrt{T \log T}),$$

where $\mathcal{L}$ denotes the set of axis-aligned paths between pairs of points $\alpha, \alpha' \in \mathbb{R}^L$. It then follows that $\mathbb{E}[\sup_{s \in \mathcal{L}} D(T,s)] = \tilde{O}(\sqrt{T})$ for $\epsilon \geq T^{-1/2}$, establishing that the sequence of utility functions $u_3^{(1)}, \ldots, u_3^{(T)}$ is $f$-point-dispersed with $f(T, \epsilon) = \tilde{O}(\epsilon T + \sqrt{T})$. $\qquad \square$

### 7.3.2 Online learning for data-driven regularized logistic regression

We consider an online learning setting for regularized logistic regression from Section 6.3. The problem instances $P_t = (\boldsymbol{X}^{(t)}, \boldsymbol{y}^{(t)}, \boldsymbol{X}_{\text{val}}^{(t)}, \boldsymbol{y}_{\text{val}}^{(t)})$ are now presented online in a sequence of rounds $t = 1, \ldots, T$, and the online learner predicts the regularization coefficient $\lambda_t$ in each round. The regret of the learner is given by

$$R_T = \mathbb{E}\left[\sum_{t=1}^{T} (h_{\lambda_t}(P_t) - h_{\lambda^*}(P_t))\right],$$

where $\lambda^* = \arg\min_{\lambda \in [\lambda_{\min}, \lambda_{\max}]} \mathbb{E} \sum_{t=1}^{T} h_\lambda(P_t)$. Our main result is to show the existence of a no-regret learner in this setting.

**Theorem 7.6.** *Consider the online learning problem for tuning the logistic regression regularization coefficient $\lambda_t$ stated above. There exists an online learner with expected regret bounded by $\mathcal{O}(\sqrt{T \log[(\lambda_{max} - \lambda_{min})T]})$.*

*Proof.* **Technical overview**. The key idea is to use an appropriate surrogate loss function which well approximates the true loss function, but is dispersed and therefore can be learned online. We then connect the regret of the learner with respect to the surrogate loss with its regret w.r.t. the true loss.

We consider an $\epsilon$-grid of $\lambda$ values. For each round $t$, to construct the surrogate loss function, we sample a uniformly random point from each interval $[\lambda_{\min} + k\epsilon, \lambda_{\min} + (k+1)\epsilon]$ and compute the surrogate loss $h_\lambda^{(\epsilon)}(P)$ at that point, and use a linear interpolation between successive points. By Theorem D.1, this has an error of at most $\mathcal{O}(\epsilon^2)$, which implies the regret gap with the true loss is at most $\mathcal{O}(\epsilon^2 T)$ when using the surrogate function.

We show that the surrogate function is $f$-point-dispersed (Definition 4, Balcan et al. (2020b)) with $f(T, r) = \frac{rT}{\epsilon}$. Using Theorem 5 of Balcan et al. (2020b), we get that there is an online learner with regret $\mathcal{O}(\sqrt{T \log((\lambda_{\max} - \lambda_{\min})/r)} + f(T, r) + Tr + \epsilon^2 T) = \mathcal{O}(\sqrt{T \log((\lambda_{\max} - \lambda_{\min})/r)} + \frac{Tr}{\epsilon} + \epsilon^2 T)$. Setting $\epsilon = T^{-1/4}$ and $r = T^{-3/4}$, we get the claimed regret upper bound.

**Proof details**. We consider an $\epsilon$-grid of $\lambda$ values given by intervals $[\lambda_{\min} + k\epsilon, \lambda_{\min} + (k+1)\epsilon]$ for $k = 0, \ldots, \lfloor \frac{\lambda_{\max} - \lambda_{\min}}{\epsilon} \rfloor$. For each round $t$, to construct the surrogate loss function, we sample a uniformly random point $\lambda_t^{(k)}$ from each interval $[\lambda_{\min} + k\epsilon, \lambda_{\min} + (k+1)\epsilon]$ and compute the surrogate model $\hat{\beta}_{(X,y)}(\lambda)$ at that point using Algorithm 1 (2) for $\ell_1$ ($\ell_2$), and use a linear interpolation between successive points $\lambda_t^{(k)}, \lambda_t^{(k+1)}$ which are at most $2\epsilon$ apart. By Theorem D.1, we have that $\|\hat{\beta}_{(X_t, y_t)}(\lambda) - \beta_{(X_t, y_t)}^{(\epsilon)}(\lambda)\|_2 = \mathcal{O}(\epsilon^2)$ for any $\lambda \in [\lambda_{\min}, \lambda_{\max}]$, where $\beta_{(X_t, y_t)}^{(\epsilon)}(\lambda)$ is the true model (that exactly optimizes the logistic loss) for $(X_t, y_t)$. By Lemma 4.2 of Balcan et al. (2023b) we can conclude that the corresponding surrogate loss $\hat{h}_\lambda^{(\epsilon)}(P_t)$ also satisfies $\|\hat{h}_\lambda^{(\epsilon)}(P_t) - h_\lambda(P_t)\| \leq \mathcal{O}(\epsilon^2)$ for any $\lambda \in [\lambda_{\min}, \lambda_{\max}]$.

This implies the regret with respect to the true loss is at most $\mathcal{O}(\epsilon^2 T)$ more than the regret when using the surrogate function. Suppose $\lambda^* = \arg \min_{\lambda \in [\lambda_{\min}, \lambda_{\max}]} \sum_{t=1}^{T} \mathbb{E}[h_\lambda(P_t)]$. We have,

$$
\begin{aligned}
R_T &= \sum_{t=1}^{T} \mathbb{E}[h_{\lambda_t}(P_t) - h_{\lambda^*}(P_t)] \\
&= \sum_{t=1}^{T} \mathbb{E}[h_{\lambda_t}(P_t) - \hat{h}_{\lambda^*}(P_t)] + \sum_{t=1}^{T} \mathbb{E}[\hat{h}_{\lambda^*}(P_t) - h_{\lambda^*}(P_t)] \\
&\leq \sum_{t=1}^{T} \mathbb{E}[h_{\lambda_t}(P_t) - \hat{h}_{\lambda^*}(P_t)] + \mathcal{O}(\epsilon^2 T) \\
&\leq \sum_{t=1}^{T} \mathbb{E}[h_{\lambda_t}(P_t) - \hat{h}_{\hat{\lambda}}(P_t)] + \mathcal{O}(\epsilon^2 T),
\end{aligned}
$$

where $\hat{\lambda} = \arg \min_{\lambda \in [\lambda_{\min}, \lambda_{\max}]} \sum_{t=1}^{T} \mathbb{E}[\hat{h}_\lambda(P_t)]$.

We next show that the surrogate function is $f$-point-dispersed (Definition 4, Balcan et al. (2020b)) with $f(T, r) = \frac{rT}{\epsilon}$. Indeed, let $0 < r < \epsilon$. Let $I = [\lambda_1, \lambda_2] \subset [\lambda_{\min}, \lambda_{\max}]$ be an interval of length $r$. Then the probability $\hat{h}_\lambda(P_t)$ has a critical point in $I$ is at most $r/\epsilon$.

Using Theorem 5 of Balcan et al. (2020b) combined with the above argument, we now get that there is an online learner with regret $\mathcal{O}(\sqrt{T \log((\lambda_{\max} - \lambda_{\min})/r)} + f(T, r) + Tr + \epsilon^2 T) = \mathcal{O}(\sqrt{T \log((\lambda_{\max} - \lambda_{\min})/r)} + \frac{Tr}{\epsilon} + \epsilon^2 T)$. Setting $\epsilon = T^{-1/4}$ and $r = T^{-3/4}$, we get the claimed upper bound. $\qquad \square$

# 8 Conclusion and future directions

In this work, we introduce the Pfaffian GJ framework for the data-driven statistical learning setting, providing learning guarantees for function classes whose computations involve the broad class of Pfaffian functions. Additionally, we propose the Pfaffian piecewise structure for data-driven algorithm design problems and establish improved learning guarantees for problems admitting such a refined piecewise structure. We apply our framework to three different previously studied data-driven algorithm design problems (including hierarchical clustering, graph-based semi-supervised learning and regularized logistic regression), where known techniques do not yield concrete learning guarantees. By carefully analyzing the underlying Pfaffian structure for these problems, we leverage our proposed framework to establish novel statistical learning guarantees. We further study the data-driven online learning setting, where we introduce a new tool for verifying the dispersion property, applicable when the transition boundaries of the utility functions involve Pfaffian functions.

Our work opens many interesting directions for future research. First, in the statistical learning setting, our work only focuses on providing an upper bound for the learning-theoretic complexity of function classes for which the dual admits the Pfaffian piecewise structure, and establishing lower-bounds remains an important question—prior work (Balcan et al., 2017; Balcan & Sharma, 2021; Balcan et al., 2023b) establishes near-tight lower bounds for simpler algorithm families of clustering, semi-supervised learning and regression, that do not involve Pfaffian functions. It is an interesting question to apply our techniques to other scenarios such as data-driven techniques for solving mixed-integer programs for example learning to cut and learning to branch (Balcan et al., 2018a; 2022c; Cheng & Basu, 2024). Another concrete question is establishing online learning results for clustering families beyond $\mathcal{M}_1$ and for tuning the graph kernel in semi-supervised learning. We expect our general tools here (e.g. Theorem 7.3) to be helpful, but a careful analysis is needed to bound the expected number of discontinuities in any parameter interval. Finally, our framework focuses on establishing statistical sample complexity and no-regret learning guarantees for algorithm configuration for structured Pfaffian settings. Techniques have been developed in prior work for *computationally efficient* configuration of algorithms for clustering and semi-supervised learning (Balcan et al., 2024b; Sharma & Jones, 2023), but only for algorithm families with simpler piecewise structure than those studied here.

# 9 Acknowledgment

This work was supported in part by NSF grants CCF-1910321 and IIS-1901403, the Defense Advanced Research Projects Agency under cooperative agreement HR00112020003, and Simons Investigator Award MPS-SICS-00826333.

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

## Appendix

## A    Preliminaries

In this section, we first recall some additional classical notions and results on learning theory, which is also helpful in our analysis. For completeness, along with the definition of pseudo-dimension (Definition 1), we now give a formal definition of the VC-dimension, which is a counterpart of pseudo-dimension when the function class is binary-valued.

**Definition 11** (Shattering and VC-dimension, Vapnik & Chervonenkis, 1974)**.** Let $\mathcal{U}$ is a class of binary-valued functions that take input from $\mathcal{X}$. Let $S = \{\boldsymbol{x}_1, \ldots, \boldsymbol{x}_m\} \subset \mathcal{X}$ be the set of $m$ input instances, we say that $S$ is shattered by $\mathcal{U}$ if

$$|\{(u(\boldsymbol{x}_1), \ldots, u(\boldsymbol{x}_m)) \mid u \in \mathcal{U}\}| = 2^m.$$

The VC-dimension of $\mathcal{U}$, denoted $\mathrm{VCdim}(\mathcal{U})$, is the largest size of the set $S$ that can be shattered by $\mathcal{U}$.

Rademacher complexity (Bartlett & Mendelson, 2002) is another classical learning-theoretic complexity measure for obtaining data-dependent learning guarantees.

**Definition 12** (Rademacher complexity, Bartlett & Mendelson, 2002)**.** Let $\mathcal{U}$ be a real-valued function class which takes input from domain $\mathcal{U}$. Let $S = \{\boldsymbol{x}_1, \ldots, \boldsymbol{x}_m\} \sim \mathcal{D}$ be a set of $m$ input instances drawn from some problem distribution $\mathcal{D}$ over $\mathcal{X}$. Then the empirical Rademacher complexity $\mathscr{R}_S(\mathcal{U})$ of $\mathcal{U}$ with respect to $\mathcal{U}$ is defined as

$$\hat{\mathscr{R}}_S(\mathcal{U}) = \mathbb{E}_{\boldsymbol{\sigma}} \left[ \sup_{u \in \mathcal{U}} \left| \frac{1}{m} \sum_{i=1}^{m} \sigma_i u(\boldsymbol{x}_i) \right| \right].$$

The Rademacher complexity of $\mathcal{U}$ for $n$ samples is then defined as

$$\mathscr{R}_m(\mathcal{U}) = \mathbb{E}_{S \sim \mathcal{D}^m}[\hat{\mathscr{R}}_S(\mathcal{U})] = \mathbb{E}_{S \sim \mathcal{D}^m} \mathbb{E}_{\boldsymbol{\sigma}} \left[ \sup_{u \in \mathcal{U}} \left| \frac{1}{m} \sum_{i=1}^{m} \sigma_i u(\boldsymbol{x}_i) \right| \right].$$

We now recall the classical notion of uniform convergence.

**Definition 13** (Uniform convergence)**.** Let $\mathcal{U}$ be a real-valued function class which takes input from $\mathcal{X}$, and $\mathcal{D}$ is a distribution over $\mathcal{X}$. If for any $\epsilon > 0$, and any $\delta \in [0, 1]$, there exists $N(\epsilon, \delta)$ s.t. with probability at least $1 - \delta$ over the draw of $m \geq N(\epsilon, \delta)$ samples $\boldsymbol{x}_1, \ldots, \boldsymbol{x}_m \sim \mathcal{D}$, we have

$$\Delta_m = \sup_{u \in \mathcal{U}} \left| \frac{1}{m} \sum_{i=1}^{m} u(\boldsymbol{x}_i) - \mathbb{E}_{\boldsymbol{x} \sim \mathcal{D}}[u(\boldsymbol{x})] \right| \leq \epsilon,$$

then we say that $\mathcal{F}$ uniformly converges.

## B    Omitted proofs from Section 4

In this section, we will present basic properties of Pfaffian chain/functions and a proof for Theorem 4.2.

### B.1    Some facts about Pfaffian functions with elementary operators

In this section, we formalize some basic properties of Pfaffian functions. Essentially, the following results say that if adding/subtracting/multiplying/dividing two Pfaffian functions from the same Pfaffian chain, we end up with a function that is also the Pfaffian function from that Pfaffian chain. Moreover, even if the two Pfaffian functions are not from the same Pfaffian chain, we can still construct a new Pfaffian chain that contains both the functions and the newly computed function as well (by simply combining the two chains).

*Fact* 1 (Addition/Subtraction)**.** Let $g$, $h$ be Pfaffian functions from the Pfaffian chain $\mathcal{C}(\boldsymbol{a}, \eta_1, \ldots, \eta_q)$ ($\boldsymbol{a} \in \mathbb{R}^d$). Then we have $g(\boldsymbol{a}) \pm h(\boldsymbol{a})$ are also Pfaffian functions from the chain $\mathcal{C}$.

*Proof.* For any $a_i$, we have $\frac{\partial}{\partial a_i}(g \pm h)(\boldsymbol{a}) = \frac{\partial}{\partial a_i}g(\boldsymbol{a}) \pm \frac{\partial}{\partial a_i}h(\boldsymbol{a})$. Since $\frac{\partial g(\boldsymbol{a})}{\partial a_i}$ and $\frac{\partial h(\boldsymbol{a})}{\partial a_i}$ are polynomial of $\boldsymbol{a}$ and $\eta_i$, $\frac{\partial}{\partial a_i}(g \pm h)(\boldsymbol{a})$ are also polynomial of $\boldsymbol{a}$ and $\eta_i$, which concludes the proof. $\qquad\square$

*Fact* 2 (Multiplication)*.* Let $g$, $h$ be Pfaffian functions from the Pfaffian chain $\mathcal{C}(\boldsymbol{a}, \eta_1, \ldots, \eta_q)$ ($\boldsymbol{a} \in \mathbb{R}^d$) of length $q$ and Pfaffain degree $M$. Then we have $g(\boldsymbol{a})h(\boldsymbol{a})$ is a Pfaffian function from the chain $\mathcal{C}'(\boldsymbol{a}, \eta_1, \ldots, \eta_q, g, h)$.

*Proof.* For any $a_i$, we have $\frac{\partial}{\partial a_i}(g(\boldsymbol{a})h(\boldsymbol{a})) = g\frac{\partial h(\boldsymbol{a})}{\partial a_i} + h\frac{\partial g(\boldsymbol{a})}{\partial a_i}$, which is a polynomial of $\boldsymbol{a}, \eta_1, \ldots, \eta_q, g, h$. $\qquad\square$

*Fact* 3 (Division)*.* Let $g, h$ be Pfaffain functions from the Pfaffian chain $\mathcal{C}(\boldsymbol{a}, \eta_q, \ldots, \eta_q)$ ($\boldsymbol{a} \in \mathbb{R}^d$) of length $q$ and Pfaffian degree $M$. Assume that $h(\boldsymbol{a}) \neq 0$, then we have $\frac{g(\boldsymbol{a})}{h(\boldsymbol{a})}$ is a Pfaffian function from the Pfaffian chain $\mathcal{C}'(\boldsymbol{a}, \eta_1, \ldots, \eta_q, \frac{1}{h}, \frac{g}{h})$.

*Proof.* For any $a_i$, we have $\frac{\partial}{\partial a_i}\left(\frac{g(\boldsymbol{a})}{h(\boldsymbol{a})}\right) = \frac{\partial g(\boldsymbol{a})}{\partial a_i}\frac{1}{h(\boldsymbol{a})} - \frac{g(\boldsymbol{a})}{h(\boldsymbol{a})}\frac{\partial h(\boldsymbol{a})^2}{\partial a_i}$ is a polynomial of $\boldsymbol{a}, \eta_1, \ldots, \eta_q, \frac{1}{h}, \frac{g}{h}$. $\qquad\square$

*Fact* 4 (Composition)*.* Let $h$ be a Pfaffian function from the Pfaffian chain $\mathcal{C}(\boldsymbol{a}, \eta_1, \ldots, \eta_q)$ ($\boldsymbol{a} \in \mathbb{R}^d$), and $g$ be Pfaffian function from the Pfaffain chain $\mathcal{C}'(b, \eta'_1, \ldots, \eta'_{q'})$ ($b \in \mathbb{R}$). Then $g(h(\boldsymbol{a}))$ is a Pfaffian function from the Pfaffian chain $\mathcal{C}''(\boldsymbol{a}, \eta_1, \ldots, \eta_q, h, \eta'_1(h), \ldots, \eta'_{q'}(h))$.

*Proof.* For any $a_i$, we have $\frac{\partial g(h(\boldsymbol{a}))}{\partial a_i} = \frac{\partial h(\boldsymbol{a})}{\partial a_i}\frac{\partial g(h(\boldsymbol{a}))}{\partial h(\boldsymbol{a})}$. Note that $\frac{\partial g(h(\boldsymbol{a}))}{\partial h(\boldsymbol{a})} = P(h(\boldsymbol{a}), \eta'_1(h(a)), \ldots, \eta'_{q'}(h(\boldsymbol{a}))$, where $P$ is some polynomial. Therefore, $g(h(\boldsymbol{a}))$ is a Pfaffian function from the Pfaffian chain $\mathcal{C}''(\boldsymbol{a}, \eta_1, \ldots, \eta_q, h, \eta'_1(h), \ldots, \eta'_{q'}(h))$. $\qquad\square$

## B.2 Background for proof of Theorem 4.2

In this section, we will present the proof of the Pfaffian GJ algorithm guarantee (Theorem 4.2). To begin with, we first recall some preliminary results about a standard technique for establishing pseudo-dimension upper-bound by analyzing the solution set connected components bound.

### B.2.1 Preliminaries on the connection between pseudo-dimension and solution set connected components bound

We first introduce the notion of a *regular value*. Roughly speaking, given a smooth map $F : \mathbb{R}^d \to \mathbb{R}^r$, where $r \leq d$, a regular value $\epsilon \in \mathbb{R}^r$ is the point in the image space such that the solution of $F(a) = \epsilon$ is well-behaved.

**Definition 14** (Regular value, Milnor & Weaver, 1997)**.** Consider $C^1$ functions $f_1, \ldots, f_r : \mathbb{R}^d \to \mathbb{R}$ where $r \leq d$, $\mathcal{A} \subseteq \mathbb{R}^d$, and the (smooth) mapping $F : \mathcal{A} \to \mathbb{R}^r$ given by $F(\boldsymbol{a}) = (f_1(\boldsymbol{a}), \ldots, f_r(\boldsymbol{a}))$. Then $(\epsilon_1, \ldots, \epsilon_r) \in \mathbb{R}^r$ is a regular value of $F$ if either: (1) $F^{-1}((\epsilon_1, \ldots, \epsilon_r)) = \emptyset$, or (2) $F^{-1}((\epsilon_1, \ldots, \epsilon_r))$ is a $(d - r)$-dimensional sub-manifold of $\mathbb{R}^d$.

It is widely known that by Sard's Lemma (Milnor & Weaver, 1997), the set of non-regular values of the smooth map $F$ has Lebesgue measure 0. Based on this definition, we now present the definition of the *solution set connected components bound*.

**Definition 15** (Solution set connected components bound, Karpinski & Macintyre, 1997)**.** Consider functions $\tau_1, \ldots, \tau_K : \mathcal{X} \times \mathcal{A} \to \mathbb{R}$, where $\mathcal{A} \subseteq \mathbb{R}^d$. Given $\boldsymbol{x}_1, \ldots, \boldsymbol{x}_N$, assume that $\tau_i(\boldsymbol{x}_j, \cdot) : \mathbb{R}^d \to \mathbb{R}$ is a $C^1$ function for $i \in [K]$ and $j \in [N]$. For any $F : \mathbb{R}^d \to \mathbb{R}^r$, where $F(\boldsymbol{a}) = (\Theta_1(\boldsymbol{a}), \ldots, \Theta_r(\boldsymbol{a}))$ with $\Theta_i$ chosen from the $NK$ functions $\tau_i(\boldsymbol{x}_j)$, if the number of connected components of $F^{-1}(\epsilon)$, where $\epsilon$ is a regular value, is upper bounded by $B$ independently on $x_i$, $r$, and $\epsilon$, then we say that $B$ is the solution set connected components bound.

The solution set connected components bound offers an alternative way of analyzing VC-dimension (or pseudo-dimension) of related parameterized function classes, which is formalized in the following lemma Karpinski & Macintyre, 1997. We include a high-level proof sketch for comprehensiveness.

**Lemma B.1** (Karpinski & Macintyre, 1997). *Consider the binary-valued function $\Phi(\boldsymbol{x}, \boldsymbol{a})$, for $\boldsymbol{x} \in \mathcal{X}$ and $\boldsymbol{a} \in \mathbb{R}^d$ constructed using the boolean operators AND and OR, and boolean predicates in one of the two forms "$\tau(\boldsymbol{x}, \boldsymbol{a}) > 0$" or "$\tau(\boldsymbol{x}, \boldsymbol{a}) = 0$". Assume that the function $\tau(\boldsymbol{x}, \boldsymbol{a})$ can be one of at most $K$ forms $(\tau_1, \ldots, \tau_K)$, where $\tau_i(\boldsymbol{x}, \cdot)$ ($i \in [K]$) is a $C^\infty$ function of $\boldsymbol{a}$ for any fixed $x$. Let $\mathscr{C}_\Phi = \{\Phi_{\boldsymbol{a}} : \mathcal{X} \to \{0, 1\} \mid \boldsymbol{x} \in \mathbb{R}^d\}$ where $\Phi_{\boldsymbol{a}} = \Phi(\cdot, \boldsymbol{a})$. Then*

$$\text{VCdim}(\mathscr{C}_\Phi) \leq 2 \log_2 B + (16 + 2 \log K) d,$$

*where $B$ is the solution set connected components bound.*

*Proof Sketch.* The key idea involved in proving the lemma is a combinatorial argument due to Warren (1968) which bounds the number of connected components induced by a collection of boundary functions by $\sum_{j=0}^{d} b_j$, where $b_j$ is the number of connected components induced by intersections of any $j$ functions. This can be combined with the solution set connected components bound $B$, to get a bound $\sum_{j=0}^{d} 2^j \binom{NK}{j} B \leq B \left(\frac{2NKe}{d}\right)^d$ on the total number of connected components. The result follows from noting $2^N \leq B \left(\frac{2NKe}{d}\right)^d$ if the $N$ instances $\boldsymbol{x}_1, \ldots, \boldsymbol{x}_N$ are to be shattered. $\square$

We now recall a result which is especially useful for bounding the solution set connected components bound $B$ for equations related to Pfaffian functions.

**Lemma B.2** (Khovanski, 1991, page 91). *Let $\mathcal{C}$ be a Pfaffian chain of length $q$ and Pfaffian degree $M$, consists of functions $f_1, \ldots, f_q$ in $\boldsymbol{a} \in \mathbb{R}^d$. Consider a non-singular system of equations $\Theta_1(\boldsymbol{a}) = \cdots = \Theta_r(\boldsymbol{a}) = 0$ where $r \leq d$, in which $\Theta_i(\boldsymbol{a})$ ($i \in [r]$) is a polynomial of degree at most $\Delta$ in the variable $\boldsymbol{a}$ and in the Pfaffian functions $f_1, \ldots, f_q$. Then the manifold of dimension $k = d - r$ determined by this system has at most $2^{q(q-1)} \Delta^d S^{d-r} [(r - d + 1)S - (r - d)]^q$ connected components, where $S = r(\Delta - 1) + dM + 1$.*

The following corollary is the direct consequence of Lemma B.2.

**Corollary B.3.** *Consider the setting as in Lemma B.1. Assume that for any fixed $\boldsymbol{x}$, $\tau_i(\boldsymbol{x}, \cdot)$ ($i \in [K]$) is a Pfaffian function of degree at most $\Delta$ from a Pfaffian chain $\mathcal{C}$ with length $q$ and Pfaffian degree $M$. Then*

$$B \leq 2^{dq(dq-1)/2} \Delta^d [(d^2(\Delta + M)]^{dq}.$$

The following lemma gives a connection between the number of sign patterns and number of connected components.

**Lemma B.4** (Section 1.8, Warren, 1968). *Given $N$ real-valued functions $h_1, \ldots, h_N$, the number of distinct sign patterns $\left|\{(sign(h_1(\boldsymbol{a})), \ldots, sign(h_N(\boldsymbol{a}))) \mid \boldsymbol{a} \in \mathbb{R}^d\}\right|$ is upper-bounded by the number of connected components $\mathbb{R}^d - \cup_{i \in [N]} \{\boldsymbol{a} \in \mathbb{R}^d \mid g_i(\boldsymbol{a}) = 0\}$.*

The following result is about the relation between the connected components and the solution set connected components.

**Lemma B.5** (Lemma 7.9, Anthony & Bartlett, 2009). *Let $\{f_1, \ldots, f_N\}$ be a set of differentiable functions that map from $\mathbb{R}^d \to \mathbb{R}$, with regular zero-set intersections. For each $i$, define $Z_i$ the zero-set of $f_i$: $Z_i = \{\boldsymbol{a} \in \mathbb{R}^d : f_i(\boldsymbol{a}) = 0\}$. Then*

$$CC\left(\mathbb{R}^d - \bigcup_{i \in [N]} Z_i\right) \leq \sum_{S \subseteq [N]} CC\left(\bigcap_{i \in S} Z_i\right).$$

*Combining with Definition 15, the RHS becomes $B \sum_{i=0}^{d} \binom{N}{i} \leq B \left(\frac{eN}{d}\right)^d$ for $N \geq d$.*

### B.2.2 Pfaffian formulae

We now introduce the building block of the Pfaffian GJ framework, named *Pfaffian formula*. Roughly speaking, a Pfaffian formula is a boolean formula that incorporates Pfaffian functions.

**Definition 16** (Pfaffian formulae). A Pfaffian formulae $f : \mathbb{R}^d \to \{True, False\}$ is a disjunctive normal form (DNF) formula over boolean predicates of the form $g(\boldsymbol{a}, \eta_1, \ldots, \eta_q) \geq 0$, where $g$ is a Pfaffian function of a Pfaffian chain $\mathcal{C}(\boldsymbol{a}, \eta_1, \ldots, \eta_q)$.

The following lemma essentially claims that for any function class, if its computation can be described by a Pfaffian formula and the corresponding Pfaffian chain exhibits bounded complexities, then the function class also possesses bounded pseudo-dimension.

**Lemma B.6.** *Suppose that each algorithm $L \in \mathcal{L}$ is parameterized by $\boldsymbol{a} \in \mathbb{R}^d$. Suppose that for every $\boldsymbol{x} \in \mathcal{X}$ and $r \in \mathbb{R}$, there is a Pfaffian formula $f_{\boldsymbol{x},r}$ of a Pfaffian chain $\mathcal{C}$ with length $q$ and Pfaffian degree $M$, that given $L \in \mathcal{L}$, check whether $L(\boldsymbol{x}) > r$. Suppose that $f_{\boldsymbol{x},r}$ has at most $K$ distinct Pfaffian functions in its predicates, each of degree at most $\Delta$. Then,*

$$\mathrm{Pdim}(\mathcal{L}) \leq d^2 q^2 + 2dq \log(\Delta + M) + 4dq \log d + 2d \log \Delta K + 16d.$$

*Proof.* This lemma is a direct consequence of Lemma B.1, and Corollary B.3. $\square$

## C   Additional details and omitted proofs for Section 5

In this section, we will present an additional comparison between our proposed framework and the general piecewise structure framework by Balcan et al. (2021a), as well as a detailed proof for Lemma 5.3.

### C.1   A detailed comparison between the piece-wise structure by Balcan et al. (2021a) and our refined Pfaffian piece-wise structure

In this section, we will do a detailed comparison between our refined Pfaffian piece-wise structure, and the piece-wise structure proposed by Balcan et al. (2021a). First, we derive concrete pseudo-dimension bounds implied by Balcan et al. (2021a) when the refined Pfaffian piece-wise structure is present. We note that this implication is not immediate and needs a careful argument to bound the learning-theoretic complexity of Pfaffian piece and boundary functions which appear in their bounds.

**Theorem C.1.** *Consider the utility function class $\mathcal{U} = \{u_{\boldsymbol{a}} : \mathcal{X} \to \mathbb{R} \mid \boldsymbol{a} \in \mathcal{A}\}$, where $\mathcal{A} \subseteq \mathbb{R}^d$. Suppose that $\mathcal{U}^*$ admits $(k_{\mathcal{F}}, k_{\mathcal{G}}, q, M, \Delta, d)$-Pfaffian piece-wise structure. Then using Theorem 5.1 by Balcan et al. (2021a), we have*

$$\mathrm{Pdim}(\mathcal{U}) = \mathcal{O}((d^2 q^2 + dq \log(\Delta + M) + dq \log d + d) \cdot \log[(d^2 q^2 + dq \log(\Delta + M) + dq \log d + d) k_{\mathcal{G}}]).$$

*Proof.* Consider a utility function class $\mathcal{U} = \{u_{\boldsymbol{a}} : \mathcal{X} \to \mathbb{R} \mid \boldsymbol{a} \in \mathcal{A}\}$, where $\mathcal{A} \subseteq \mathbb{R}^d$, we assume that $\mathcal{U}^*$ admits $(k_{\mathcal{F}}, k_{\mathcal{G}}, q, M, \Delta, d)$-Pfaffian piece-wise structure. Then, $\mathcal{U}^*$ also admits $(\mathcal{F}, \mathcal{G}, k_{\mathcal{G}})$ piece-wise structure following Definition 7. Here, $\mathcal{H}$ is the set of Pfaffian functions of degree $\Delta$ from a Pfaffian chain of length $q$ and Pfaffian degree $M$, and $\mathcal{G}$ is the set of threshold functions which are also Pfaffian functions of degree $\Delta$ from the same Pfaffian chain of length $q$ and Pfaffian degree $M$.

The first challenge in using the framework of Balcan et al. is that it only reduces the problem of bounding the learning-theoretic complexity of the piecewise-structured utility function to that of bounding the complexity of the piece and boundary functions involved, which is non-trivial in the case of Pfaffian functions. That is, we still need to bound $\mathrm{Pdim}(\mathcal{F}^*)$ and $\mathrm{VCdim}(\mathcal{G}^*)$, where $\mathcal{F}^*$ and $\mathcal{G}^*$ are dual function classes of $\mathcal{F}$ and $\mathcal{G}$, respectively. To bound $\mathrm{VCdim}(\mathcal{G}^*)$, we first consider the set of $N$ input instances $S = \{g_1, \ldots, g_N\} \subset \mathcal{G}$, and bound the number of distinct sign patterns,

$$\Gamma_S(N) = |\{(g_{\boldsymbol{a}}^*(g_1), \ldots, g_{\boldsymbol{a}}^*(g_N)) \mid \boldsymbol{a} \in \mathcal{A}\}| = |\{(g_1(\boldsymbol{a}), \ldots, g_N(\boldsymbol{a})) \mid \boldsymbol{a} \in \mathcal{A}\}|.$$

From Lemma B.5 and B.4, and B.2, we can bound $\Gamma_S(N)$ as

$$\Gamma_S(N) = \mathcal{O}(2^{dq(dq-1)/2}\Delta^d[(d^2(\Delta + M)])^{dq}\left(\frac{eN}{d}\right)^d.$$

Solving the inequality $2^N = \mathcal{O}(2^{dq(dq-1)/2}\Delta^d[(d^2(\Delta + M)]^{dq}\left(\frac{ek}{d}\right)^d)$, we have $\mathrm{VCdim}(\mathcal{G}^*) \leq \frac{dq(dq-1)}{2} + d\log\Delta + dq\log(d^2(\Delta + M))$.

We now bound $\mathrm{Pdim}(\mathcal{F}^*)$. By definition, we have $\mathcal{F}^* = \{f_{\boldsymbol{a}}^* : \mathcal{F} \to \mathbb{R} \mid \boldsymbol{a} \in \mathcal{A}\}$. Again, to bound $\mathrm{Pdim}(\mathcal{F}^*)$, we first consider the set $S = \{f_1, \ldots, f_N\} \subset \mathcal{F}$ and a set of thresholds $T = \{r_1, \ldots, r_N\}$, and we want to bound the number of distinct sign patterns

$$\begin{aligned}\Gamma_{S,T}(N) &= |\{(\mathrm{sign}(f_{\boldsymbol{a}}^*(f_1) - r_1), \ldots, \mathrm{sign}(f_{\boldsymbol{a}}^*(f_N) - r_N)) \mid \boldsymbol{a} \in \mathcal{A}\}| \\ &= |\{(\mathrm{sign}(f_1(\boldsymbol{a}) - r_1), \ldots, \mathrm{sign}(f_N(\boldsymbol{a}) - r_N)) \mid \boldsymbol{a} \in \mathcal{A}\}|.\end{aligned}$$

Using similar argument as for $\mathcal{G}^*$, we have $\mathrm{Pdim}(\mathcal{F}^*) = \mathcal{O}(d^2q^2 + dq\log(\Delta + M) + dq\log d + d)$. Combining with Theorem 3.3 by Balcan et al. (2021a), we conclude that

$$\mathrm{Pdim}(\mathcal{U}) = \mathcal{O}((d^2q^2 + dq\log(\Delta + M) + dq\log d + d)\log[(d^2q^2 + dq\log(\Delta + M) + dq\log d + d)k_{\mathcal{G}}]).$$

$\square$

**Remark 6.** Compared to our bounds in Theorem 5.2, Theorem C.1 (implied by Balcan et al. (2021a)) has two notable differences. First, our bound is sharper by a logarithmic factor $\mathcal{O}(\log(d^2q^2 + dq\log(\Delta + M) + dq\log d + d))$, which is a consequence of using a sharper form of the Sauer-Shelah lemma. Second, we have a dependence on a logarithmic term $k_{\mathcal{F}}$ which is asymptotically dominated by the other terms, and corresponds to better multiplicative constants than Theorem C.1.

### C.2 Omitted proofs for Section 5.3.1

*Lemma* 5.3 (restated). Consider a function class $\mathcal{V} = \{v_{\boldsymbol{a}} : \mathcal{X} \to \mathbb{R} \mid \boldsymbol{a} \in \mathcal{A}\}$ where $\mathcal{A} \subseteq \mathbb{R}^d$. Assume there is a partition $\mathcal{P} = \{\mathcal{A}_1, \ldots, \mathcal{A}_n\}$ of the parameter space $\mathcal{A}$ such that for any problem instance $\boldsymbol{x} \in \mathcal{X}$, the dual utility function $u_x^*$ is a Pfaffian function of degree at most $\Delta$ in region $\mathcal{A}_i$ from a Pfaffian chain $\mathcal{C}_{\mathcal{A}_i}$ of length at most $q$ and Pfaffian degree $M$. Then the pseudo-dimension of $\mathcal{V}$ is upper-bounded as follows

$$\mathrm{Pdim}(\mathcal{V}) = O(q^2d^2 + qd\log(\Delta + M) + qd\log d + \log n).$$

*Proof.* Consider $N$ problem instances $\boldsymbol{x}_1, \ldots, \boldsymbol{x}_N$ with $N$ corresponding thresholds $\tau_1, \ldots, \tau_N$, we first want to bound the number of distinct sign patterns $\Pi_{\mathcal{V}}(N) = |\{(\mathrm{sign}(v_{\boldsymbol{a}}(\boldsymbol{x}_1) - \tau_1), \ldots, \mathrm{sign}(v_{\boldsymbol{a}}(\boldsymbol{x}_N) - \tau_N)) \mid \boldsymbol{a} \in \mathcal{A}\}|$.

Denote $\Pi_{\mathcal{V}}^{\mathcal{A}_i}(N) = |\{(\mathrm{sign}(v_{\boldsymbol{a}}(\boldsymbol{x}_1) - \tau_1), \ldots, \mathrm{sign}(v_{\boldsymbol{a}}(\boldsymbol{x}_N) - \tau_N)) \mid \boldsymbol{a} \in \mathcal{A}_i\}|$, we have $\Pi_{\mathcal{V}}(N) \leq \sum_{i=1}^n \Pi_{\mathcal{V}}^{\mathcal{A}_i}(N)$. For any $i \in [n]$, from the assumptions, Lemma 5.2 and using Sauer's Lemma, we have

$$\Pi_{\mathcal{V}}(N) \leq \left(\frac{eN}{S}\right)^S,$$

where $S = C(q^2d^2 + qd\log(\Delta + M) + qd\log d)$ for some constant $C$. Therefore $\Pi_{\mathcal{V}}(N) \leq n\left(\frac{eN}{S}\right)^S$. Solving the inequality $2^N \leq \Pi_{\mathcal{V}}(N)$, we conclude that $\mathrm{Pdim}(\mathcal{V}) = O(q^2d^2 + qd\log(\Delta + M) + qd\log d + \log n)$ as expected. $\square$

## D  Additional background and omitted proofs for Section 6

In this section, we will present the deferred proofs as well as additional for various applications in Section 6.

### D.1 Omitted proofs for Section 6.1

*Theorem* 6.2. Let $\mathcal{H}_2$ be a class of functions

$$\mathcal{H}_2 = \{u_2^{\alpha,\beta} \colon (S, \boldsymbol{\delta}) \mapsto u(A_2^{\alpha,\beta}(S, \boldsymbol{\delta})) \mid \alpha \in \mathbb{R} \cup \{-\infty, +\infty\}, \beta \in \Delta([L])\}$$

mapping clustering instances $(S, \boldsymbol{\delta})$ to $[0, 1]$ by using merge functions from class $\mathcal{M}_2$ and an arbitrary merge function. Then $\mathrm{Pdim}(\mathcal{H}_2) = \mathcal{O}(n^4 L^2)$.

*Proof.* Fix the clustering instance $(S, \boldsymbol{\delta})$. Suppose $A, B \subseteq S$ and $A', B' \subseteq S$ are two candidate clusters at some merge step of the algorithm. Then $A, B$ is preferred for merging over $A', B'$ iff

$$\left( \frac{1}{|A||B|} \sum_{a \in A, b \in B} (\delta_\beta(a, b))^\alpha \right)^{1/\alpha} \leq \left( \frac{1}{|A'||B'|} \sum_{a \in A', b \in B'} (\delta_\beta(a, b))^\alpha \right)^{1/\alpha},$$

or equivalently,

$$\frac{1}{|A||B|} \sum_{a \in A, b \in B} (\delta_\beta(a, b))^\alpha \leq \frac{1}{|A'||B'|} \sum_{a \in A', b \in B'} (\delta_\beta(a, b))^\alpha.$$

For distinct choices of the point sets $A, B, A', B'$, we get at most $\binom{2^n}{2}^2 \leq 2^{4n}$ distinct boundary conditions across which the merge decision at any step of the algorithm may change.

We next show that the boundary functions constitute a Pfaffian system in $\alpha, \beta_1, \ldots, \beta_L$ and bound its complexity. For each pair of points $a, b \in S$, define $f_{a,b}(\alpha, \beta) := \frac{1}{\delta_\beta(a,b)}$, $g_{a,b}(\alpha, \beta) := \ln \delta_\beta(a, b)$ and $h_{a,b}(\alpha, \beta) := \delta_\beta(a, b)^\alpha$. Similar to the proof of Theorem 6.1, these functions form a Pfaffian chain $\mathcal{C}(\alpha, \beta, f_{a,b}, g_{a,b}, h_{a,b})$ for $a, b \in S$. We can see that $\mathcal{C}$ is of length $q = 3n^2$ and Pfaffian degree $M = 2$. The boundary conditions can all be written in terms of the functions $\{h_{a,b}\}_{a,b \in S}$, meaning $k_{\mathcal{G}} = n^8$ and degree $\Delta = 1$. Again, note that $k_{\mathcal{F}} = n+1$. Therefore, $\mathcal{H}_2^*$ admits $(n+1, n^8, 3n^2, 2, 1, L+1)$-Pfaffian piece-wise structure. Applying Theorem 5.2 now gives that $\mathrm{Pdim}(\mathcal{H}_2) = \mathcal{O}(n^4 L^2 + n^2 L \log L + L \log 2^{4n}) = \mathcal{O}(n^4 L^2)$. $\qquad\square$

*Theorem* 6.3. Let $\mathcal{H}_3$ be a class of functions

$$\mathcal{H}_3 = \{u_3^\alpha \colon (S, \boldsymbol{\delta}) \mapsto u(A_3^\alpha(S, \boldsymbol{\delta})) \mid \alpha_i \in \mathbb{R} \cup \{-\infty, \infty\} \setminus \{0\}\}$$

mapping clustering instances $(S, \boldsymbol{\delta})$ to $[0, 1]$ by using merge functions from class $\mathcal{M}_3$. Then $\mathrm{Pdim}(\mathcal{H}_3) = \mathcal{O}(n^4 L^2)$.

*Proof.* Fix the clustering instance $(S, \boldsymbol{\delta})$. Suppose $A, B \subseteq S$ and $A', B' \subseteq S$ are two candidate clusters at some merge step of the algorithm. Then $A, B$ is preferred for merging over $A', B'$ iff

$$\left( \frac{1}{|A||B|} \sum_{a \in A, b \in B} \Pi_{i \in [L]} (\delta_i(a, b))^{\alpha_i} \right)^{1/\sum_i \alpha_i} \leq \left( \frac{1}{|A'||B'|} \Pi_{i \in [L]} (\delta_i(a, b))^{\alpha_i} \right)^{1/\sum_i \alpha_i},$$

or equivalently,

$$\frac{1}{|A||B|} \sum_{a \in A, b \in B} \Pi_{i \in [L]} (\delta_i(a, b))^{\alpha_i} \leq \frac{1}{|A'||B'|} \sum_{a \in A', b \in B'} \Pi_{i \in [L]} (\delta_i(a, b))^{\alpha_i}.$$

For distinct choices of the point sets $A, B, A', B'$, we get at most $\binom{2^n}{2}^2 \leq 2^{4n}$ distinct boundary conditions across which the merge decision at any step of the algorithm may change.

We next show that the boundary functions constitute a Pfaffian system in $\alpha_1, \ldots, \alpha_L$ and bound its complexity. For each pair of points $a, b \in S$, define $h_{a,b}(\alpha_1, \ldots, \alpha_L) := \Pi_{i \in [L]} (\delta_i(a, b))^{\alpha_i}$. Note that $\frac{\partial h}{\partial \alpha_i} := \ln \delta_i(a, b) h_{a,b}(\alpha_1, \ldots, \alpha_L)$, and thus these functions form a Pfaffian chain of chain length $n^2$ and Pfaffian degree 1. The boundary conditions can be all written in terms of the functions $\{h_{a,b}\}_{a,b \in S}$, meaning that $k_{\mathcal{G}} = 2^{4n}$, and $\Delta = 1$. Again, note that $k_{\mathcal{F}} = n+1$. Therefore $\mathcal{H}_3^*$ admits $(n+1, 2^{4n}, n^2, 1, 1, L)$-Pfaffian piece-wise structure. Applying Theorem B.6 now gives that $\mathrm{Pdim}(\mathcal{H}_2) = \mathcal{O}(n^4 L^2)$. $\qquad\square$

### D.2 Additional background and omitted proofs for Section 6.3

**Theorem D.1** (Rosset, 2004). *Given a problem instance $P = (X, y, X_{val}, y_{val}) \in \Pi_{m,p}$, for sufficiently small $\epsilon > 0$, if we use Algorithm 1 (2) to approximate the solution $\hat{\beta}_{(X,y)}(\lambda)$ of RLR under $\ell_1$ ($\ell_2$) constraint by $\beta^{(\epsilon)}_{(X,y)}(\lambda)$ then there is a uniform bound $O(\epsilon^2)$ on the error $\|\hat{\beta}_{(X,y)}(\lambda) - \beta^{(\epsilon)}_{(X,y)}(\lambda)\|_2$ for any $\lambda \in [\lambda_{\min}, \lambda_{\max}]$.*

*For any $\lambda \in [\lambda_t, \lambda_{t+1}]$, where $\lambda_k = \lambda_{\min} + k\epsilon$, the approximate solution $\beta^{(\epsilon)}(\lambda)$ is calculated by*

$$\beta^{(\epsilon)}_{(X,y)}(\lambda) = \beta^{(\epsilon)}_t - \left[\nabla^2 l\left(\beta^{(\epsilon)}_t, (X,y)\right)_{\mathcal{A}}\right]^{-1} \cdot \left[\nabla l\left(\beta^{(\epsilon)}_t, (X,y)\right)_{\mathcal{A}} + \lambda \operatorname{sgn}\left(\beta^{(\epsilon)}_t\right)_{\mathcal{A}}\right] = a_t \lambda + b_t,$$

*if we use Algorithm 1 for RLR under $\ell_1$ constraint, or*

$$\beta^{(\epsilon)}_{(X,y)}(\lambda) = \beta^{(\epsilon)}_t - \left[\nabla^2 l\left(\beta^{(\epsilon)}_t, (X,y)\right) + 2\lambda_{t+1}I\right]^{-1} \cdot \left[\nabla l\left(\beta^{(\epsilon)}_t, (X,y)\right) + 2\lambda\beta^{(\epsilon)}_t\right] = a'_t \lambda + b'_t,$$

*if we use Algorithm 2 for RLR under $\ell_2$ constraint.*

---

**Algorithm 1** Approximate incremental quadratic algorithm for RLR with $\ell_1$ penalty, Rosset, 2004

---

Set $\beta^{(\epsilon)}_0 = \hat{\beta}_{(X,y)}(\lambda_{\min})$, $t = 0$, small constant $\delta \in \mathbb{R}_{>0}$, and $\mathcal{A} = \{j \mid [\hat{\beta}_{(X,y)}(\lambda_{\min})]_j \neq 0\}$.
**while** $\lambda_t < \lambda_{\max}$ **do**
$\quad \lambda_{t+1} = \lambda_t + \epsilon$
$\quad \left(\beta^{(\epsilon)}_{t+1}\right)_{\mathcal{A}} = \left(\beta^{(\epsilon)}_t\right)_{\mathcal{A}} - \left[\nabla^2 l\left(\beta^{(\epsilon)}_t, (X,y)\right)_{\mathcal{A}}\right]^{-1} \cdot \left[\nabla l\left(\beta^{(\epsilon)}_t, (X,y)\right)_{\mathcal{A}} + \lambda_{t+1} \operatorname{sgn}\left(\beta^{(\epsilon)}_t\right)_{\mathcal{A}}\right]$
$\quad \left(\beta^{(\epsilon)}_{t+1}\right)_{-\mathcal{A}} = \vec{0}$
$\quad \mathcal{A} = \mathcal{A} \cup \{j \neq \mathcal{A} \mid \nabla l(\beta^{(\epsilon)}_{t+1}, (X,y)) > \lambda_{t+1}\}$
$\quad \mathcal{A} = \mathcal{A} \setminus \{j \in \mathcal{A} \mid \left|\beta^{(\epsilon)}_{t+1,j}\right| < \delta\}$
$\quad t = t + 1$

---

**Algorithm 2** Approximate incremental quadratic algorithm for RLR with $\ell_2$ penalty, Rosset, 2004

---

Set $\beta^{(\epsilon)}_0 = \hat{\beta}_{(X,y)}(\lambda_{\min})$, $t = 0$.
**while** $\lambda_t < \lambda_{\max}$ **do**
$\quad \lambda_{t+1} = \lambda_t + \epsilon$
$\quad \beta^{(\epsilon)}(\lambda) = \beta^{(\epsilon)}_t - \left[\nabla^2 l\left(\beta^{(\epsilon)}_t, (X,y)\right) + 2\lambda_{t+1}I\right]^{-1} \cdot \left[\nabla l\left(\beta^{(\epsilon)}_t, (X,y)\right) + 2\lambda_{t+1}\beta^{(\epsilon)}_t\right]$
$\quad t = t + 1$

---

## E  Additional background and omitted proofs for Section 7

We record here some fundamental results from the theory of Pfaffian functions (also known as Fewnomial theory) which will be needed in establishing our online learning results. The following result is a Pfaffian analogue of Bezout's theorem from algebraic geometry, useful in bounding the multiplicity of intersections of Pfaffian hypersurfaces.

**Theorem E.1** (Khovanski, 1991; Gabrielov, 1995). *Consider a system of equations $g_1(\boldsymbol{x}) = \cdots = g_n(\boldsymbol{x}) = 0$ where $g_i(\boldsymbol{x}) = P_i(\boldsymbol{x}, f_1(\boldsymbol{x}), \ldots, f_q(\boldsymbol{x}))$ is a polynomial of degree at most $d_i$ in $\boldsymbol{x} \in \mathbb{R}^n$ and $f_1, \ldots, f_q$ are a sequence of functions that constitute a Pfaffian chain of length $q$ and Pfaffian degree at most $M$. Then the number of non-degenerate solutions of this system does not exceed*

$$2^{q(q-1)/2} d_1 \ldots d_n (\min\{q, n\}M + d_1 + \cdots + d_n - n + 1)^q.$$

### E.1 Online learning for data-driven agglomerative hierarchical clustering

We will now present some useful lemmas for establishing Theorem 7.4. The following lemma generalizes Lemma 25 of Balcan et al. (2020b).

**Lemma E.2.** *Let $X_1, \ldots, X_n$ be a finite collection of independent random variables each having densities upper bounded by $\kappa$. The random variable $Y = \sum_{i=1}^n \beta_i X_i$ for some fixed scalars $\beta_1, \ldots, \beta_n$ with $\sum_{i=1}^n \beta_i = 1$ has density $f_Y$ satisfying $f_Y(y) \le \kappa$ for all $y$.*

*Proof.* We proceed by induction on $n$. First consider $n = 1$. Clearly $Y = X_1$ and the conclusion follows from the assumption that $X_1$ has density upper bounded by $\kappa$.

Now suppose $n > 1$. We have $Y = \beta_1 X_1 + \beta_2 X_2 + \cdots + \beta_n X_n = \beta_1 X_1 + (1 - \beta_1) X'$, where $X' = \frac{1}{1-\beta_1} \sum_{i=2}^n \beta_i X_i$. By the inductive hypothesis, $X'$ has a density $f_{X'}$ which is upper bounded by $\kappa$. Let $f_{X_1}$ denote the density of $X_1$. Noting $X' = \frac{Y - \beta_1 X_1}{1 - \beta_1}$ and using the independence of $X'$ and $X_1$, we get

$$f_Y(y) = \int_{-\infty}^{\infty} f_{X'}\left(\frac{y - \beta_1 x}{1 - \beta_1}\right) f_{X_1}(x) dx \le \int_{-\infty}^{\infty} \kappa f_{X_1}(x) dx = \kappa.$$

This completes the induction step. $\qquad\square$

We will now present a useful algebraic lemma for establishing Theorem 7.4.

**Lemma E.3.** *If $\alpha > 0$, $a, b, c > 0$, and $a^\alpha + b^\alpha - c^\alpha > 0$, then*

$$\frac{a^\alpha \ln a + b^\alpha \ln b - c^\alpha \ln c}{a^\alpha + b^\alpha - c^\alpha} \le \frac{1}{\alpha} + \ln \max\{a, b, c\}.$$

*Proof.* We consider two cases. First suppose that $c > a$ and $c > b$. We have that

$$\frac{a^\alpha \ln a + b^\alpha \ln b - c^\alpha \ln c}{a^\alpha + b^\alpha - c^\alpha} \le \frac{a^\alpha \ln c + b^\alpha \ln c - c^\alpha \ln c}{a^\alpha + b^\alpha - c^\alpha} \le \ln c$$

in this case.

Now suppose $c \le a$ (the case $c \le b$ is symmetric). Observe that for $\alpha > 0$, the function $f(x) = x^\alpha \ln \frac{K}{x}$ is monotonically increasing for $x \le Ke^{-1/\alpha}$ when $K, \alpha, x > 0$. This implies for $K = \max\{a, b\}e^{1/\alpha}$,

$$c^\alpha \ln \frac{K}{c} \le a^\alpha \ln \frac{K}{a} \le a^\alpha \ln \frac{K}{a} + b^\alpha \ln \frac{K}{b},$$

or, equivalently,

$$a^\alpha \ln a + b^\alpha \ln b - c^\alpha \ln c \le \ln K(a^\alpha + b^\alpha - c^\alpha).$$

Since, $a^\alpha + b^\alpha - c^\alpha > 0$, we further get

$$\frac{a^\alpha \ln a + b^\alpha \ln b - c^\alpha \ln c}{a^\alpha + b^\alpha - c^\alpha} \le \ln K = \frac{1}{\alpha} + \ln \max\{a, b\}.$$

$\qquad\square$

To establish Theorem 7.5, we first present a useful lemma, and restate a useful result from Balcan & Sharma (2021).

**Lemma E.4.** *Suppose $X$ is a real-valued random variable taking values in $[0, M]$ for some $M \in \mathbb{R}^+$ and suppose its probability density is upper-bounded by $\kappa$. Then, $Y = X^\alpha$ for $\alpha \in [\alpha_{\min}, 1]$ for some $\alpha_{\min} > 0$ takes values in $[0, M]$ with a $\kappa'$-bounded density with $\kappa' \leq \frac{\kappa}{\alpha_{\min}} \max\{1, M^{\frac{1}{\alpha_{\min}} - 1}\}$.*

*Proof.* The cumulative density function for $Y$ is given by

$$
F_Y(y) = \Pr[Y \leq y] = \Pr[X \leq y^{1/\alpha}]
$$
$$
= \int_0^{y^{1/\alpha}} f_X(x)dx,
$$

where $f_X(x)$ is the probability density function for $X$. Using Leibniz's rule, we can obtain the density function for $Y$ as

$$
f_Y(y) = \frac{d}{dy} F_Y(y)
$$
$$
= \frac{d}{dy} \int_0^{y^{1/\alpha}} f_X(x)dx
$$
$$
\leq \kappa \frac{d}{dy} y^{1/\alpha}
$$
$$
= \frac{\kappa}{\alpha} y^{\frac{1}{\alpha} - 1}.
$$

Now for $y \leq 1$, $y^{\frac{1}{\alpha} - 1} \leq y \leq 1$ and therefore $f_Y(y) \leq \frac{\kappa}{\alpha_{\min}}$. Else, $y \leq M$, and $f_Y(y) \leq \frac{\kappa}{\alpha_{\min}} M^{\frac{1}{\alpha_{\min}} - 1}$. $\square$

We will also need the following result due to Balcan & Sharma (2021) which is useful to establish dispersion when the discontinuities of the loss function are given by roots of an exponential equation in the parameter with random coefficients. We refer the readers to the original paper for a proof.

**Theorem E.5** (Balcan & Sharma, 2021). *Let $\phi(x) = \sum_{i=1}^n a_i e^{b_i x}$ be a random function, such that coefficients $a_i$ are real and of magnitude at most $R$, and distributed with joint density at most $\kappa$. Then for any interval $I$ of width at most $\epsilon$, $\Pr(\phi$ has a zero in $I) \leq \tilde{O}(\epsilon)$ (dependence on $b_i, n, \kappa, R$ suppressed).*

