# OpenReview forum: "Algorithm Configuration for Structured Pfaffian Settings"
_TMLR — Accepted by TMLR_

### Review · Reviewer_XaQa · 2024-12-17

**Summary Of Contributions:**

The paper belongs to the line of work on learning theory for data-driven algorithms. Gupta and Roughgarden (2016) initiated the study of proving VC dimension bounds for parameterized families of algorithms, or more precisely, for the classes of loss functions corresponding to those algorithms. These bounds yield generalization bounds for selecting approximately optimal parameters from a bounded number of samples drawn from an application-specific distribution. Balcan et al. (2018) extended this to the online learning setting, where the goal is to select parameters with vanishing regret compared to the optimal parameters in hindsight.

So far, there have been two general approaches for proving VC dimension bounds for data-driven algorithms. One by Balcan et al. (2021) relies on piecewise decomposition of the dual function classes of losses. It is very general, albeit it requires the non-trivial step of bounding the VC dimension of the classes arising in the piecewise decomposition of the specific task in question. Furthermore, due to its generality, it falls short of fully exploiting the problem structure in some cases, leading to suboptimal or vacuous bounds. The other approach by Bartlett et al. (2022) relies on a refinement of the GJ algorithm framework of Goldbert and Jerrum (1993). It leads to sharper bounds in some cases, albeit it is limited in applicability to cases where the class of losses can be expressed or approximated by rational functions of the parameters.

This paper extends these approaches to the case of Pfaffian functions (Khovanskii 1991). Pfaffian functions are a much more general class of functions than rational functions handled in Bartlett et al. (2022), and yet usefully more specific than the generic piecewise decomposition from Balcan et al. (2021). Thus this approach addresses cases not handled by previous approaches, and substantially expands the scope of algorithms for which VC dimension bounds are achievable. Much like Karpinski and Macintyre (1997) extended the GJ framework to Pfaffian functions in order to prove VC dimensions for a broader class of neural network activations, this paper takes the natural step of carrying out this extension for data-driven algorithms. It extends the GJ algorithms framework from Bartlett et al. (2022) to Pfaffian GJ algorithms, and the piecewise decomposition framework from Balcan et al. (2021) to Pfaffian piecewise functions. It then uses these general results to prove generalization bounds for families of parameterized algorithms that were either suboptimally handled or not handled by previous approaches. It also shows how to apply these techniques in the online learning setting.

**Audience:**

Yes

**Claims And Evidence:**

Yes

**Requested Changes:**

Possible technical issues -- please explain and/or correct.

This is inevitably a partial list, so as written above, please make a thorough technical correction pass over the manuscript.

Theorem 4.2: Clarification needed about the degrees.
1. Could you specify the precise definition of "degree" $\Delta$ of an intermediate value? Is it the polynomial for a Pfaffian operator step or the rational degree for an arithmetic operator step or what exactly?
2. In the statement of the theorem, $M$ is the Pfaffian degree of the chain and $\Delta$ is the degree of intermediate values. But in the proof, $\Delta$ is the Pfaffian degree of the chain. Shouldn't this be $M$? Does this change the claimed bound on the pseudo-dimension in the theorem?
3. In Lemma B.2, could you please point to the specific result within (Khovanskii 1991) that you cite? This is a monograph-length work. Also, little $m$ in the $\Delta^m$ term is undefined -- presumably a typo.
4. In Corollary B.3, $\Delta$ appears in the final bound but nowhere before. What is it here? Is it the polynomial degree of the $\tau_i$'s?

All these issues make it  very hard to properly read and verify the theorem.

Lemma 5.4:
1. What is "messy $G$" in the RHS? Presumably a typo "messy $R$"?
2. Did you write "$\mathbf{x}i_i$" instead of $\xi_i$? This is quite confusing.

Lemma 5.6:

Is $H$ (the upper bound on the utility range) missing from the lemma's conclusion?

Theorem 6.5:
1. What is going on with $m$? Did you use the same letter in the same theorem statement for both the matrix dimension and the number of samples?
2. $\epsilon$ appears in the final bound but nowhere before in the theorem statement, what is it here? Is it any $\epsilon>0$, according to the proof?

Theorem A.1:

It looks you mistakenly switched between then $N(\epsilon,\delta)$ and the $N(\epsilon/2,\delta)$

No future directions mentioned under "future directions"

**Strengths And Weaknesses:**

The strength of this paper is in applying a strong and appropriate technique to substantially advance the line of work of generalization bounds for data-driven algorithm design. The conceptual and technical content, and the results it yields, are sufficiently strong to merit acceptance for publication.

The main weakness of this manuscript is that it seems quite careless and haphazard in the technical details. Many seem glossed over, not properly written formally, and possibly imprecise. I stress that I don't doubt that the overall approach is sound, given its history in the related literature and its presentation in this paper. I do have doubts about the level of formality in some parts, which in turn lead to doubts about some of the precise bounds stated, especially those stated with explicit rather than hidden constants. These are fixable issues and I wouldn't want to see a significant paper rejected because of them, but since this is an archival submission in a journal that emphasizes methodological soundness, I feel the authors should thoroughly clean up the manuscript of such formal issues and gaps. I detail some of these issues below under requested changes, though inevitably I was unable to list all of them.

Apart from the technical content, the writing itself evidently had not undergone proper proofreading prior to submission, in terms of linguistic errors and misuse of citation tags (citep/citet), and these would need to be cleaned up as well to make the paper more readable. There are too many instances of this for me to list, so I give up trying altogether.

Perhaps the main weakness at the technical level is the somewhat notorious $q^2$ dependence incurred by the technique of (Khovanskii 1991) in the Pfaffian chain length $q$. In the application to data-driven algorithms given in this paper, this leads to VC dimension bounds with quadratic dependence on the number of parameters, which is likely loose as one typically expects it to be near-linear (there is no discussion of lower bounds in the paper). But since this dependence is a long-standing issue, and that no prior bounds were known for the problems considered in this paper, this is far from an argument against the paper.

---

> ### Author Response · Authors · 2025-01-16
> **Author response**
>
> We thank the reviewer for thoroughly reviewing our paper and providing constructive feedback. We are glad that the reviewer acknowledges our theoretical contributions and finds our work interesting. We have addressed the reviewer’s comments and proofread the paper, making several updates as noted below.
>
> ## Changes made
> We made several careful passes through the paper trying to catch typos that were not caught at the submission time. We also made the following changes in the draft:
> - Added an example of a problem instance in the Preliminaries section (page 5) to make it easier for the reader to follow the setting.
> - Added references in which naively using the general framework by Balcan et al. (STOC’21) will lead to loose or vacuous bounds (page 10).
> - Removed a sentence from the first paragraph of page 16 which is potentially confusing.
> - Elaborated on the definition of kappa-bounded density in the statements of Theorem 7.4, 7.5 (page 27, 29).
>
> Additionally, we have rephrased the text in several other places to improve the flow and presentation of the draft. __To make it easier for the reviewer to track the changes, we have presently highlighted the changes in red.__
>
> ## On the reviewer’s questions/concerns
>
> 1. “_Potential misuses of citet/citep_”: Thank you for your suggestion. We have carefully reviewed the manuscript again and ensured that citet/citep/citealt/citealp were properly used.
>
> 2. In Theorem 4.2.
>
>      a. "_On the definition of $\Delta$_": thank you for this clarifying question. Note that any intermediate value computed is a Pfaffian function from the Pfaffian chain $C_{\boldsymbol{x}, r}(a, f_1, \dots, f_q)$. Here, the degree $\Delta$ of the intermediate value $v$ is the degree of the Pfaffian function representation, that is, if $v$ is represented as $v(a) = Q(a, f_1, …, f_q)$ where $Q$ is a polynomial in $a$, $f_1, \dots, f_q$ of degree $\Delta$, then $v(a)$ has degree $\Delta$. To make this point clear, we added more details in Definition 4 (Pfaffian functions), as well as additional clarification in Examples 1 and 2.
>
>      b. "_Potential typos in the proof; should be $M$ instead of $\Delta$_": thank you for pointing this out. It is a typo in the 3rd paragraph of the proof, and it should be $M$ instead of $\Delta$. We have fixed this in the revised draft. Note that this minor typo does not affect the claim and other parts of the proof.
>
>      c. "_The precise reference for Lemma B.2 (within Fewnomials by Khovanskii); what is $m$ in $\Delta^m$_": The precise reference of our Lemma B.2 is the Example of Estimates 2 (an example of Corollary 2, page 90, Fewnomials), located on page 91 of Fewnomials. Besides, it is true that there is a typo in $\Delta^m$, which should be $\Delta^d$ instead (where $d$ is the dimensionality of $\boldsymbol{a}$). We have incorporated this precise reference and fixed the typo in the revised draft.
>
>      d. "_Is $\Delta$ in Corollary B.3 the polynomial degree of \tau_{i}_": It is true that $\Delta$ here is the maximum degree of $\tau_i$. We have incorporated this in the revised draft.
>
>
> 3. "_Potential typos in Lemma 5.4; $xi_i, \mathscr{G}$_": thank you for pointing it out. These are typos, and should be $\xi_i, \mathscr{R}$ instead. We have fixed this in the revised draft.
>
>
> 4. "_In Lemma 5.6, is $H$ missing from the Lemma conclusion?_": Thank you for your careful observation. We have fixed this issue in the revised draft.
>
>
> 5. In Theorem 6.5,
>
>       a. “_It seems that the notation $m$ is duplicated ”: Thank you for pointing this out. It is true that the notation $m$ is duplicated here: one is for the number of problem instances used in the generalization bound, and one is for the set of problem instances $\mathcal{R}_{m, p, m’}$.  To fix this issue and improve the presentation, we have made a minor revision to the notation in Section 6.3. The changes are reflected in the revised draft.
>
>       b. “_What is $\epsilon$, is it any $\epsilon > 0$, according to the proof_”: Thank you for pointing this out. Here, $\epsilon$ has to be sufficiently small (satisfying Theorem D.1.). We have modified the statement of Lemma 6.5 and its proof to emphasize this point.
>
>
> 6. “_In Theorem A.1: Did you mistakenly switch from $N(\epsilon/2, \delta)$ to $N(\epsilon, \delta)$_”: Thank you for pointing out the typo in the classical result Theorem A.1, included for completeness. We have fixed it in the revised draft.
>
>
> 7. “_No future direction in the future direction section._”: thank you for pointing this out. We have reincorporated this part in the revised draft.
>
>
> ## Summary
>
> Overall, we thank the reviewer again for carefully reading our paper and providing detailed constructive feedback. As the reviewer suggested, we incorporated those points in the revised draft (marked in red) to improve the clarity and presentation of the paper. We hope that our answers address the reviewer’s concerns, and we are happy to address any further questions by the reviewer.

---

### Review · Reviewer_EpQD · 2025-02-18

**Summary Of Contributions:**

This paper studies data-driven algorithm design in both stochastic and online learning setting. For the stochastic setting, the authors proposed the Pfaffian GJ framework, which can provide Pseudo dimension bounds for a broad class of utility functions (i.e., Pfaffian functions). It include important functions such as the exponential function. The proposed framework is more general compared to the  previous GJ framework as it includes Pfaffian functions. The authors also provide several applications for their proposed framework, some of them show that the proposed framework can lead to similar guarantees as previous work under milder conditions. For the online learning setting, the authors propose a novel way of varifying the dispersion property of Pfaffian functions.

**Audience:**

Yes

**Claims And Evidence:**

Yes

**Requested Changes:**

Please see the weaknesses part of the section above.

**Strengths And Weaknesses:**

Weaknesses:

I have the following comments/questions:

Definition 5: I don’t understand Definition 5. The first sentence says the inputs of algorithm is a \in R^d, which is a vector, but in the operation discerption, v seems to be a real value (e.g., the Pfaffian operators). Is v supposed to be a? If it is not a, please give the definition of v and also its domain.


“Therefore, we can use the established results for the Pfaffian GJ algorithm (Theorem 5) to derive learning guarantees for such problems.”is it supposed to be Theorem 4.2 (I don’t see Theorem 5 in the paper)?

Lemma 5.4:
1)	R shoule be \mathbb{R}.
2)	Should it be S\subseteq X^{N} not S\subseteq X?
3)	Finally, in the last equation, xi_i should be xi_N.

Lemma 5.6: We are talking about utility here not loss in this section, so I think hat{a} is defined as argmin, not argmax?

Mismatch of the notations: before Section 6, the authors talk about maximizing utility, while after section 6 it is switched to minimizing the loss, which is confusing. I suggest the authors unity the notation.

Figure 2: mathbb{1} should be mathbb{I}; Moreover, how is P_{b,j} defined? Note that be is a vector, not a real value.

I don’t fully understand Remark 5. I wonder if the authors could discuss more about: what is the benefit of applying the proposed framework for this problem, compared to Balcan et al. (2023b)? I think it is an important question since it is the major application of the results given in Section 5.3.

Definition 9: Why there is an expectation, and which element is stochastic? If ell_ts are drawn from some distribution in this online learning setting, please add more discussion about this. Since the stochastic setting is already discussed before, I thought here the losses should be adversarial. I understand that there might be an expectation in Theorem 7.1 as the algorithm might be stochastic, but I don’t understand Definition 9. Similarly, I am not sure about the proof of Theorem 7.2. Is there an implicit assumption that all loss are iid?

What is M in Theorem 7.1?

Theorem 7.3: l should be ell.

Theorem 7.3: Here L has two definitions: one the Lipschitz constant, and the other is the set of axis-aligned paths between pairs of points. The second one should use a different notation.

Theorem 7.6: there are missing [ ] and ( ).


Strengths:

Significance and Novelty:

1. A more generalized analysis framework. The proposed Pfaffian GJ framework generalizes the GJ framework studied by previous work (Bartlett et al. (2022)). , which can involve more complicated functions, such as exponential and logarithmic functions. However, note that it seems that logarithmic functions (logarithmic regression) has been considered by previous work (Balcan et al. (2023b)). The proposed framework can recover pervious case as a special case.

The authors also provide several new applications of there framework. For example, in data-driven agglomerative hierarchical clustering, the proposed methods is applible under much more general conditions. It broadens the application domain of  data-driven algorithms and analysis.


2. For the online learning setting, the proposed efficient ways to verify the dispersion property of a border class of loss functions (i.e.,  Pfaffian functions). The authors also show that it is applicable to many real world problems.

3. Presentation: Except from the comments in the beginning, I think the authors did a good job on introducing the Pfaffian function related ideas. I also appreciate it that the authors clearly written down the connection to previous work.

---

> ### Author Response · Authors · 2025-02-24
> **Author response**
>
> ## Answers and changes made
>
> We thank the reviewer for carefully reviewing our paper and providing constructive feedback. We are glad that the reviewer acknowledges the novelty and significance of our theoretical contribution. We will address the reviewer’s comments/questions as follows (changes are marked in **blue** in the revised draft):
>
>   1. _Clarification on Definition 5_: In Definition 5, $v$ corresponds to either a component of the input vector $\boldsymbol{a}$ or an intermediate value computed in one of the previous steps. Indeed, $v$ is a real-valued number.
>
>  2. _Reference typos: "Theorem 5" mentioned in the draft was supposed to be Theorem 4.2_: thank you for pointing it out, we fixed it in the revised draft.
>
>  3. _In Lemma 5.4_:
>
>      a. _Should $\mathcal{R}$ be $\mathbb{R}$_: Actually, $\mathcal{R}$ represents the parameterization set of the function classes $\mathcal{F}$ and $\mathcal{G}$, which can be any index set. However, to unify the notations, we write explicitly by replacing $\mathcal{R}$ with $\mathcal{A} \subseteq \mathbb{R}^d$. We made these changes in the revised draft.
>
>     b. _Should $S \subset \mathcal{X}$ be $S \subset \mathcal{X}^N$_: Here, $S = \\{\boldsymbol{x_1}, \dots, \boldsymbol{x_N}\\}$ is a set of problem instances, so it should be $X \subset \mathcal{X}^N$. However, we can write $S \in \mathcal{X}^N$ to imply that $S$ can contain duplicated problem instances, as suggested by the reviewer. We updated the revised draft accordingly.
>
>     c. It should be $\mathbb{E}_{\xi_1, \dots, \xi_N}$: we fixed that in the revised draft.
>
> 4. _Typos in Lemma 5.6_: it is true that $\arg\max$ should be used here. We fixed it in the revised draft.
>
> 5. _Notation mismatch before and after Section 6_: Thank you for pointing it out. However, we note that the result we presented here can still be applicable to Section 6 since we can simply define the new utility function as $H$ minus the loss function. Nevertheless, we agree that synchronizing notations helps the presentation of the paper, and we have updated the draft accordingly.
>
>  6. _Potential typos in Figure 2_: thank you for pointing it out, we have fixed it in the revised draft.
>
>  7. *How $P_{\\mathbf{b}, j}$ is defined in Figure 2, elaborate on $\\mathbf{b}$*: In this figure, we are demonstrating that both piece function $f_{\\boldsymbol{x}, \\mathbf{b}}$ and the boundary function $g^{(i)}_{\\boldsymbol{x}}$ are part of the same chain.
>
>        It means that $\\frac{\\partial f_{x, \\mathbf{b}}}{\\partial a_j}$ is some polynomial $P_{\\mathbf{b}, j}$ of $\\boldsymbol{a}, \\eta_1, \\dots, \\eta_q$. Here, $\\mathbf{b}$ is just an index (binary) vector, encoding the position of the piece function w.r.t. to the boundaries $g^{(i)}_{\\boldsymbol{x}}$.
>
>   8. *On Remark 5*: Thank you for the question. Our approach recovers the result of Balcan et al. (2023b) for tuning the reguarlization parameter in logistic regression. Our framework is an alternative technique to obtain matching bounds of for the problem, and we do not claim any additional improvements here either in Section 5.3, or in our bulleted list of contributions
>
>  9. *Further elaborate on Definition 9*: Here, we are considering the setting by Balcan et al. (2020), where the sequence of loss functions can be thought of as being generated by a "smoothed" adversary. The notion of smoothness comes from the celebrated literature of smoothed analysis (Spielman and Teng, 2023), where the adversarial instance is perturbed slightly using a bounded density distribution (need not be i.i.d, although we assume independence for simplicity e.g. in Theorem 7.3). The exact nature of this smoothness needed to guarantee dispersion varies from problem to problem, although there are known examples where the stochasticity of the online algorithm is sufficient to guarantee dispersion (Balcan et al., 2018). We have added a paragraph summarizing this discussion after Definition 9.
>
>
>  10. *What is M in Theorem 7.1*: $M$ is the size of a partition of the hyperparameter space, such that the loss function in each online round is only observed for the piece in the partition where the learner played her point (before observing the loss), corresponding to the semi-bandit feedback system of Balcan et al. (2020b). For a simplified exposition of our results, we only need to case $M = 1$ (corresponding to full-feedback, although our dispersion-based results should extend to any $M$). So we have simplified the theorem statement and explicitly clarified this point.
>
>   11. *Potential typos and notation overloading in Theorem 7.3*. Thank you for pointing that out. We have fixed this in the revised draft.
>
> ## Summary
>
> Overall, we thank the reviewer again for carefully reading our paper and providing detailed constructive feedback. We incorporated the changes suggested by the reviewer in **blue**. We hope that our answers address the reviewer's concerns, and we are happy to address any further questions.

---

> ### Comment · Reviewer_EpQD · 2025-02-25
> **Response**
>
> I would like thank the authors for the detailed feedback. I do not have further questions/comments, except for the following comments on Definition 9 and the related assumptions.  I would like to note that these comments are only minor suggestions for the paper, not major concerns:
>
> 1. In Definition 9, there is only the "smoothed adversary", and there is no learner (since it is for all $\rho, \rho'$), so I am not sure why " randomization of the learner is sufficient to ensure that Definition 9 is satisfied" (it is written in the comments in blue blow Definition 9). I wonder if you mean randomization of the adversary?
>
> 2. For the same reason, I am not sure if "The expectation here is over both the randomization of the learner and the “smoothed” adversary" makes sense for Definition 9. I think "The expectation wrt the randomization of the learner" only happens for e.g., Theorem 7.1, not Definition 9.
>
> 3. It would be great if the "smoothed adversary" can be defined in a more formal way, not being described by words.

---

> > ### Author Response · Authors · 2025-02-27
> > **Response to Reviewer EpQD**
> >
> > We thank the reviewer for the great question. We have edited our paragraph after Definition 9, taking into account these questions, and provided further clarification below.
> >
> > Recall that our applications are to algorithms for learning problems (clustering, regression, etc.), and the losses in the online learning setting measure the loss (as a function of the algorithm’s parameters) of the algorithm for the given problem (e.g. clustering algorithm) on an online sequence of input instances generated by the adversary. So, in addition to the randomization involved in smoothing the input instances generated by the adversary, there may be some randomization in the learning algorithm itself (this is different from the online learning algorithm used to learn the parameters of the learning algorithm). The reviewer has raised an important point: We clearly need to distinguish the two algorithms, the one for the given problem we are trying to solve and the online procedure for doing the algorithm selection, and we have updated the draft to reflect this.
> >
> > Regarding a formal example of smoothing, we note that the exact nature of smoothing needed on the input instances generated by the adversary in order to satisfy Definition 9 varies with the algorithm family under study. For the hierarchical clustering algorithm family, it is sufficient to add a smooth (e.g., small Gaussian) noise to the points being clustered (formally, the pairwise distances between points being clustered are assumed to follow a bounded density distribution in Theorem 7.4).

---

### Review · Reviewer_BX7E · 2025-02-25

**Summary Of Contributions:**

This work provides learning guarantees for parameterized data-driven algorithm
problems in distributional and online settings (i.e., how to learn good
hyperparameters for general algorithms based on a historical sequence of
problem instances).
It generalizes many prior works of Balcan (10+ citations) to _Pfaffian functions_ (i.e., an extension of the class of rational functions that includes exponential and logarithmic function calls).
Specifically, it shows that if dual utility function class $\mathcal{U}^*$ admits Pfaffian piecewise structure,
then we can recover distributional and learning gaurantees for the given utility function $\mathcal{U}$.

There are three main contributions:

**Distributional learning** (Section 5)

  - Extends the Goldberg-Jerrum (GJ) framework [Goldberg-Jerrum, COLT 1993] from rational functions to Pfaffian functions
  - The main learning guarantee result is Lemma 5.6
  - The main technical building block is Lemma 5.3 (Appendix C), which is a refined analysis of [Balcan-DeBlasio-Dick-Kingsford-Sandholm-Vitercik, STOC 2021]

**Online learning** (Section 7)

  - Provides a new method for verifying the _dispersion property_ of a sequence of loss functions with Pfaffian hypersurfaces, which is a sufficient condition for obtaining regret bounds
  - Studies the same set of applications in the distributional learning section

**Applications** (Section 6 and 7.3)

- The authors present several practical applications and show how these problems fit into their framework to inherit its guarantees:
  * Agglomerative hierarchical clustering
  * Graph-based semi-supervised learning with Gaussian radius basis function kernel
  * Regularized logistic regression (i.e., learning the best regularization strength to use)

**Audience:**

Yes

**Claims And Evidence:**

Yes

**Requested Changes:**

No required changes, but see the following questions, typos, and suggestions for v2 of the manuscript.

**Questions**

- [page 02] It seems that online learning is more general than the
  distributional learning setting (since we could each draw in the distributional
  setting as a problem instance chosen by the adversary). Can you comment on the
  difference in the paper, and whether or not you give stronger gaurantees in the
  distributional setting? It may be good to also compare to more practical _learning with predictions_ settings, e.g., [Mitzenmacher-Vassilvitskii, Communications of the ACM, 2022]
- [page 05] What are the fixed set of distance functions $\mathbf{\delta}$ in
  an instance of hierarchical clustering? I think this example in its current
  form creates more confusion than in clears up.
- [page 44] Is the proof of Theorem E.5 intentionally missing?

**Typos and suggestions**

- [page 05] $\alpha$ in the definition of utility class function should be bold.
- [page 05] Disregard if "dual utility function class" is a standard term in the literature, but something like "transpose utility function class" better describes the behavior
- [page 06] "(See Appendix A.1 ...)" --> "(see Appendix A.1 ...)"
- [page 06] Given that Theorem 3.1 has $\text{PDim}(\mathcal{U})$ in the hypothesis
  and is from 1974, it would be good to find the original source for pseudo-dimension,
  as the one given is from 2012.
- [page 07] In definition of Pfaffian chain: some of the $\eta_j$ should be $\eta_q$.
- [page 09] Add `\qedhere` after the last line in the proof of Theorem 4.2.
- [page 15] Consider using $= O(\cdots)$ instead of $\le O(\cdots)$ when the
  RHS uses big-O notation
- [page 16] In $d = \delta_\beta$, the $\beta$ should be boldfaced.
- [page 31] Typo: "... providing an upper-bound bound for ..." --> "...
  providing an upper bound for ..."

**Strengths And Weaknesses:**

**Strengths**

- Extends GJ framework to Pfaffian GJ framework (rational functions --> Pfaffian functions)
  * inspired by [Bartlett-Indyk-Wagner, COLT 2022]
  * builds on [Goldberg-Jerrum, COLT 1993]
- Introduces a general approach (Theorem 7.2, Theorem 7.3) for verifying the
  dispersion property [Balcan-Dick-Vitercik, FOCS 2018]
- The two examples in Section 4.1 are very helpful.
- Excellent applications: hierarchical clustering, graph-based SSL, regularized logistic regression
- Section 6 is a good tutorial for how to use the GJ Pfaffian framework (**highlight of the paper**)
- Section 7.3 shows how to apply the online learning ideas to agglomerative hierarchical clustering and regularized logistic regression

**Weaknesses**
- The online case assumes a large number of samples and compute to find good hyperparams,
  but it may be better in practice to run a portfolio solver with the same compute budget
  (i.e., trying lots of different hyperparameter combinations) to get a good solution for the
  sample at time $t$. This doesn't have the same provable gauarantees, but it's more practical.
- These are interesting theoretical results, but it's unclear how useful they are in practice.

---

> ### Author Response · Authors · 2025-02-27
> **Author response**
>
> We thank the reviewer for carefully reviewing our paper and providing constructive feedback. We are glad that the reviewer acknowledges the novelty and significance of our theoretical contribution. We will address the reviewer's concerns and requested changes as follows.
>
> ## Typos and suggested changes
> We thank the reviewer for pointing out the typos. We have updated the draft accordingly and marked it using **olive color**.
>
> ## Questions
>
> 1.  *On the difference between distributional and online learning settings and connection with Algorithm with predictions*:
>
>     a. *"It seems that online learning is more general than the distributional learning setting. What is the difference between distributional learning and online learning (comment on the paper)? Can the online learning result presented in the paper give us stronger guarantees in the distributional setting?"*: Even though the online learning setting is more general than the distributional setting, the results in the online learning setting need some “smoothness” assumptions, which are not made in the distributional setting. As noted in prior work (e.g., [1]), no-regret online learning for worst-case adversarial sequences is typically impossible, and additional assumptions are needed. We use the dispersion assumption from [2] (Definition 9). Roughly speaking, this means that the discontinuities of the dual function do not concentrate in any small region of the parameter space. So, by online-to-batch conversion arguments, our online learning results imply sample complexity guarantees, but only if the distribution satisfies dispersion. In contrast, our sample complexity results in the distributional setting do not make such an assumption and hold even if the discontinuities concentrate. **We explicitly added this point on page 24 in the paper.**
>
>     b. *Comparing this setting with the algorithm with predictions*: While both algorithms with predictions and our setting use machine learning to improve algorithm design, there are important differences that make the two approaches somewhat complementary. Concretely, under the algorithms with predictions paradigm, one designs better algorithms using predictions of some ML model. With a few exceptions, the ML model itself is fixed and unspecified, and there is no actual “learning” in this setting. In particular, there is typically no guarantee that the predictions are good (and performance gains are conditional on the predictions being good). With a few exceptions for very specific cases, these algorithms do not enjoy end-to-end guarantees, which remains an interesting open direction for further research. **We have added further discussion (see section Related works, page 5)** to clarify the distinction of our work from the algorithm with predictions literature.
>
> 2. *What is the fixed set of the distance function $\\boldsymbol{\\delta}$?*: In that example, the set of distance function $\\boldsymbol{\\delta}$ contains distance functions that we want to combine (via the weights $\\boldsymbol{a}$ that linearly interpolate the different functions in the set $\\boldsymbol{\\delta}$, here $\\boldsymbol{a}$ is the hyperparameter that we want to tune). The combined distance functions are then used for the linkage-based clustering algorithm. For example, $\\boldsymbol{\\delta}$ might contain different $\\ell_p$ distances for some value $p \\in [0, \\infty)$ or some other distance functions suitable for the set of points $S$. **We have elaborated on this part clearly in the revised draft (see page 5)**.
>
> 3.  *"Is the proof of Theorem E.5 intentionally missing?":* Theorem E.5 serves as a supporting lemma from prior work and is not a new result. Therefore, we did not incorporate its proof in our draft and pointed the readers to the original paper instead. **We added this point in the revised draft (page 45)**.
>
> ## Summary
>
> Overall, we thank the reviewer again for carefully reading our paper and providing detailed constructive
> feedback. As the reviewer suggested, we incorporated those points in the revised draft (marked in olive) to improve the clarity and presentation of the paper. We hope that our answers address the reviewer’s concerns, and we are happy to address any further questions by the reviewer.
>
> ## References
>
> 1. Balcan et al., How much data is sufficient to learn high-performing algorithms? generalization guarantees for data-driven algorithm design, JACM'24
>
> 2. Balcan et al.., Dispersion for Data-Driven Algorithm Design, Online Learning, and Private Optimization, FOCS'18

---

### Decision · Action_Editor_Y3Mm · 2025-04-21

**Recommendation:** Accept as is

**Comment:**

First of all, I would like to apologize to the authors for taking unusually long to complete the review procedure for this paper. Two factors have played into this: 1) the relative underrepresentation of the topic of the paper at TMLR and within the reviewer pool and 2) the length of the paper which was beyond the limits for regular papers at TMLR. Thanks to these challenges, it took quite long to find suitable reviewers for this paper.

Eventually, I have managed to find three expert reviewers who all wrote detailed and informative reviews about the paper. All of them have appreciated the contributions of the paper, especially after the meticulous updates made by the authors in response to the reviews. Given this strong support and my own reading of the paper, I am happy to concur with the reviewers and recommend this paper for acceptance at TMLR.

**Audience:**

The topic of the paper is somewhat niche for a venue like TMLR, but there is definitely an audience for it. Although this audience may be small at the moment, this may change in the future, especially if TMLR continues to publish excellent papers on the topic.

**Claims And Evidence:**

All claims made in the paper are supported by rigorous mathematical proofs.